# Modeling the spatio-temporal variability in subsurface thermal regimes across a low-relief polygonal tundra landscape

Jitendra Kumar[1], Nathan Collier[2], Gautam Bisht[3], Richard T. Mills[4], Peter E. Thornton[1], Colleen M. Iversen[1], and Vladimir Romanovsky[5]

[1]Environmental Sciences Division, Oak Ridge National Laboratory, Oak Ridge, TN, USA
[2]Computer Science and Mathematics Division, Oak Ridge National Laboratory, Oak Ridge, TN, USA
[3]Lawrence Berkeley National Laboratory, Berkeley, CA, USA
[4]Intel Corporation, Hillsboro, OR, USA
[5]Geophysical Institute, University of Alaska Fairbanks, AK, USA

*Correspondence to:* Jitendra Kumar (jkumar@climatemodeling.org)

**Abstract.** Vast carbon stocks stored in permafrost soils of Arctic tundra are under risk of release to the atmosphere under warming climate scenarios. Ice – wedge polygons in the low – gradient polygonal tundra create a complex mosaic of microtopographic features. This microtopography plays a critical role in regulating the fine scale variability in thermal and hydrological regimes in the polygonal tundra landscape underlain by continuous permafrost. Modeling of thermal regimes of this sensitive ecosystem is essential for understanding the landscape behavior under current as well as changing climate. We present here an end-to-end effort for high resolution numerical modeling of thermal hydrology at real–world field sites, utilizing the best available data to characterize and parameterize the models. We develop approaches to model the thermal hydrology of polygonal tundra and apply them at four study sites at Barrow, Alaska spanning across low to transitional to high-centered polygon, representing a broad polygonal tundra landscape. A multi – phase subsurface thermal hydrology model (PFLOTRAN) was developed and applied to study the thermal regimes at four sites. Using high resolution LiDAR DEM, microtopographic features of the landscape were characterized and represented in the high resolution model mesh. Best available soil data from field observations and literature were utilized to represent the complex heterogeneous subsurface in the numerical model. Simulation results demonstrate the ability of the developed modeling approach to capture—without recourse to model calibration—several aspects of the complex thermal regimes across the sites, and provide insights into the critical role of polygonal tundra microtopography in regulating the thermal dynamics of the carbon rich permafrost soils. Areas of significant disagreement between model results and observations highlight the importance of field–based observations of soil thermal and hydraulic properties for modeling – based studies of permafrost thermal dynamics, and provide motivation and guidance for future observations that will help address model and data gaps affecting our current understanding of the system.

## 1 Introduction

Coastal Arctic landscapes – dominated by wetlands and patterned ground–cover approximately 5 – 10% of Earth's land surface and play an important role in the hydrology, geomorphology, biogeochemistry and vegetation dynamics of the vast Arctic

region. The low-gradient topography of the polygonal tundra characteristic of these landscapes is a complex mosaic of micro-topographic features created by ice wedge polygons. This micro-topography leads to strong fine – scale variability in thermal and hydrological regimes of landscapes underlain by continuous permafrost. Permafrost landforms like drained-lakes, low-centered polygons and high-centered polygons retard surface runoff after snowmelt, leading to increased surface water storage

(in form of lakes, ponds and wetlands) (Kane et al. (2003)). Complex surface drainage patterns lead to heterogeneous soil moisture and substrate conditions supporting a wide range of vegetation composition across the landscape. Arctic tundra soil pools are estimated to contain 190 Pg of carbon (Post et al. (1982)), much of which is under risk of rapid release to the atmosphere in a warming climate. Hobbie et al. (2000) studied the controls over carbon storage and turn-over in Arctic soils and found temperature, micro-topography and vegetation composition to be the primary controls at regional scale.

Changes in the surface geomorphology which lead to the creation of ice-wedge polygons are induced by thermal dise-quilibrium and permafrost degradation. Lowland polygonal relief is dominated by low-centered polygons and high-centered polygons. Low-centered polygons are the most common polygonal landscape feature and are characteristic of poorly drained tundra. They consist of a raised rim with a wet central depression. Raised rims are the result of growing ice wedges that push material away from the center of the ice wedges to the sides (French (2007)). The standing water in the ice-wedge troughs

leads to thermal erosion (i.e., accelerated thawing) along the rim. This preferential thaw may cause the ridge to collapse and form trenches surrounding the polygon center, inverting the relief to form high-centered polygon. High-centered polygon are well-drained with often dry centers, leading to low peat accumulation and deeper active layers. The micro-topographic relief and associated heterogeneity in soil moisture support a diverse distribution of vegetation in the Arctic, with wet centers and troughs of low-centered polygon covered by mosses and sedges, while drier rims, and centers of high-centered polygon and

rims of low-centered polygon are dominated by mosses, lichen and dwarf shrubs. These diverse land-cover types can also alter the surface energy balance and thermal properties through changes in albedo, surface roughness, and evapotranspiration (Langer et al. (2011)).

Large scale climate and terrestrial ecosystem processes are represented in global to regional scale climate and ecosystem models. However, most of these models lack the representation of fine scale heterogeneity in surface and subsurface processes

at subgrid scale that exercise significant control on the landscape scale behavior. Representation of the fine scale heterogeneity is important to model the non-linear processes involved (Cresto Aleina et al. (2013)).

Accurate characterization and modeling of subsurface thermal regimes in polygonal tundra is critical for our understanding of this sensitive system and our ability to predict its fate under climate change. In this study we developed approaches to 1) characterize the surface micro-topography and subsurface structure of the polygonal tundra, 2) represent heterogeneous

subsurface stratigraphy and hydraulic and thermal properties, 3) numerically model permafrost hydrology, and 4) combine the above to simulate the permafrost thermal regime at field sites in a polygonal tundra region near Barrow, Alaska.

## 2   Study Area

The study area is located within the Barrow Environmental Observatory (BEO), (Figure 1) which lies 6 $km$ East of Barrow, Alaska (71°18'N, 156°35'W), and is a field site of the U.S. Department of Energy's Next Generation Ecosystem Experiments (NGEE) – Arctic project. The BEO spans 32.21 $km^2$ of natural tundra, lakes, and wetlands, and is reserved for scientific research. The landscape has low topographic relief, with elevations ranging from 0 to 7 $m$ above sea level and low hydraulic gradient present across the region. Barrow has a polar maritime climate with mean annual air temperature of -12.0°$C$ and 3.3°$C$ during summer (June – August) (Liljedahl et al. (2011)). The winter snowpack averages 20 to 40 $cm$, but snow accumulation is spatially variable due to variations in terrain roughness and drifting from strong easterly winds (Bockheim et al. (2001)). Annual adjusted precipitation is 173 $mm$, with the majority of precipitation falling during summer months (Liljedahl et al. (2011)). The polygonal tundra landscape is punctuated by thermokarst lakes and drained lake basins, with grass, moss, and sedge as dominant vegetation types. Basins at Barrow are underlain by permafrost within 1 $m$ of the surface and are classified as Gelisols, with an organic-rich surface layer underlain by a horizon of silt and clay to silt-loam textured mineral material and a frozen organic-rich mineral layer (Bockheim et al. (2001)). The seasonal active layer thickness ranges between 30 to 70 $cm$ at the BEO.

The various stages of geomorphologial and ecological change from low to transitional to high-centered polygon, lakes and drained lakes are all represented at the BEO. Following a "space for time" philosophy, NGEE–Arctic intensive field sites at BEO were chosen across the landscape to observe and study polygonal landscapes at all stages of transition. Table 1 shows the characteristics of four sites (A, B, C, D) where our current study is focused.

**Table 1.** Areas A, B, C, D polygonal features and environmental characteristics.

| Area | Characteristics | Relative Elevation (Min / Max / Median $m$) |
|------|-----------------|---------------------------------------------|
| A | Low center polygons (with ridges and troughs) | Low (4.5 / 4.9 / 4.6) |
| B | High center polygons | High (4.5 / 5.1 / 4.8) |
| C | Transitional low center polygons (with ridges and troughs) | Moderate (4.3 / 4.9 / 4.6) |
| D | Low center polygons (no troughs) | Low (4.1 / 4.6 / 4.3) |

Relative elevation is qualitative summary of topography in the region, while Min/Max/Median are minimum, maximum and median elevations.

A suite of observations are being collected at each of these intensive sites. Since 2012, meteorological data (including air temperature, summer precipitation, snow depth, relative humidity, wind speed and radiation) are being collected at all four sites (Hinzman et al. (2014b)). Surface temperature data is being collected along a transect from the center of the polygon to ridge to trough. Each location on the transect consists of nine soil temperature sensors ranging in depth from 2 $cm$ to 150 $cm$ (Romanovsky and Cable (2012)). Figure C.1 shows the observed time series of hourly air temperature and liquid precipitation at our four sites for the period of October 1, 2013 – September 30, 2014. Data for the one year period (October 1, 2013 – September 30, 2014) were chosen for this study since it was the only complete year for which all the necessary observations

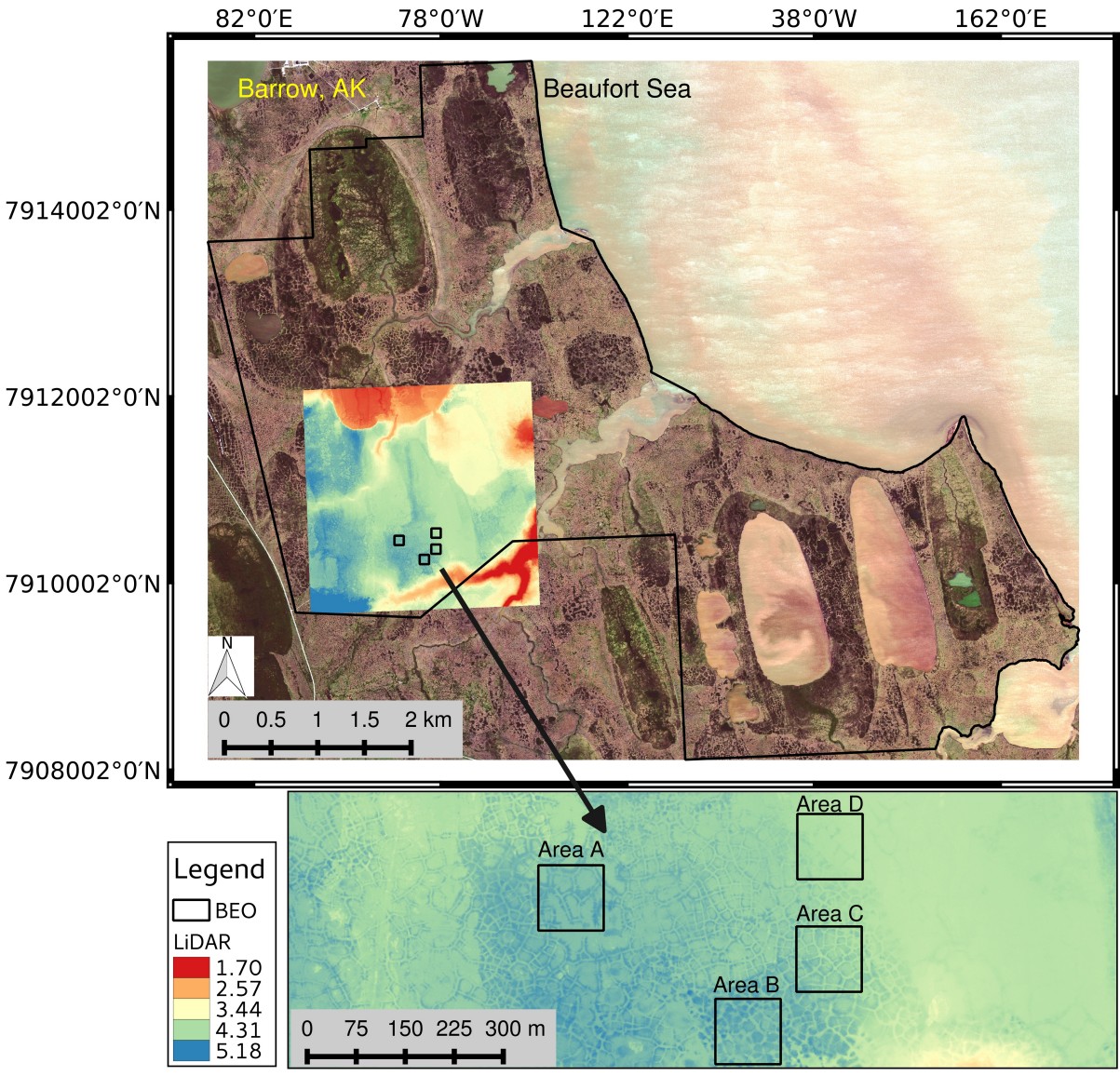

**Figure 1.** NGEE-Arctic Field Sites at BEO (LiDAR elevations are in unit of meter above mean sea level)

were available. Figure 2a, d, g, j shows high resolution imagery of the sites where the observations have been collected. Figure 2a, d, g, j shows the boundary of the region around the intensive sites where we conducted the detailed modeling study presented here.

## 3 Methodology

To model the thermal regimes of the heterogeneous polygonal tundra ecosystem we developed approaches to 1) characterize the surface micro-topography and subsurface structure of the polygonal tundra 2) represent heterogeneous subsurface stratigraphy and hydraulic and thermal properties, and 3) numerically model permafrost hydrology.

### 3.1 Representation of landscape heterogeneity

Accurate representation of polygonal tundra in the model requires 1) identification of micro-topographic features on the landscape and 2) characterization of soil stratigraphy and properties across the landscape.

#### 3.1.1 Identification of polygonal features

The human eye can discern polygonal patterns and features in satellite high resolution satellite imagery with relative ease. However, automated recognition and delineation of polygonal features are challenging due to the variability in their spectral appearance, irregularity of polygon shape, dimension and orientation and lack of unique spectral signatures associated with the features (Skurikhin et al. (2013)). Muster et al. (2012) investigated the subpixel heterogeneity in Landsat satellite imagery over ice-wedge polygonal tundra using a range of multi-scale data (field measurements and remote sensing) and concluded that resolutions of 4 $m$ or less are necessary to map the fine-scale landscape elements of polygonal tundra. Skurikhin et al. (2013) used a combination of segmentation and shape-based classification approaches using high-resolution WorldView-2 satellite imagery (60 $cm$ resolution) to identify the landscape elements within the BEO. While they reported an overall accuracy of 95%, their study region was limited to a 1000 × 1100 pixels subimage. The scalability of such a specialized algorithm based on high-resolution satellite imagery (of limited availability) is untested and difficult for application for landscape–scale studies like ours.

Thus, with landscape–scale application in mind, we employed a relatively simple and generic approach using a high resolution Digital Elevation Model (DEM) that exploits the relative difference in surface elevations that distinguish the polygonal features (center, ridges and troughs). High-resolution LiDAR data (25 $cm$ resolution) were collected on October 4, 2005 by Tweedie (2010). The LiDAR data horizontal and vertical accuracy were approximately 30 and 15 $cm$ respectively. Covering an approximately 2.5 $km$ × 2.5 $km$ area, the data set encompasses all of the NGEE-Arctic intensive sampling sites (Figure 1) where our study was focused. Using a high resolution DEM created from this data set, elevation contours (10 $cm$ interval) were developed to segment and classify the landscape in centers, ridges and troughs.

- *Site A*: Elevation at Site A ranged from a minimum of 4.5 $m$ to a maximum of 4.9 $m$ (Figure 2(a)). Low elevation depressions (4.5 − 4.6 $m$) were classified as Center, surrounded by elevated rim (4.7 − 4.9 $m$), and deep troughs ad-

jacent to them ($4.6 – 4.7\ m$). Center, Rim and Troughs occupied approximately 35%, 24% and 41% area respectively (Figure 2(b)).

- *Site B*: At Site B with high-centered polygons, elevation ranges from $4.5 – 5.1\ m$ (Figure 2(c)). High elevation areas ($4.8 – 5.1\ m$) were classified as Center, low elevation ($4.7 – 4.8\ m$) rim and deep troughs ($4.5 – 4.7\ m$), occupying 39%, 21%, and 40% of total area respectively (Figure 2(d)).

- *Site C*: A wider range ($4.3 – 4.9\ m$) of elevations are present at Site C (Figure 2(e)) which are often considered *low* or flat-centered polygon. Low elevation areas ($4.5 – 4.6\ m$) were classified as Center, with raised rims ($4.6 – 4.9\ m$) and deep ($4.3 – 4.5\ m$) troughs. Center, Rim and Trough occupied 35%, 36%, and 29% of the total area (Figure 2(f)).

- *Site D*: Site D is relatively flat and is thus identified as flat-centered polygons, with the entire area within a narrow elevation range of $4.1 – 4.6\ m$. While polygonal features were evident in $0.25\ m$ resolution aerial optical image (Figure **??**(d)), they were difficult to identify in the LiDAR DEM (Figure 2(g)) due to the limitations of the vertical accuracy of LiDAR. Trough features in flat-centered polygons are not well pronounced. Thus the area was classified only as Center ($4.1 – 4.3\ m$) and Rim ($4.3 – 4.6\ m$) features. About 72% of the area was classified as Center while 28% as Rim (Figure 2(h)).

We did not apply any specialized rules to enforce any shape, dimension and/or patterns of the polygon features (Center, Ridge, Trough), allowing us to scale our approach to the entire region where high resolution DEMs were available.

### 3.1.2 Subsurface characterization

The structure and properties of subsurface soils are important factors controlling the pattern and variability of permafrost thermal processes in the tundra environment, and accurate characterization and representation of the heterogeneous subsurface properties is critical to understanding and modeling the subsurface thermal dynamics. However, the limited availability of soil properties in tundra environments and at our sites at the BEO presents a significant challenge.

During the period July 31 – August 3, 2012 a field campaign was conducted by NGEE-Arctic researchers to collect soil cores at one replicate polygon at each of the sites A, B, C, and D at three micro-topographic positions (Center, Ridge, Trough) per polygon. Cores were collected using a hammer and a $5.08\ cm$ diameter corer to collect one soil core per location to a depth of $30\ cm$. The soil horizons (moss, organic layer, mineral layer) were measured for each core to the nearest centimeter. A deep organic layer was found at several of the locations. However the total depth of the deep organic layer was not determined if it extended beyond the $30\ cm$ core depth. Cores were collected for the purpose of biogeochemical analysis and thus no soil hydraulic or thermal properties were measured by the team. Figure 3 illustrates the subsurface soil horizons based on observations that we used in all our modeling studies presented here. In absence of co-located observations for soil hydraulic and thermal properties we derived the data for use in our studies from the published literature in tundra regions (Hinzman et al. (1991), Hinzman et al. (1998)) and a recent parameter calibration study conducted at one of our sites (site C) (Atchley et al. (2015)). Table D.1 shows the soil hydraulic and thermal properties used in our modeling study.

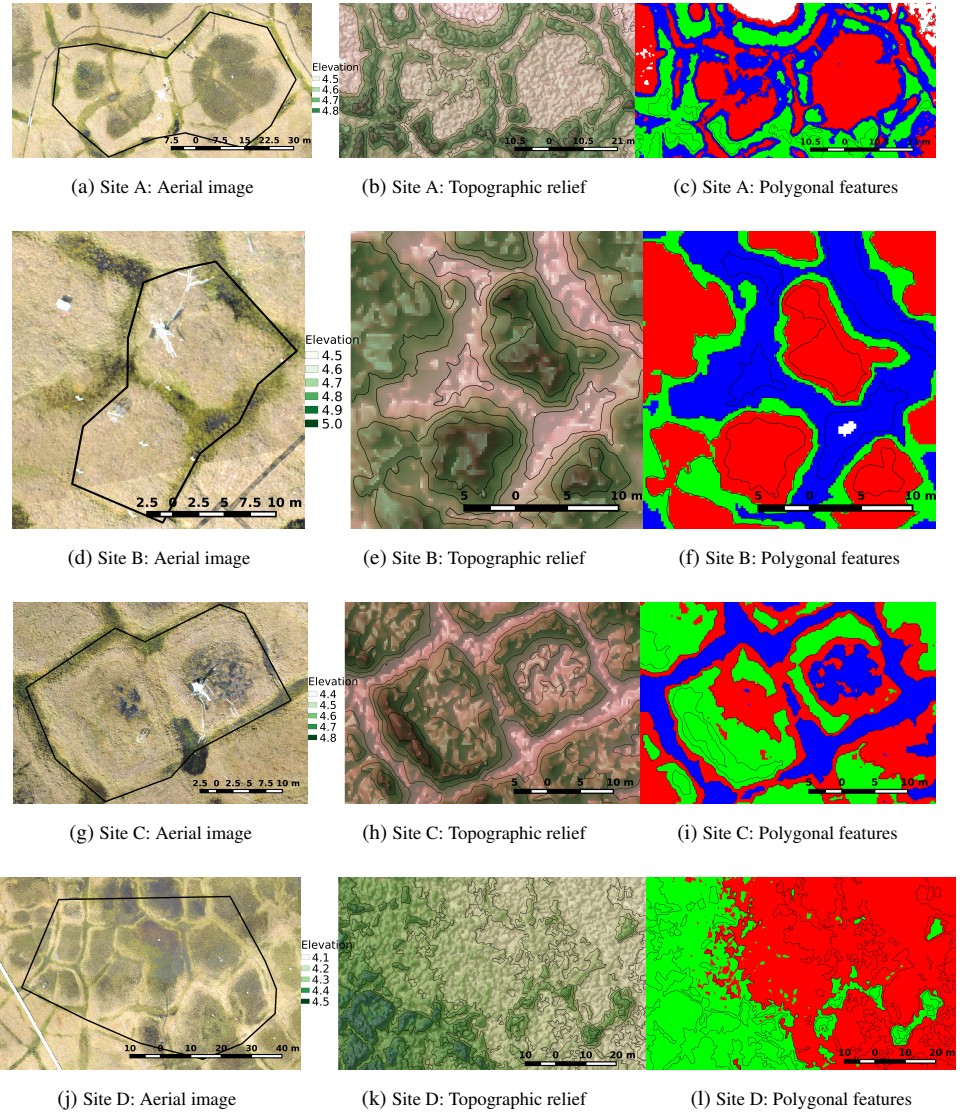

**Figure 2.** Elevation contour based classification of the study areas at Sites A, B, C, D. In subfigures (c, f, i, and l) the colors reflect polygon type (red: center; green: ridge; blue: trough).

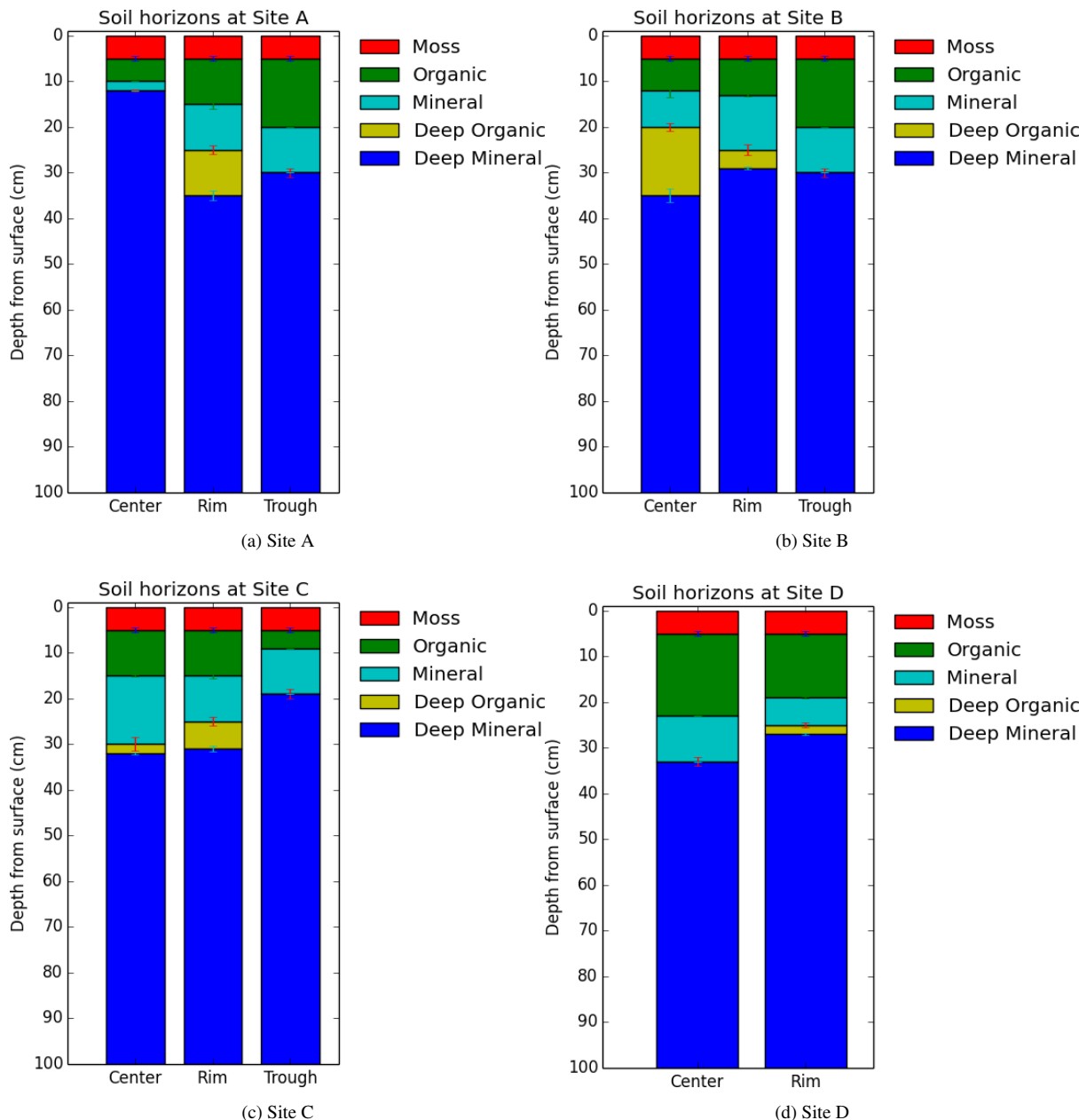

**Figure 3.** Subsurface soil horizons across micro-topographic positions at Sites A, B, C, D based on field observations. Models at study sites were paramterized using these data.

**Table 2.** Mesh resolution in vertical column. Finer resolution was used in the layers closer to the surface, while a coarser resolution was used in deeper soils.

| Vertical depth | Resolution |
|:---:|:---:|
| $0 - 0.5\ m$ | $5\ cm$ |
| $0.5\ m$ - $1.0\ m$ | $10\ cm$ |
| $1.0\ m - 5.0\ m$ | $25\ cm$ |
| $5.0\ m - 10.0\ m$ | $50\ cm$ |
| $10.0\ m - 50.0\ m$ | $1.0\ m$ |

### 3.1.3   Development of micro-topography resolving computational mesh

Collier and Kumar (2016) developed MeshMaker, a Python-based meshing framework to create high resolution computational meshes for use in numerical simulation of permafrost thermal hydrologic processes at our polygonal tundra study sites. The meshing framework uses a high-resolution DEM and landscape classification (Figure 2) to develop Triangulated Irregular Networks (TINs). Non-uniform locally refined TINs adapt to the topographic complexity to create fine-resolution elements in areas with sharp changes in topography while creating coarser elements elsewhere, thus creating a high quality micro-topography resolving mesh (Figure 5). Variable resolution in vertical column was employed as described in Table 2.

Data from Sections 3.1.1 and 3.1.2 were embedded within the generated meshes (Figure 4) to represent the heterogeneity in the thermal hydrology models. By overlaying the TIN mesh with classified maps from Section 3.1.1, micro-topographic position (Center/Ridge/Trough of a polygon) of each element in the mesh was identified. Polygon type and micro-topographic specific soil horizons data (Section 3.1.2) were used to determine the soil horizons in the model mesh. While our data set was limited to a single replicate for each polygon type and location, significant spatial heterogeneity exists in reality. We assumed a variability of 10% in soil horizon (moss, organic, mineral and deep organic soil) depths and stochastically generated the soil horizon depths at each spatial location in the modeling domain.

### 3.2   Three phase model for permafrost hydrology

In this study, we will use the open–source code PFLOTRAN to model the flow of mass and energy in the subsurface. PFLO-TRAN (Hammond et al. (2016, 2014)) is a state-of-the-art, massively parallel subsurface flow and reactive transport code. PFLOTRAN solves a system of generally nonlinear partial differential equations (PDEs) describing multiphase, multicomponent and multiscale reactive flow and transport in porous materials. The PDEs are spatially discretized using a finite volume technique, and backward Euler scheme is used for implicit time discretization. One system of PDEs which PFLOTRAN implements is a three-phase, thermal-hydrology model (the TH process model in PFLOTRAN parlance) which describes a balance of mass

$$\frac{\partial}{\partial t}\left[\varphi\left(s_\ell\eta_\ell + s_i\eta_i + s_g\eta_g\right)\right] + \nabla\cdot(\eta_\ell\mathbf{q}_\ell) = Q_M \tag{1}$$

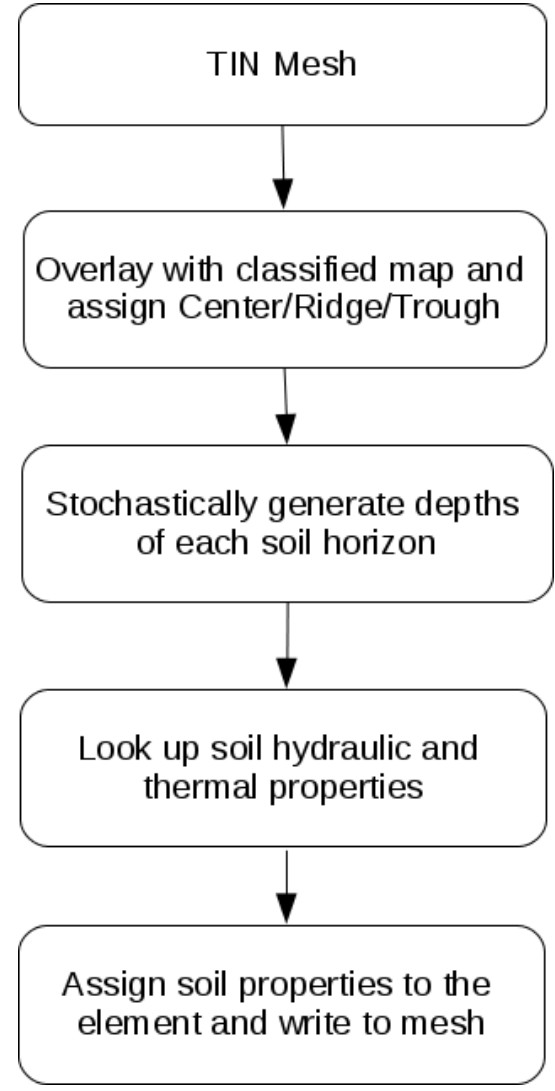

**Figure 4.** Workflow for heterogeneous soil parameter assignment in the computational mesh.

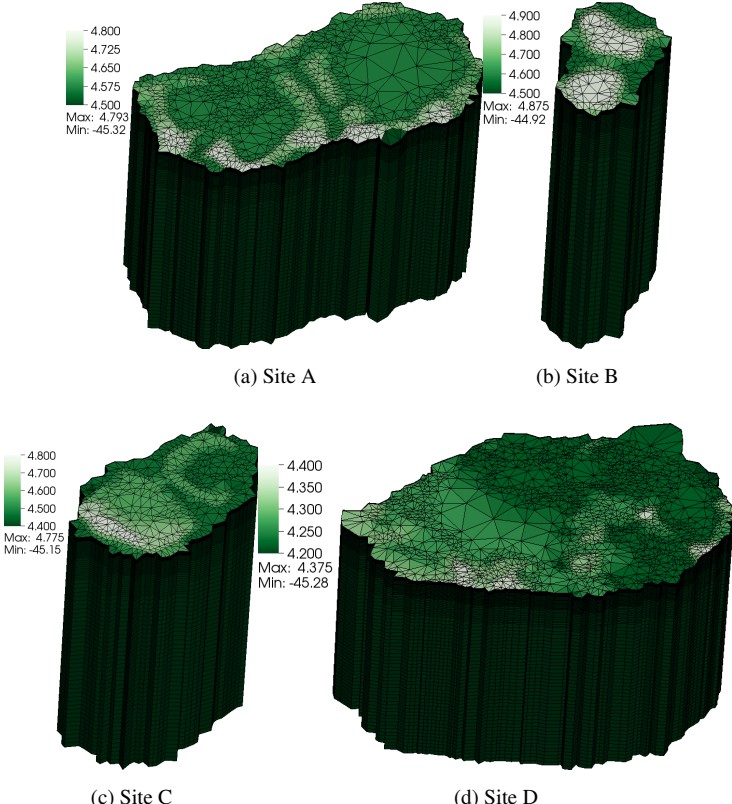

(a) Site A      (b) Site B

(c) Site C      (d) Site D

**Figure 5.** Micro-topography resolving unstructured meshes. The coloring reflects the ground surface elevation. Variable resolution mesh adapts to complex topography and uses finer resolution in areas of strong relief and coarser resolution in simpler terrain.

and energy,

$$\frac{\partial}{\partial t}\left[\varphi\left(s_\ell\eta_\ell U_\ell + s_i\eta_i U_i + s_g\eta_g U_g\right) + (1-\varphi)\rho_s c_s T\right]$$
$$+\nabla\cdot\left(\eta_\ell \mathbf{q}_\ell H_\ell - \kappa_{eff}\nabla T\right) = Q_E \tag{2}$$

in which the liquid pressure $P_\ell$ and the bulk temperature $T$ are the primary unknown variables. In equations (1) and (2), $\varphi$ refers to porosity, $s$ to percent saturation, $\eta$ to molar density, $U$ to internal energy, $\rho$ to mass density, $c$ to specific heat, and $H$ to enthalpy. The subscripts $\{\ell,g,i\}$ refer to the liquid, gas, and ice phases of water, respectively, and the subscript $s$ to the soil matrix. The Darcy velocity is given by,

$$\mathbf{q}_\ell = -\frac{kk_r}{\mu_\ell}\nabla\left(P_\ell - \rho_\ell gz\right) \tag{3}$$

where $k$ denotes intrinsic permeability, $k_r$ relative permeability, $\mu$ viscosity, $g$ unsigned gravity, and $z$ the vertical component of the position vector $\mathbf{x}$. The effective thermal conductivity is expressed as

$$\kappa_{eff} = K_i\kappa_i + K_\ell\kappa_\ell + (1 - K_i - K_\ell)\kappa_g \tag{4}$$

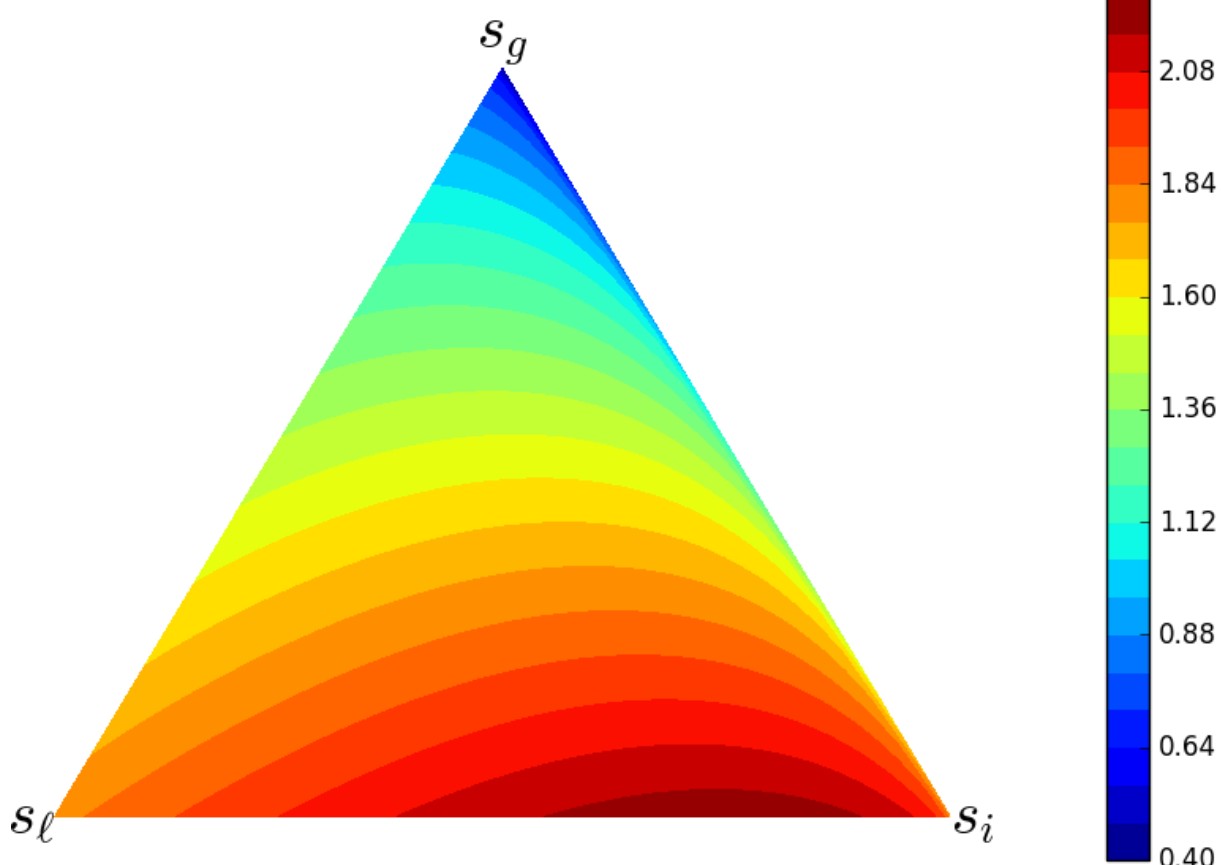

**Figure 6.** Model dependence of effective thermal conductivity on liquid ($s_\ell$), ice ($s_i$) and gas phase ($s_g$) fraction of water

where $\kappa$ are the thermal conductivities of each pure phase and $K$ represents the Kersten number of the frozen and unfrozen phase,

$$K_i = (s_i + \epsilon)^{\alpha_i} \tag{5}$$

$$K_\ell = (s_\ell + \epsilon)^{\alpha_\ell} \tag{6}$$

5   where $\epsilon = 1 \times 10^{-6}$ and $\alpha_i$ and $\alpha_\ell$ are parameters of the assumed power law. Figure 6 shows the modeled dependence of effective thermal conductivity ($\kappa_{eff}$) on fraction of water present in liquid ($s_\ell$)), ice ($s_i$) and gas ($s_g$) phases.

    The variables $Q_M$ and $Q_E$ represent generic mass and energy sources and sinks. We emphasize that the saturations, densities, and internal energies are all nonlinear functions of the liquid pressure and temperature and include the latent heat of fusion associated with change of phase. We also note that PFLOTRAN implements several choices of constitutive models for relating

10   the saturations to the liquid pressure and bulk temperature. While only a brief overview of the numerical formulation has been provided here, we would refer to Painter (2011), Painter and Karra (2014), and Karra et al. (2014) for detailed discussion of the formulation. Key parameters for the model relevant for current study are described in Table D.1.

### 3.3 Initial and boundary conditions

3-D subsurface models for each of the four sites were initialized by freezing the entire modeling domain at a temperature of -1.0 °C. The models were spun up to a thermal periodic steady state using a time series of mean daily temperatures applied to the top of the domain (ground surface). Spin up simulations were conducted for a period of by cycling annual time series of forcing. Spin up simulations were continued until a periodic steady state was achieved (i.e. close to zero inter annual variability in annual thermal regime). Spin up duration of 10 years was used at all the sites and was determined to be sufficient. We conducted a series of initialization simulations by varying initial temperatures at start of spin up and found them to not have any significant impact on the final periodic steady state, besides simulation period required to reach that steady state.

Mean daily near-surface temperature time series for period October 1, 2013 – September 30, 2014 were derived from hourly in situ temperatures from sensors located at 2 $cm$ depths. At all four sites, using sensors installed at center, ridge and troughs, three different time series were prepared. Using the classification (center, ridge, trough) embedded in the model (Section 3.1.1), these micro-topography specific temperature time series were applied in a spatially heterogeneous, micro-topography aware fashion to simulate the complex thermal hydrologic regimes in permafrost soils at the BEO. A no–flow boundary condition was applied to the sides of the domain, while the deep bottom boundary was held at constant -10 °C, based on the temperature from West Dock site (Figure 3 of Romanovsky et al. (2010)) which is located at a comparable latitude (70.4°N) to the BEO (71.29°N). It would be important to note that due to lack of observations of drainage pattern at the site we opted to use a no–flow boundary condition, however, surface runoff in and out of the region occurs in reality. Thus the soil moisture states were not very well constrained in the model and also has consequences for the simulated thermal regime. Surface processes (such as vegetation cover and snow) play an important role in regulating the thermal regimes of permafrost soils. While the surface processes are not represented in our model, use of near ground surface (2 $cm$ depth) temperature as the boundary condition for the simulation allows us to isolate (though not completely) the effect of surface processes.

## 4 Modeling permafrost thermal hydrology

### 4.1 Simulation of permafrost thermal regimes

After the models were spun up to periodic steady state condition, the simulation was continued for another year and outputs were used for validation and analysis. Soil temperature observations from the thermal sensors at 2 $cm$ depths at the sites for the period October 1, 2013 – September 30, 2014 were used to drive the time dependent (Dirichlet) boundary condition at the top (ground surface) of the model. In addition, mean daily time series (October 1, 2013 – September 30, 2014) of liquid (summer time) precipitation was also applied as moisture input to the model. Groundwater infiltration was considered to be zero if ground surface temperature was below freezing or if the domain was fully saturated. While soil moisture plays an important role, the focus of this study was on thermal regime and thus all results and discussions presented are focused on soil temperature. Section E3 show the spatial pattern of maximum water table depth across study region and exhibits strong correlation with micro-topography. Soil temperature data from all the sensors (sixteen sensors at depths from 5 $cm$ to 150 $cm$)

**Table 3.** Model performance statistics at Site A compared to observed soil temperatures (for annual, winter and summer periods). Model shows higher bias during summer season compared to winter.

| Depth from surface [$m$] | Center | | | Rim | | | Trough | | |
|---|---|---|---|---|---|---|---|---|---|
| | $RMSE$ | $R^2$ | $Bias$ | $RMSE$ | $R^2$ | $Bias$ | $RMSE$ | $R^2$ | $Bias$ |
| | Annual [Oct. 1, 2013 - Sept. 30, 2014] | | | | | | | | |
| 0.05 | 0.73 | 0.99 | -0.06 | 0.80 | 0.99 | 0.13 | 0.91 | 0.99 | 0.48 |
| 0.10 | 0.85 | 0.98 | 0.03 | 0.72 | 0.99 | 0.31 | 1.13 | 0.98 | 0.67 |
| 0.50 | 1.30 | 0.94 | 0.39 | 1.01 | 0.99 | 0.72 | 1.35 | 0.98 | 1.16 |
| 1.50 | 1.42 | 0.94 | 1.01 | 1.55 | 0.99 | 1.52 | 1.95 | 0.99 | 1.92 |
| | Winter [Oct. 1, 2013 - May 31, 2014 | | | | | | | | |
| 0.05 | 3.54 | 0.85 | -2.45 | 1.96 | 0.96 | 1.30 | 2.71 | 0.91 | 1.76 |
| 0.10 | 3.40 | 0.85 | -2.36 | 1.89 | 0.96 | 1.24 | 2.66 | 0.91 | 1.71 |
| 0.50 | 2.91 | 0.80 | -1.82 | 1.79 | 0.98 | 1.32 | 2.25 | 0.95 | 1.78 |
| 1.50 | 1.91 | 0.79 | -0.44 | 2.04 | 0.99 | 1.95 | 2.39 | 0.97 | 2.30 |
| | Summer [June 1, 2014 - Sept. 31, 2014] | | | | | | | | |
| 0.05 | 1.58 | 0.74 | -1.18 | 1.40 | 0.60 | 0.58 | 1.97 | 0.78 | 1.61 |
| 0.10 | 0.84 | 0.81 | -0.59 | 1.69 | 0.50 | 1.25 | 2.44 | 0.62 | 2.08 |
| 0.50 | 1.27 | 0.38 | 1.20 | 1.75 | 0.12 | 1.62 | 2.43 | 0.09 | 2.29 |
| 1.50 | 1.95 | 0.97 | 1.93 | 1.82 | 0.97 | 1.81 | 2.30 | 0.97 | 2.29 |

$RMSE$ = Root Mean Squared Error, $R^2$ = Coefficient of determination, $Bias = negative$ bias indicates cold bias in the model while $positive$ indicates a warm bias

were used to evaluate the accuracy of the models. While we have selected sensors at four depths 5 $cm$, 10 $cm$, 50 $cm$ and 150 $cm$ for discussion the results at all depths can be found in Section E1.

At all the sites, simulated soil temperatures in the top most soil layer (5 $cm$ thick) compared well with the observed near surface temperature at 5 $cm$ depth (Figures 7(a), 8(a), 9(a), 10(a)). Simulated soil temperatures in deeper soils show warm bias increasing with depth as compared to the observed temperatures at the sensors (coefficient of determination $R^2$ 0.93 – 0.99) (Tables 3, 4, 5, 6). Model matched the observations with a Root Mean Square Error ($RMSE$) of 0.60 – 0.99 $°C$ near surface, with an increasing errors at deeper soils. While modeled temperature bias was in range of -0.30 – 0.10 $°C$ near surface, a warm bias of up to 1.0 – 1.8 $°C$ was found at deep soils. Warm bias in the model was more pronounced during summer season, compared to winter season.

Figures 7, 8, 9, 10 shows the comparison between the simulated and observed soil temperatures at several select depths (5 $cm$, 10 $cm$, 50 $cm$, 150 $cm$ from surface) at sites A, B, C, and D respectively. Simulated temperatures across all the sites matched well with the observed temperatures at shallow depths, but showed a deviation towards warmer than observed temperatures in deep soils. A number of factor may be contributing to this bias in simulations, including applied boundary

**Table 4.** Model performance statistics at Site B compared to observed soil temperatures (for annual, winter and summer periods). Model shows higher bias during summer season compared to winter.

| Depth from surface [$m$] | Center | | | Rim | | | Trough | | |
|---|---|---|---|---|---|---|---|---|---|
| | $RMSE$ | $R^2$ | $Bias$ | $RMSE$ | $R^2$ | $Bias$ | $RMSE$ | $R^2$ | $Bias$ |
| Annual [Oct. 1, 2013 - Sept. 30, 2014] | | | | | | | | | |
| 0.05 | 0.83 | 0.99 | 0.34 | 0.50 | 0.99 | -0.08 | 0.68 | 0.99 | 0.23 |
| 0.10 | 0.80 | 0.99 | 0.38 | 0.46 | 0.99 | 0.04 | 0.72 | 0.99 | 0.31 |
| 0.50 | 1.02 | 0.98 | 0.71 | 0.98 | 0.98 | 0.39 | 1.23 | 0.98 | 0.81 |
| 1.50 | 1.89 | 0.96 | 1.49 | 1.95 | 0.92 | 1.34 | 2.09 | 0.94 | 1.63 |
| Winter [Oct. 1, 2013 - May 31, 2014 | | | | | | | | | |
| 0.05 | 0.77 | 0.99 | 0.29 | 0.88 | 0.99 | -0.55 | 3.25 | 0.86 | -2.10 |
| 0.10 | 0.75 | 0.99 | 0.30 | 0.83 | 0.98 | -0.51 | 2.98 | 0.86 | -1.93 |
| 0.50 | 0.61 | 0.99 | 0.40 | 0.82 | 0.97 | -0.20 | 2.18 | 0.86 | -1.12 |
| 1.50 | 1.40 | 0.96 | 1.02 | 1.47 | 0.91 | 0.74 | 1.84 | 0.80 | 0.25 |
| Summer [June 1, 2014 - Sept. 31, 2014] | | | | | | | | | |
| 0.05 | 0.99 | 0.73 | 0.48 | 1.14 | 0.68 | 0.62 | 0.87 | 0.76 | -0.11 |
| 0.10 | 0.93 | 0.76 | 0.62 | 1.44 | 0.58 | 1.11 | 0.66 | 0.81 | 0.26 |
| 0.50 | 1.73 | 0.10 | 1.63 | 1.96 | 0.11 | 1.87 | 1.81 | 0.13 | 1.73 |
| 1.50 | 2.87 | 0.94 | 2.87 | 2.98 | 0.96 | 2.98 | 3.09 | 0.95 | 3.09 |

$RMSE$ = Root Mean Squared Error, $R^2$ = Coefficient of determination, $Bias = negative$ bias indicates cold bias in the model while $positive$ indicates a warm bias

conditions (Section 3.3). We believe that insufficient characterization and parameterization of heterogeneous properties due to limited data availability is one of the key reasons for this bias. Model was evolved to a periodic thermal steady state through a spin up process and soil temperatures in vertical profile are strongly dependent on the soil properties. Forced with surface temperature boundary condition, while the soil temperature in the model after spin up stage was close to the observed, a warm bias was observed in the deeper soils. That warm bias was carried over to the final stage of the simulation resulting in bias in the simulated soil temperatures and thaw depths reported here. With soil cores collected at the sites limited to top 30 $cm$ of the soil, our understanding of structure and physical and thermal properties of deeper soils is limited. For example, while we know that presence of ground ice (like ice wedges, segregated ice, ice lens etc.) is common in subsurface of Arctic tundra, their representation in the model is completely missing. Lack of representation of these cryostructures are potentially one of the reasons for warmer soils in our simulations. While PFLOTRAN has the ability to capture and model such cryostructures (via heterogeneous subsurface structure and properties but not their formation and evolution), we lack any quantitative data to characterize them for representation in the model. Ongoing efforts under NGEE–Arctic project by Kneafsey and Ulrich (2016)

**Table 5.** Model performance statistics at Site C compared to observed soil temperatures (for annual, winter and summer periods). Model shows higher bias during summer season compared to winter.

| Depth from surface [$m$] | Center | | | Rim | | | Trough | | |
|---|---|---|---|---|---|---|---|---|---|
| | $RMSE$ | $R^2$ | $Bias$ | $RMSE$ | $R^2$ | $Bias$ | $RMSE$ | $R^2$ | $Bias$ |
| | Annual [Oct. 1, 2013 - Sept. 30, 2014] | | | | | | | | |
| 0.05 | 0.80 | 0.99 | 0.18 | 1.84 | 0.96 | 0.50 | 0.69 | 0.99 | 0.29 |
| 0.10 | 0.63 | 0.99 | 0.12 | 1.23 | 0.98 | 0.35 | 0.78 | 0.99 | 0.34 |
| 0.50 | 0.96 | 0.97 | 0.42 | 0.75 | 0.99 | 0.42 | 1.13 | 0.98 | 0.91 |
| 1.50 | 1.35 | 0.99 | 1.27 | 1.23 | 0.99 | 1.19 | 1.81 | 1.00 | 1.76 |
| | Winter [Oct. 1, 2013 - May 31, 2014 | | | | | | | | |
| 0.05 | 1.32 | 0.98 | -0.88 | 1.70 | 0.96 | 0.77 | 0.93 | 0.98 | -0.36 |
| 0.10 | 1.26 | 0.98 | -0.92 | 1.09 | 0.98 | 0.43 | 0.95 | 0.98 | -0.35 |
| 0.50 | 1.22 | 0.96 | -0.69 | 0.51 | 0.99 | 0.03 | 0.63 | 0.98 | 0.20 |
| 1.50 | 0.88 | 0.97 | 0.57 | 0.93 | 0.99 | 0.88 | 1.34 | 0.99 | 1.27 |
| | Summer [June 1, 2014 - Sept. 31, 2014] | | | | | | | | |
| 0.05 | 2.04 | 0.57 | -1.34 | 1.93 | 0.45 | -0.91 | 0.84 | 0.79 | 0.28 |
| 0.10 | 1.58 | 0.64 | -1.13 | 1.23 | 0.55 | -0.56 | 0.80 | 0.83 | 0.56 |
| 0.50 | 0.67 | 0.67 | 0.61 | 0.86 | 0.58 | 0.80 | 1.66 | 0.31 | 1.58 |
| 1.50 | 1.70 | 0.98 | 1.70 | 1.52 | 0.98 | 1.52 | 2.33 | 0.98 | 2.33 |

$RMSE$ = Root Mean Squared Error, $R^2$ = Coefficient of determination, $Bias = negative$ bias indicates cold bias in the model while $positive$ indicates a warm bias

and Dafflon et al. (2016) using X-ray computed tomography (CT) scanner technology on ice cores from BEO can potentially provide detail 3-dimensional soil structure and density information and help address this missing piece.

Spatial variability in soil temperatures was observed in the simulations (Figures 7, 8, 9, 10) arising in part due to three dimensional heat flow and heterogeneous subsurface structure and soil properties represented in the model. Simulated soil temperatures also show a seasonal pattern of spatial variability with high variability during the cold winter season and lower spatial variability during summer. Figure 11 shows time series of spatial variability (standard deviation) in soil temperature at Site A during the simulation period, showing strong seasonality, the magnitude of which is reduced at deeper soils. Similar patterns of variability was observed at Site B, C, and D (Figures E.1, E.2, E.3).

Heat flow in the permafrost soils which are frozen for a significant part of the year occurs primarily due to conduction. Thermal conductivity of the soil is sensitive to the temperatures and thus the fraction of water present in liquid vs ice phase (Equation (4), Figure 6). Effective thermal conductivity ($\kappa_{eff}$) of the soil is higher during the winter months when almost the entire soil domain is in a frozen state (thus high ice saturation ($s_i$) and low liquid saturation ($s_\ell$)), compared to summer months when active layer is in thawed state. Higher conductivity of the soil and thus higher conductive heat flows during the winter

**Table 6.** Model performance statistics at Site A compared to observed soil temperatures (for annual, winter and summer periods). Model shows higher bias during summer season compared to winter.

| Depth from surface [m] | Center | | | Rim | | |
|---|---|---|---|---|---|---|
| | RMSE | $R^2$ | Bias | RMSE | $R^2$ | Bias |
| Annual | | | | | | |
| 0.05 | 0.80 | 0.99 | 0.34 | 0.86 | 0.99 | 0.26 |
| 0.10 | 1.03 | 0.98 | 0.45 | 0.96 | 0.98 | 0.36 |
| 0.50 | 1.49 | 0.95 | 0.86 | 1.02 | 0.98 | 0.57 |
| 1.50 | 1.65 | 0.97 | 1.47 | 1.29 | 0.98 | 1.14 |
| Winter | | | | | | |
| 0.05 | 0.59 | 0.99 | 0.17 | 0.64 | 0.99 | 0.12 |
| 0.10 | 0.68 | 0.99 | 0.12 | 0.65 | 0.99 | 0.10 |
| 0.50 | 0.62 | 0.99 | 0.27 | 0.61 | 0.99 | 0.22 |
| 1.50 | 1.32 | 0.98 | 1.17 | 1.03 | 0.98 | 0.89 |
| Summer | | | | | | |
| 0.05 | 1.21 | 0.83 | 0.85 | 1.30 | 0.64 | 0.69 |
| 0.10 | 1.68 | 0.77 | 1.42 | 1.54 | 0.55 | 1.15 |
| 0.50 | 2.76 | 0.06 | 2.61 | 1.73 | 0.15 | 1.60 |
| 1.50 | 2.35 | 0.97 | 2.35 | 1.86 | 0.97 | 1.85 |

$RMSE$ = Root Mean Squared Error, $R^2$ = Coefficient of determination, $Bias = negative$ bias indicates cold bias in the model while $positive$ indicates a warm bias

season in a heterogeneous soil domain leads to higher temperature variability in model simulations of the permafrost thermal regimes at our sites.

During the summer season, advective heat flow processes occur within the thawed soil layers. The study sites also receive liquid precipitation and thus infiltration during the summer season (Figure C.1) which leads to vertical as well as horizontal flows in the thawed soil. For example, at Site A horizontal velocities are close to zero during the early summer (Figure 12(a)) when soil temperatures are close to freezing. After the ground has thawed, liquid precipitation events during summer (Figure 12(b, c)) lead to significant lateral flows. High elevation rim regions drain to the center and trough of the low-centered polygons.

Large summer bias in the model is partly due to the ill-constrained flow boundary conditions and lack explicit representation of surface flow processes. As ground thaws with the rise of summer temperature, significant lateral flow occurs even in the absence of rainfall events (Figure 12(d)). No flow boundary condition applied in our model prevents any runoff out of the modeling domain, leading to increased soil moisture (Figure E.4) in low elevation areas where the higher warm bias in the model is observed (Figure 13, 14, 15, 16).

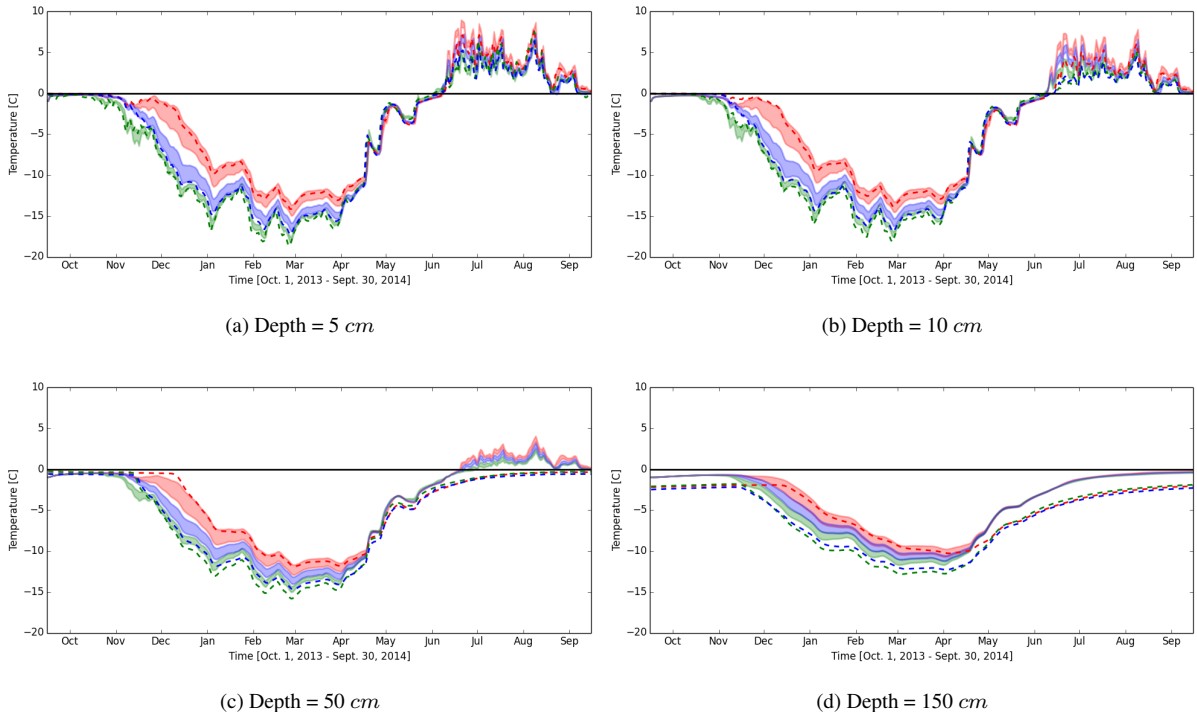

(a) Depth = 5 $cm$

(b) Depth = 10 $cm$

(c) Depth = 50 $cm$

(d) Depth = 150 $cm$

**Figure 7.** Simulated vs observed soil temperatures at Site A at depths 5 cm, 10 cm, 50 cm and 150 cm show bias in model in deeper soil and during warm summer season.

dotted lines represent observed data at center (in red), rim (green), and trough (blue) locations, while shaded curves show mean +/- standard deviation of simulated daily soil temperatures across the domain.

A reference 1-Dimensional simulation was conducted at Site A (described in Section E4) to isolate and understand the role of lateral flows. Simulations show that ignoring lateral flows can lead to a temperature bias of -0.55 - 0.85 $^\circ C$ across the site depending on the microtopography (Figure E.5).

In permafrost environment, the active layer is the top layer of soil that thaws during the summer and freezes again during
5  the autumn. While PFLOTRAN solves for soil temperature as primary variable, we derived active layer depth (or thaw depth) as sum of thickness of soil layers above freezing temperature (0°C). Figure 13, 14, 15, 16 show the temporal dynamics of thaw depth during the period of simulation and spatial pattern of maximum thaw depth across the region. A wide spatial variability in the thaw depth was observed in the simulations which were strongly correlated to the micro-topography. The variability is primarily derived by the micro-topography and the subsurface heterogeneity. The warm bias in soil temperature in the model
10  translates to a bias towards deeper thaw depths (0.60 m – 1.0 m) as compared to the observations (0.30 m – 0.74 m) at the site (Section F, F.1).

1-D reference simulations (Section E4) demonstrates the importance of 3-D representation of thermal and hydrologic processes to capture the effect of micro-topography. While, the lateral flow may not significantly effect the average thaw depths across a large region (Figure E.7(a)), they lead to micro-topography dependent differences in thaw depths (Figures E.7(b,c,d))

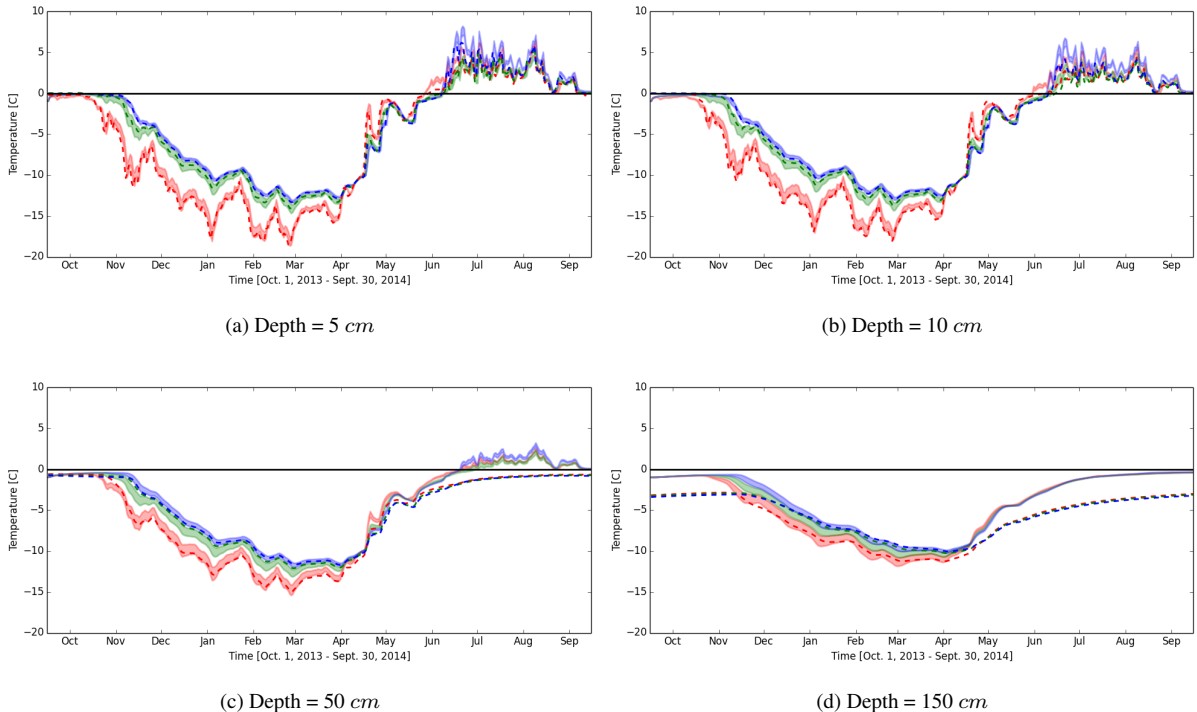

(a) Depth = 5 $cm$             (b) Depth = 10 $cm$

(c) Depth = 50 $cm$             (d) Depth = 150 $cm$

**Figure 8.** Simulated vs observed soil temperatures at Site B at depths 5 cm, 10 cm, 50 cm and 150 cm show bias in model in deeper soil and during warm summer season.

dotted lines represent observed data at center (in red), rim (green), and trough (blue) locations, while shaded curves show mean +/- standard deviation of simulated daily soil temperatures across the domain.

throughout the thaw season, which may have an implication for other ecological processes like biogeochemistry and vegetation dynamics.

### 4.2 Understanding the thermal regimes of polygonal tundra

Micro-topography of the polygonal tundra exerts critical controls on the flow of water and energy at local to regional scales
5 which further influences the ecological and biogeochemical processes on the landscape. Surface processes (not modeled in our study) like vegetation and snow cover also play a critical role in regulating the subsurface thermal regimes through thermal insulation effects. In our modeling approach we represented the micro-topographic features Center, Rim and Trough across four low to transitional to high-centered polygons. PFLOTRAN successfully simulated the overall pattern of thermal regimes in Center, Rim and Trough across the four sites A, B, C, and D, though a significant warm bias is observed in deeper soils.
10 (Table 1).

- *Site A*: Site A is located in a poorly drained region dominated by low-centered polygons with low elevation centers, raised rims and troughs. Center areas are warmer than rim and trough areas while rims are coldest (Figure 7). Centers

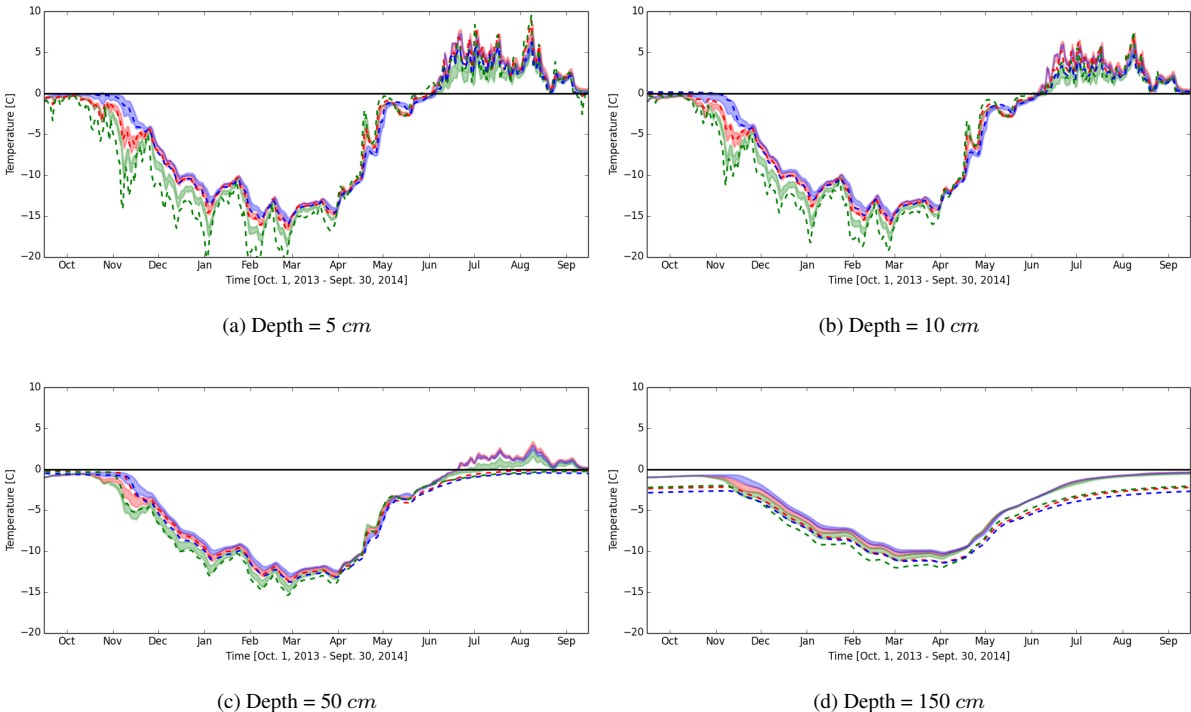

(a) Depth = 5 cm

(b) Depth = 10 cm

(c) Depth = 50 cm

(d) Depth = 150 cm

**Figure 9.** Simulated vs observed soil temperatures at Site C at depths 5 cm, 10 cm, 50 cm and 150 cm show bias in model in deeper soil and during warm summer season.

dotted lines represent observed data at center (in red), rim (green), and trough (blue) locations, while shaded curves show mean +/- standard deviation of simulated daily soil temperatures across the domain.

in low-centered polygon are often inundated and relatively wet (Figure E.4) most of the year and support vegetation (mosses and sedges). Low elevation centers also receive higher snow cover. Vegetation and snow cover provide thermal insulation to the ground keeping the center region warmer compared to rim and trough. Dry rims (Figure E.4) with low vegetation cover and low snow accumulation (Figure G.1) are most exposed to the winter temperatures and are thus the coldest.

3-Dimensional representation of the processes is important to simulate these microtopography driven processes. 1-D reference simulation (Section E4) show that the lack of full 3-D representation can lead to topography driven bias in simulated thermal regimes (Figure E.7).

– *Site B*: Site B is dominated by well drained high-centered polygons with relatively dry elevated centers and deep troughs (Figure E.4) . Vegetation in high-centered polygons are dominated by lichens, moss and dwarf shrubs. In contrast to the low-centered polygons, centers and dry-tundra graminoids have low vegetation and snow cover (Figure G.1), are most exposed to the changes in air temperatures, and are thus colder than rim and trough which show relatively warmer soil temperature regimes during the winter. (Figure 8)

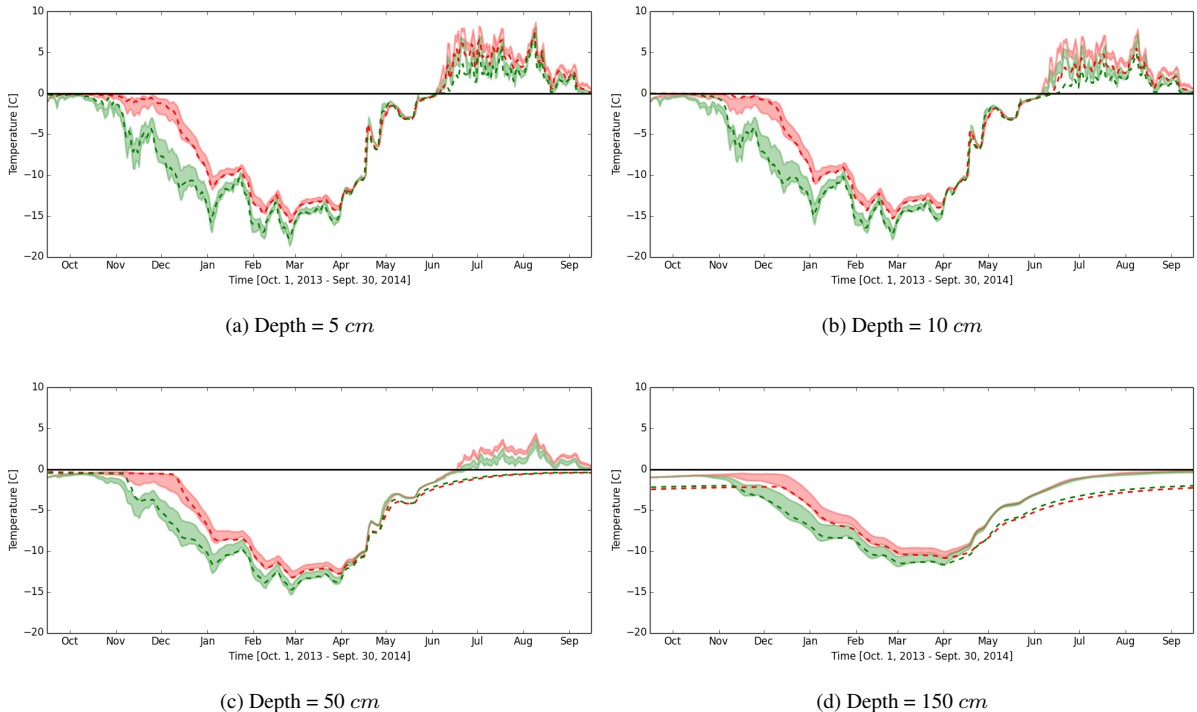

**Figure 10.** Simulated vs observed soil temperatures at Site D at depths 5 cm, 10 cm, 50 cm and 150 cm show bias in model in deeper soil and during warm summer season.

dotted lines represent observed data at center (in red), rim (green), and trough (blue) locations, while shaded curves show mean +/- standard deviation of simulated daily soil temperatures across the domain.

- _Site C_: Site C is located in an area of geomorphological transition from low to high-centered polygons, characterized as flat-centered polygons. They consist of shallow flat centers, deep troughs and raised rim regions. Soils in deep troughs are thermally insulated by higher snow cover (Figure G.1) and thus show warmer soil temperature regimes compared to centers and rims (Figure 9). Center and rim regions show similar thermal regimes with centers being slightly warmer due to higher vegetation and snow cover.

- _Site D_: Site D is characterized as low centered polygons with no pronounced rims. Site D is wettest among the four study areas, with low elevation center areas that remain inundated for most of the summer season (Figure E.4). While snow accumulation was fairly uniform (Figure G.1) across the flat region, vegetation cover plays an important role. Wet centers supports rich vegetation, leading to a warmer soil temperatures as compared to the trough regions (Figure 10).

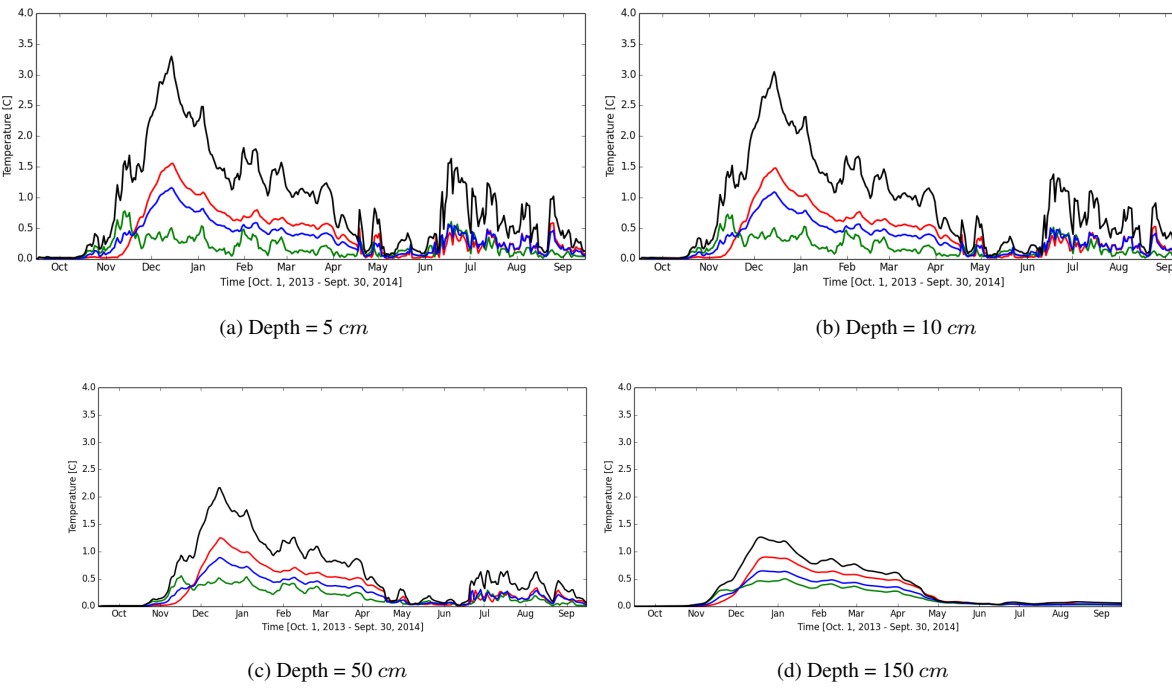

(a) Depth = 5 $cm$

(b) Depth = 10 $cm$

(c) Depth = 50 $cm$

(d) Depth = 150 $cm$

**Figure 11.** Spatio temporal variability (standard deviation) in simulated soil temperature at Site A at depths 5 cm, 10 cm, 50 cm, and 150 cm. High variability in soil temperatures was observed across the region during the year, especially cold winters with centers showing highest variability.

Red, green and blue lines represent center, rim and trough respectively, and black line represent the standard deviation across the Site A.

## 5 Model uncertainties and limitations

### 5.1 Why not to calibrate

Accurate simulation of permafrost thermal regimes requires the mechanistic representation of thermal hydrologic processes in the model. However, equally important is the accurate representation of subsurface structure and soil properties, model param-
5 eters and initial and boundary conditions. Given the lack of co-located observations for the soil properties, in this study we used soil thermal and hydraulic properties data from a different tundra site based on Hinzman et al. (1998). While PFLOTRAN was able to simulate the thermal hydrologic processes and match fairly well the soil temperature observations at the sites across a range of polygonal landscape and micro-topography features, simulated soil temperatures show deviations from the observed temperatures at times. Simulated temperatures show warm bias in deep soils where data for soil characterization and properties
10 are almost completely missing. Parameter calibration is a popular technique that has been widely used in hydrologic modeling to determine model parameters and properties to optimize the model fit to target observations. The high resolution 3-D PFLO-TRAN thermal hydrology model used in this study includes many degrees of freedom and parameters, which combined with complex non-linearity of hydrologic processes poses a complex high dimensional optimization problem. While a wide range of

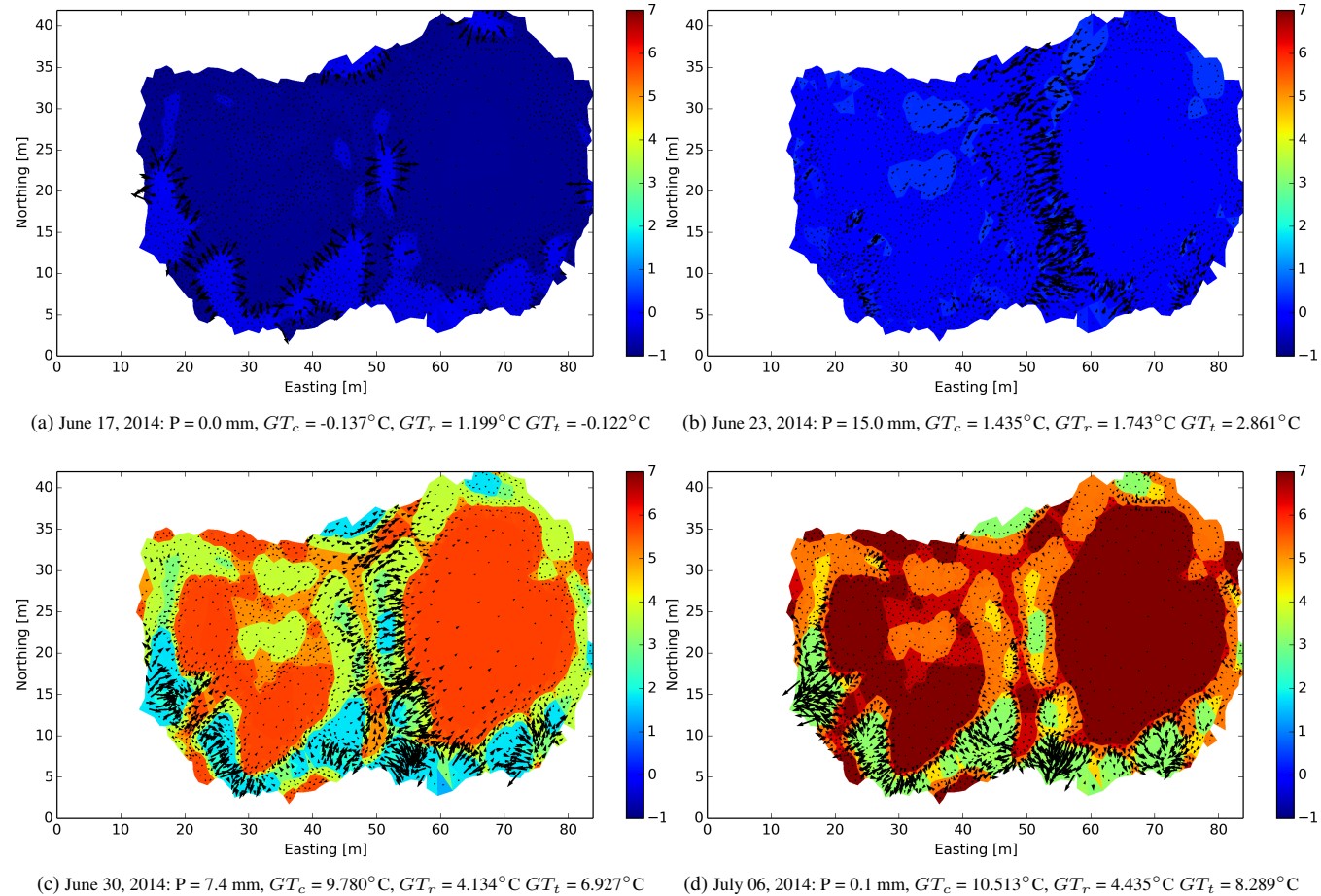

(a) June 17, 2014: P = 0.0 mm, $GT_c$ = -0.137°C, $GT_r$ = 1.199°C $GT_t$ = -0.122°C

(b) June 23, 2014: P = 15.0 mm, $GT_c$ = 1.435°C, $GT_r$ = 1.743°C $GT_t$ = 2.861°C

(c) June 30, 2014: P = 7.4 mm, $GT_c$ = 9.780°C, $GT_r$ = 4.134°C $GT_t$ = 6.927°C

(d) July 06, 2014: P = 0.1 mm, $GT_c$ = 10.513°C, $GT_r$ = 4.435°C $GT_t$ = 8.289°C

**Figure 12.** Lateral water flow velocity fields at Site A during early summer. In the horizontal cross section plot for Site A, background color shows the soil temperature distribution while the vector arrows show the magnitude and direction of lateral flow fields. (P = total daily precipitation, $GT_c$, $GT_r$, $GT_t$ = mean ground surface temperature at center, rim and trough respectively.

calibration approaches (Heuvelmans et al. (2006), Madsen (2000), Shafii and De Smedt (2009), Singh and Minsker (2008)) are available to determine optimal model parameters to fit the observed data (soil temperatures in this study), we face the problem of non-uniqueness (equifinality). A diverse set of possible parameter values can lead to similar model performance (Beven and Freer (2001)). The issue of non-uniqueness is especially pronounced in tundra ecosystem due to poor availability of data and thus poor bounds on parameters which leads to a high degree of uncertainty in the models with or without calibration.

While systematic calibration can help identify effective parameters for the model, transfer of parameters across models and modeling domains is difficult (Bárdossy, 2007). At data limited study sites like ours, while a ill-constrained calibration may compensate for lack of data, it does not improve our understanding of the system. In this study we choose not to calibrate

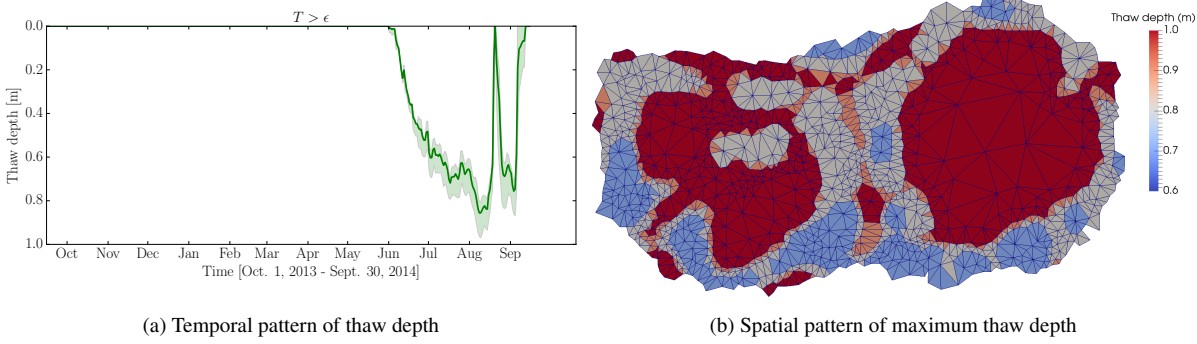

(a) Temporal pattern of thaw depth
(b) Spatial pattern of maximum thaw depth

**Figure 13.** Temporal (left) patterns of simulated thaw depths (defined as thickness of soil layers above $0°C$) and spatial pattern of maximum thaw depth (right) at Site A. Bold line represents the mean thaw depth across the site, while shaded curve represents standard deviation

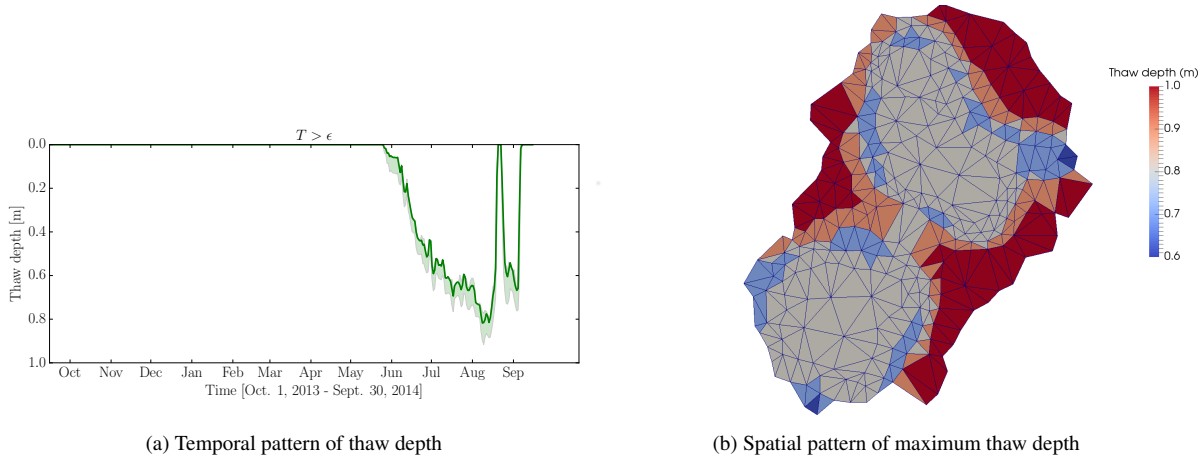

(a) Temporal pattern of thaw depth
(b) Spatial pattern of maximum thaw depth

**Figure 14.** Temporal (left) patterns of simulated thaw depths (defined as thickness of soil layers above $0°C$) and spatial pattern of maximum thaw depth (right) at Site B. Bold line represents the mean thaw depth across the site, while shaded curve represents standard deviation

the model parameters to achieve better fit with the observations, instead use the uncalibrated results to diagnose the potential model deficiencies and identify the characterization and data needed to better represent the real world in our simulations.

When properly constrained using observed data, the model calibration is a powerful tool that can provide improved physical representation of processes and parameters and new insights. Atchley et al. (2015) and Harp et al. (2016) has successfully demonstrated the use of such techniques for three phase hydrology models at one of our sites (site C) and would inform our future studies.

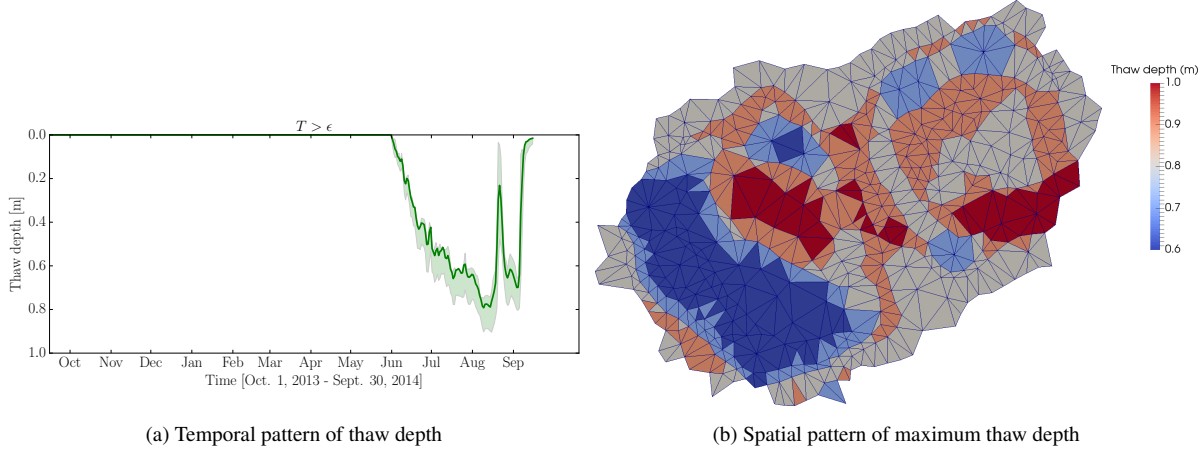

(a) Temporal pattern of thaw depth      (b) Spatial pattern of maximum thaw depth

**Figure 15.** Temporal (left) patterns of simulated thaw depths (defined as thickness of soil layers above $0°$C) and spatial pattern of maximum thaw depth (right) at Site C. Bold line represents the mean thaw depth across the site, while shaded curve represents standard deviation

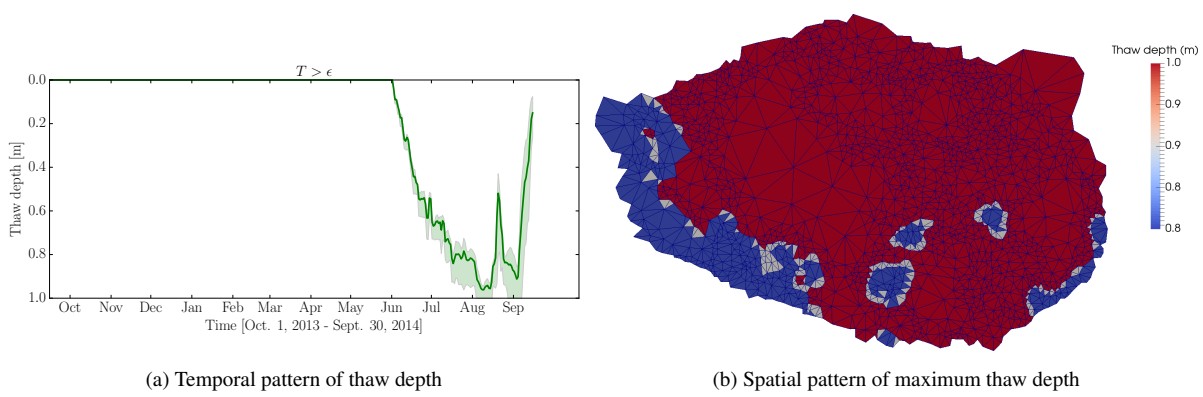

(a) Temporal pattern of thaw depth      (b) Spatial pattern of maximum thaw depth

**Figure 16.** Temporal (left) patterns of simulated thaw depths (defined as thickness of soil layers above $0°$C) and spatial pattern of maximum thaw depth (right) at Site D. Bold line represents the mean thaw depth across the site, while shaded curve represents standard deviation

## 5.2 Identifying model and data gaps

While the agreement between modeled and observed simulated soil temperature demonstrates the ability of the model to simulated the thermal hydrologic processes in the polygonal tundra, disagreements help us identify the existing gaps in data and model.

5      Modeling results highlight the need for co-located measurements of soil thermal and hydraulic properties for accurate modeling of hydrologic processes. While most soil core observations, including those used in our study, are focused on the shallow active layer, characterization of deeper permafrost soils is essential for understanding the thermal regimes and potential changes expected under warming climate. Modeling soil temperatures, beyond the high level estimation of thaw depth (or active layer thickness) is important to understand the thermal regime of permafrost soil and its behavior under warming conditions. For

example, during winter seasons even when the soils are completely frozen, variability in in soil temperatures (Figure 7, 8, 9, 10) exist and may impact carbon fluxes from the system (Zona et al. (2016)). Warmer than observed soil temperature in deep soils in our thermal periodic steady state solutions are due to inaccurate soil characterization and poorly bounded boundary conditions at the bottom of the modeling domain. Heat flux observations in deep permafrost, while hard to measure, would help provide

accurate bounds for the thermal hydrology model. In addition to rainfall events (which was captured in our models), surface drainage processes provides inputs to the ground water system (not captured in our models). Surface drainage observations in the local catchments (not available to us) are needed to appropriately model and constrain this process.

## 6   Summary and conclusions

Low-relief polygonal tundra ecosystems consist of micro-topographic features that controls the local scale hydrology. The wa-

ter and energy flow patterns on the landscape in turn regulate biogeochemical processes and vegetation dynamics. The objective of this study was to develop an end-to-end modeling approach for landscape scale modeling of permafrost thermal hydrology of real world polygonal tundra sites, improving our ability to model and to understand the patterns of thermal regimes. Using the best available data for our study sites we developed techniques to characterize the polygonal micro-topographic features and represent the heterogeneous soil hydraulic and thermal properties at the sites. These data sets were embedded within

topography–following high–resolution meshes to simulate the thermal hydrologic processes in PFLOTRAN thermal hydrology model. We employed detailed surface meteorology and subsurface soil temperature observations from the site to simulate and analyze the thermal regimes at four representative sites in polygonal tundra ecosystem at the Barrow Environmental Observatory.

Our modeling–based study reveals the role of micro-topographic features in regulating the permafrost thermal dynam-

ics across heterogeneous polygonal tundra landscape. Simulation results at four sites across the polygonal tundra landscape demonstrate the viability of the developed mechanistic approach to model the thermal regimes. Thermal regimes of center, rim and trough features of polygonal tundra exhibit distinct patterns in low to transitional to high-centered polygon landscape, which are governed by the micro-topography, surface and subsurface hydrology and surface processes (like air temperature, snow cover, and vegetation). Our reference 1-D model simulation demonstrate that while 1-D approaches may be sufficient to

understand mean thermal behavior of a large landscape, detailed 3-Dimensional process representation is required to capture micro-topography controls at high resolution. Our PFLOTRAN–based modeling approach was able to simulate these overall patterns at four study sites. Comparing the simulated soil temperatures against available observations, the model demonstrates the ability to simulate the soil temperature at shallow depths, but deviations from observations in deep soils highlight the need for better soil characterization using deep cores in these ecosystems. Our study also highlights the need for co-located

observations for accurate modeling and understanding of the tundra landscape. Model disagreements with the observations in this study may partially be due to use of soil properties from literature in absence of site–based measurements. Under the NGEE–Arctic project we are working with field scientists for improved co-located measurements. The present study was limited to a single year for which we had all the necessary data for model forcings and validation available, and thus was not able

to address the role of interannual variability. We plan to address this important problem as more data from our sites become available. While we have not addressed all the deficiencies in model process representation and parameterization identified and reported here in this study, we believe we have developed and presented a process rich modeling framework as a first critical step that would enable such studies. The modeling approach developed in this study—when combined with needed co-located observations—promises to allow accurate modeling of permafrost thermal hydrology and provide a framework for identifying and guiding the future observations required for improved modeling and understanding of the polygonal tundra ecosystem.

In a warming world, wet low-centered polygon landscapes are expected to go through geomorphological change to drier high-centered polygonn landscapes. The modeling approach developed in this study is a first step towards enabling future investigations of the impact of thermal hydrologic changes in these landscapes under projected climate scenarios. The model developed here does not have the ability to simulate the dynamic changes in micro-topography expected due to ice-wedge degradation (Liljedahl et al. (2016)). While beyond the scope of the current study, ongoing developments in biogeochemical modeling within PFLOTRAN (Tang et al. (2015)) in combination with our thermal hydrology model developments will also allow modeling of the terrestrial carbon cycle in this sensitive landscape under future warming scenarios.

While the knowledge gained by developing and evaluating fine-scale 3D simulations is valuable from the perspective of increased understanding of complex process interactions, the explicit long-term goal of the NGEE–Arctic project is to improve predictions of Arctic ecosystem processes at scales relevant to coupled climate and Earth system simulation. One element of our strategy to migrate knowledge across scales is to improve the grid and sub-grid representations in the land model component of our Earth system models to capture observed modes of variability in physical, biological, and biogeochemical processes. For example, our new top-level grid topology for global-scale land modeling follows watershed boundaries instead of the typical and arbitrary rectangular gridcell arrangement (Tesfa et al. (2014)). Sub-grid schemes are being developed that represent topographic variation within basins, and our goal is to apply those methods in the micro-topographic setting of polygonal tundra to capture the variation in thermal, hydrologic, and biogeochemical regimes, and interactions with vegetation communities. The current study is one step toward identifying the relevant modes of variation among diverse landforms in the polygonal tundra region. Another element of our scaling strategy is to use, to the full extent possible, a common set of modeling tools to construct simulations at various spatial scales. Even though many processes that can be represented explicitly at the finest scales (such as lateral flows of energy and water) must be parameterized for efficiency in a larger-scale simulation, having a common underlying set of equations helps to reduce unintentional loss of information across scales due, for example, to aggregation and disaggregation operators.

**Data availability**

Model source codes, input, forcings and output data sets from the study are publicly available at http://dx.doi.org/10.5440/1184018 (Kumar et al. (2016)).

**Author contributions**

JK designed the study and carried out the simulations. NC, GB, JK and RM developed the thermal hydrology model in PFLOTRAN. NC and JK developed the meshing, characterization and parameterization techniques. CI collected and analyzed the soil cores and assisted with vegetation characterization. VR designed and maintained meteorological station and thermal
sensors at the sites. JK led the analysis and writing of manuscript with contributions from NC, GB, RM, PT, CI and VR.

**Acknowledgments**

The Next Generation Ecosystem Experiments (NGEE) – Arctic project (http://ngee-arctic.ornl.gov/) is supported by the Office of Biological and Environmental Research in Department of Energy Office of Science. This manuscript has been authored by UT-Battelle, LLC under Contract No. DE-AC05-00OR22725 with the U.S. Department of Energy. The United States Gov-
ernment retains and the publisher, by accepting the article for publication, acknowledges that the United States Government retains a non-exclusive, paid-up, irrevocable, world-wide license to publish or reproduce the published form of this manuscript, or allow others to do so, for United States Government purposes. The Department of Energy will provide public access to these results of federally sponsored research in accordance with the DOE Public Access Plan (http://energy.gov/downloads/doe-public-access-plan). This research used resources of the Oak Ridge Leadership Computing Facility at the Oak Ridge National
Laboratory, which is supported by the Office of Science of the U.S. Department of Energy under Contract No. DE-AC05-00OR22725.

## Appendix A:  Model Reproducibility

Reproducibility, rigour, transparency and independent verification are cornerstones of the scientific method (Nature (2014)). We document here model/software and computational platform used in the work reported in this study.

*Software:*

Subsurface thermal hydrology modeling was conduced using PFLOTRAN, which is an open source, state-of-the-art massively parallel subsurface flow and reactive transport code. PFLOTRAN (http://www.pflotran.org/) code is developed under GNU LGPL license and publicly available at https://bitbucket.org/pflotran.

```
PFLOTRAN version used:
Repository URL: https://bitbucket.org/pflotran/pflotran-dev
Changeset: 18ec488fc6ac
```

PFLOTRAN employs parallelization through domain decomposition using the MPI-based PETSc framework with *pflotran-dev* tracking the *git maint* branch of PETSc available through https://bitbucket.org/petsc/petsc.

```
PETSc version used:
Repository URL: https://bitbucket.org/petsc
Changeset: c41c7662de68b036bda6be236f939e8b55959cb0
Version: v3.5.2-137-gc41c766
```

*Computational platform*: All simulations were conducted on the *Titan Cray XK7* at Oak Ridge Leadership Computing Facility (https://www.olcf.ornl.gov/computing-resources/titan-cray-xk7/). GNU compilers were used to compile PFLOTRAN and PETSc.

## Appendix B:  Data archiving and distribution

Model input files for all simulations reported along with the forcing data files, and computational mesh are publicly available through NGEE–Arctic long term data archive http://dx.doi.org/10.5440/1184018 (Kumar et al. (2016)). Summary outputs and statistics presented in various figures in this article are also available as part of the data collection. While the long term archiving of complete PFLOTRAN simulation outputs reported here was not possible due to large data volume, they can be obtained by contacting the lead author.

## Appendix C:  Meteorology at the study site

Figure C.1 show the observed hourly time series of liquid precipitation and air temperatures at the study sites A, B, C and D.

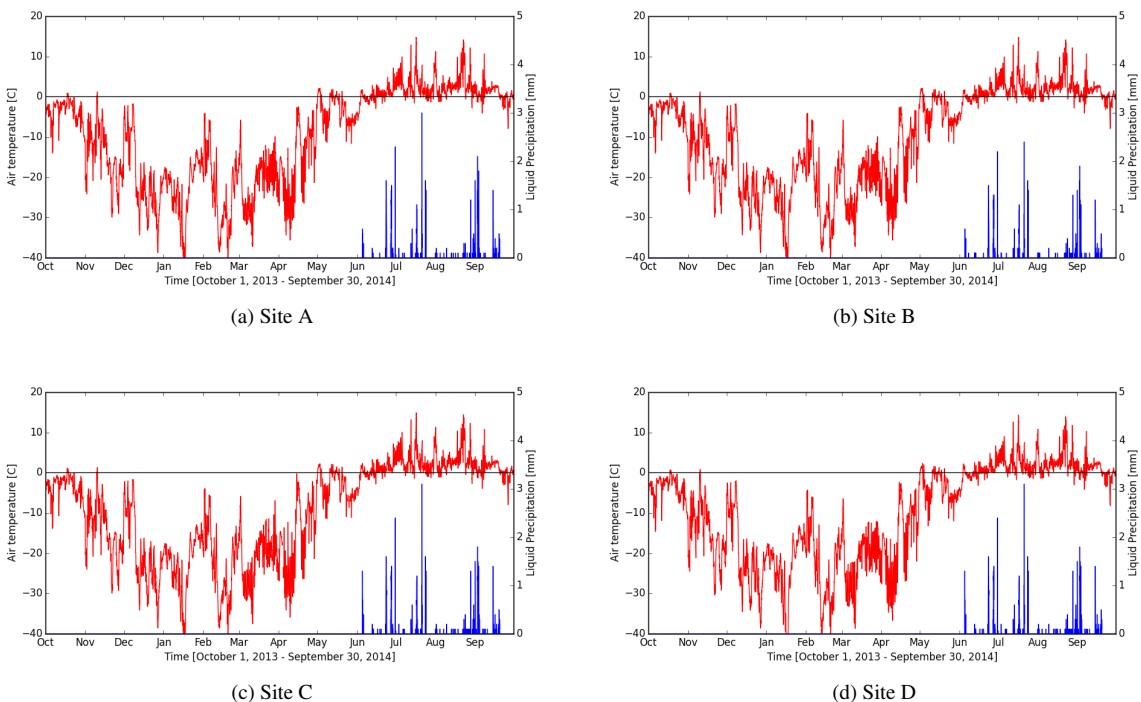

**Figure C.1.** Meteorological conditions at study sites. Red lines: hourly air temperature, Blue histograms: hourly summer time liquid precipitation

## Appendix D: Soil hydraulic and thermal properties

Table D.1 shows the soil hydraulic and thermal properties used in our modeling study.

## Appendix E: Model validation

### E1 Simulated soil temperature

Tables E.1, E.2, E.3 and E.4 presents the simulated soil temperature validation statistics at sixteen sensors at sites A, B, C and D respectively.

### E2 simulated variability in soil temperature

### E3 Water table depth

Figure E.4 show the spatial distribution of maximum water table elevation during the simulation period at sites A, B. C and D. Ground surface is at 50 $m$ elevation at all the sites.

**Table D.1.** Soil hydraulic and thermal properties used in the models

| Parameter [Unit] | Moss | Organic | Mineral | Deep organic | Deep mineral | Data source |
|---|---|---|---|---|---|---|
| Porosity [–] | 0.90 | 0.86 | 0.60 | 0.86 | 0.54 | Hinzman et al. (1991) |
| Hydraulic conductivity [m/s] | 1.94 | 1.04 | 0.376 | 1.08 | 0.14 | Hinzman et al. (1991) |
| Bulk density [g/$cm^3$] | 0.15 | 0.18 | 1.39 | 0.18 | 1.33 | Hinzman et al. (1991) |
| VG Alpha ($\alpha$) [1/Pa] | $1.5 \times 10^{-4}$ | $1.5 \times 10^{-4}$ | $1.5 \times 10^{-4}$ | $1.5 \times 10^{-4}$ | $1.5 \times 10^{-4}$ | – |
| VG Lambda ($\lambda$) [–] | 0.23 | 0.95 | 0.33 | 0.95 | 0.33 | – |
| Residual saturation [–] | 0.05 | 0.34 | 0.20 | 0.34 | 0.20 | – |
| $\kappa_l$ [–] | 0.45 | 0.43 | 0.8 | 0.43 | 0.8 | Hinzman et al. (1998) |
| $\kappa_i$ [–] | 1.81 | 1.73 | 3.2 | 1.73 | 3.2 | Hinzman et al. (1998) |
| $\kappa_g$ [–] | 1.81 | 1.73 | 3.2 | 1.73 | 3.2 | Hinzman et al. (1998) |
| $\alpha_\ell$ [–] | 0.45 | 0.45 | 0.45 | 0.45 | 0.45 | – |
| $\alpha_i$ [–] | 0.97 | 0.07 | 0.97 | 0.97 | 0.97 | – |
| Specific heat [$J/kg°C$] | $1.04 \times 10^4$ | $8.65 \times 10^4$ | $2.36 \times 10^3$ | $3.19 \times 10^2$ | $2.46 \times 10^3$ | – |

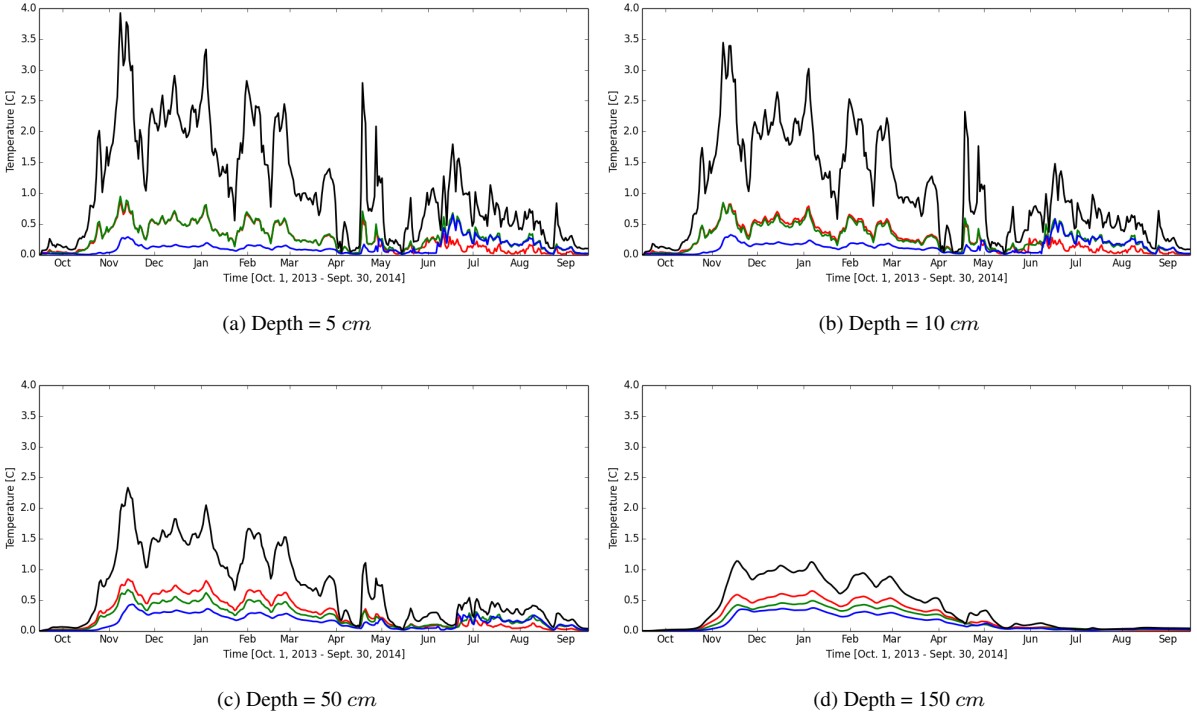

(a) Depth = 5 $cm$

(b) Depth = 10 $cm$

(c) Depth = 50 $cm$

(d) Depth = 150 $cm$

**Figure E.1.** Spatio temporal variability (standard deviation) in simulated soil temperature at Site B.

Red, green and blue lines represent center, rim and trough respectively, and black line represent the standard deviation across the Site B.

**Table E.1.** Model performance statistics at Site A compared to observed soil temperatures

| Depth from surface [$m$] | Center | | | Rim | | | Trough | | |
|---|---|---|---|---|---|---|---|---|---|
| | $RMSE$ | $R^2$ | $Bias$ | $RMSE$ | $R^2$ | $Bias$ | $RMSE$ | $R^2$ | $Bias$ |
| 0.02 | 0.97 | 0.98 | -0.32 | 0.70 | 0.99 | -0.09 | 0.81 | 0.99 | 0.10 |
| 0.05 | 0.73 | 0.99 | -0.06 | 0.80 | 0.99 | 0.13 | 0.91 | 0.99 | 0.48 |
| 0.10 | 0.85 | 0.98 | 0.03 | 0.72 | 0.99 | 0.31 | 1.13 | 0.98 | 0.67 |
| 0.15 | 1.35 | 0.96 | 0.23 | 0.87 | 0.99 | 0.39 | 1.32 | 0.97 | 0.76 |
| 0.20 | 1.52 | 0.94 | 0.28 | 0.90 | 0.99 | 0.43 | 1.47 | 0.97 | 0.89 |
| 0.25 | 1.69 | 0.93 | 0.37 | 0.98 | 0.98 | 0.50 | 1.55 | 0.96 | 0.98 |
| 0.30 | 1.39 | 0.94 | 0.32 | 1.00 | 0.98 | 0.58 | 1.36 | 0.97 | 1.01 |
| 0.35 | 1.34 | 0.94 | 0.31 | 1.02 | 0.98 | 0.64 | 1.35 | 0.98 | 1.07 |
| 0.40 | 1.41 | 0.93 | 0.34 | 1.03 | 0.98 | 0.65 | 1.38 | 0.98 | 1.11 |
| 0.50 | 1.30 | 0.94 | 0.39 | 1.01 | 0.99 | 0.72 | 1.35 | 0.98 | 1.16 |
| 0.60 | 1.27 | 0.94 | 0.44 | 1.01 | 0.99 | 0.78 | 1.38 | 0.98 | 1.23 |
| 0.70 | 1.18 | 0.95 | 0.47 | 1.03 | 0.99 | 0.86 | 1.38 | 0.99 | 1.27 |
| 0.80 | 0.93 | 0.97 | 0.47 | 1.14 | 0.99 | 0.98 | 1.48 | 0.99 | 1.38 |
| 1.00 | 1.08 | 0.96 | 0.59 | 1.17 | 0.99 | 1.08 | 1.57 | 0.99 | 1.52 |
| 1.25 | 1.28 | 0.94 | 0.79 | 1.36 | 0.99 | 1.30 | 1.76 | 0.99 | 1.72 |
| 1.50 | 1.42 | 0.94 | 1.01 | 1.55 | 0.99 | 1.52 | 1.95 | 0.99 | 1.92 |

$RMSE$ = Root Mean Squared Error, $R^2$ = Coefficient of determination, $Bias = negative$ bias indicates cold bias in the model while $positive$ indicates a warm bias

## E4    1-D model simulations at Site A

To evaluate the benefits of 3-D representation of thermal and hydrologic processes over 1-D representation, a reference 1-D simulation was conducted at Site A. To allow for a fair comparison of the 3-D vs 1-D models, we implemented a vertical flow only mode in PFLOTRAN that allows to turn off all lateral flows. The mode was applied at Site A with exactly same mesh, soil properties, initial conditions and forcings as 3-D simulation at Site A (Section 4.1), effectively converting the 3-D modeling domain into 2,305 1-D columns.

Lack of horizontal flows in 1-D simulations show an impact on the soil temperatures during the summer season, when lateral flows are observed in the 3-D model (Figure 12). Difference in temperatures are strongly correlated with the topography of the domain, with high elevation area experiencing warm bias (water and energy not leaving due to lack of horizontal flow), while low elevation areas experience cold bias (water and energy not entering the cell due to lack of horizontal flow) (Figure E.5). Temperature difference ranging -0.55 - 0.85 $^\circ C$ was observed across the Site A during the summer season.

Lack of horizontal flow also effect the simulated estimates of thaw depth across the study region. A difference in maximum thaw depth of -10 $cm$ to +20 $cm$ was observed across the entire study region during the summer season (Figure E.6). These

**Table E.2.** Model performance statistics at Site B compared to observed soil temperatures

| Depth from surface [$m$] | Center | | | Rim | | | Trough | | |
|---|---|---|---|---|---|---|---|---|---|
| | $RMSE$ | $R^2$ | $Bias$ | $RMSE$ | $R^2$ | $Bias$ | $RMSE$ | $R^2$ | $Bias$ |
| 0.02 | 0.99 | 0.98 | 0.06 | 0.54 | 0.99 | -0.05 | 0.83 | 0.98 | -0.07 |
| 0.05 | 0.83 | 0.99 | 0.34 | 0.50 | 0.99 | -0.08 | 0.68 | 0.99 | 0.23 |
| 0.10 | 0.80 | 0.99 | 0.38 | 0.46 | 0.99 | 0.04 | 0.72 | 0.99 | 0.31 |
| 0.15 | 0.90 | 0.99 | 0.43 | 0.77 | 0.99 | 0.14 | 1.11 | 0.98 | 0.49 |
| 0.20 | 0.96 | 0.99 | 0.49 | 0.90 | 0.98 | 0.20 | 1.27 | 0.97 | 0.60 |
| 0.25 | 1.05 | 0.98 | 0.52 | 1.01 | 0.98 | 0.29 | 1.43 | 0.96 | 0.72 |
| 0.30 | 0.95 | 0.98 | 0.59 | 0.84 | 0.98 | 0.27 | 1.20 | 0.97 | 0.70 |
| 0.35 | 0.95 | 0.98 | 0.63 | 0.83 | 0.98 | 0.30 | 1.17 | 0.97 | 0.72 |
| 0.40 | 1.03 | 0.98 | 0.64 | 0.96 | 0.98 | 0.34 | 1.26 | 0.97 | 0.77 |
| 0.50 | 1.02 | 0.98 | 0.71 | 0.98 | 0.98 | 0.39 | 1.23 | 0.98 | 0.81 |
| 0.60 | 1.06 | 0.98 | 0.75 | 1.07 | 0.97 | 0.47 | 1.28 | 0.98 | 0.88 |
| 0.70 | 1.11 | 0.98 | 0.84 | 1.12 | 0.97 | 0.55 | 1.31 | 0.98 | 0.94 |
| 0.80 | 1.10 | 0.99 | 0.97 | 1.06 | 0.98 | 0.66 | 1.22 | 0.99 | 0.99 |
| 1.00 | 1.35 | 0.98 | 1.09 | 1.41 | 0.96 | 0.85 | 1.51 | 0.98 | 1.16 |
| 1.25 | 1.67 | 0.97 | 1.28 | 1.71 | 0.94 | 1.08 | 1.81 | 0.96 | 1.36 |
| 1.50 | 1.89 | 0.96 | 1.49 | 1.95 | 0.92 | 1.34 | 2.09 | 0.94 | 1.63 |

$RMSE$ = Root Mean Squared Error, $R^2$ = Coefficient of determination, $Bias = negative$ bias indicates cold bias in the model while $positive$ indicates a warm bias

differences are primarily due to the lack of lateral flows in 1-D models and demonstrate strong effect of microtopography. High elevations areas (like rims), do not loose water and energy due to lack of lateral flows, and show increased thaw depths, while low elevation areas (like polygons centers) show reduced thaw depths.

### Appendix F:  Active layer thickness observation at BEO

5  Hubbard et al. (2013) and Peterson (2016) collected repeat observations of active layer thickness at a 500 $m$ transect close to (and representative of) our sites A, B, C, D. Peterson (2016) provides the repeat observations collected on July 02, 2014, August 16, 2014, and September 23, 2014. Figure F.1 show the maximum active layer thickness observed at each of the point along the transect. While the average active layer thickness was around 50 $cm$, a significant amount of variability was observed with active layer depths up to 74 $cm$ being observed.

10  **Appendix G:  Observed distribution of snow**

**Table E.3.** Model performance statistics at Site C compared to observed soil temperatures

| Depth from surface [$m$] | Center | | | Rim | | | Trough | | |
|---|---|---|---|---|---|---|---|---|---|
| | $RMSE$ | $R^2$ | $Bias$ | $RMSE$ | $R^2$ | $Bias$ | $RMSE$ | $R^2$ | $Bias$ |
| 0.02 | 0.90 | 0.99 | -0.08 | 0.95 | 0.98 | 0.15 | 0.67 | 0.99 | -0.04 |
| 0.05 | 0.80 | 0.99 | 0.18 | 1.84 | 0.96 | 0.50 | 0.69 | 0.99 | 0.29 |
| 0.10 | 0.63 | 0.99 | 0.12 | 1.23 | 0.98 | 0.35 | 0.78 | 0.99 | 0.34 |
| 0.15 | 0.85 | 0.99 | 0.21 | 0.78 | 0.99 | 0.31 | 1.09 | 0.98 | 0.57 |
| 0.20 | 0.93 | 0.98 | 0.26 | 0.80 | 0.99 | 0.37 | 1.11 | 0.98 | 0.62 |
| 0.25 | 0.98 | 0.98 | 0.29 | 0.80 | 0.99 | 0.36 | 1.16 | 0.98 | 0.68 |
| 0.30 | 0.81 | 0.98 | 0.29 | 0.86 | 0.99 | 0.41 | 1.06 | 0.98 | 0.70 |
| 0.35 | 0.87 | 0.98 | 0.32 | 0.83 | 0.99 | 0.41 | 1.12 | 0.98 | 0.80 |
| 0.40 | 1.03 | 0.97 | 0.38 | 0.80 | 0.98 | 0.40 | 1.16 | 0.98 | 0.84 |
| 0.50 | 0.96 | 0.97 | 0.42 | 0.75 | 0.99 | 0.42 | 1.13 | 0.98 | 0.91 |
| 0.60 | 0.91 | 0.98 | 0.44 | 0.73 | 0.99 | 0.46 | 1.15 | 0.99 | 0.98 |
| 0.70 | 0.87 | 0.98 | 0.53 | 0.73 | 0.99 | 0.52 | 1.17 | 0.99 | 1.06 |
| 0.80 | 0.78 | 0.99 | 0.64 | 0.77 | 0.99 | 0.56 | 1.20 | 0.99 | 1.14 |
| 1.00 | 0.93 | 0.99 | 0.80 | 0.86 | 0.99 | 0.76 | 1.34 | 0.99 | 1.29 |
| 1.25 | 1.13 | 0.99 | 1.02 | 1.02 | 0.99 | 0.95 | 1.58 | 1.00 | 1.52 |
| 1.50 | 1.35 | 0.99 | 1.27 | 1.23 | 0.99 | 1.19 | 1.81 | 1.00 | 1.76 |

$RMSE$ = Root Mean Squared Error, $R^2$ = Coefficient of determination, $Bias$ = $negative$ bias indicates cold bias in the model while $positive$ indicates a warm bias

**Table E.4.** Model performance statistics at Site D compared to observed soil temperatures

| Depth from surface [$m$] | Center | | | Rim | | |
|---|---|---|---|---|---|---|
| | $RMSE$ | $R^2$ | $Bias$ | $RMSE$ | $R^2$ | $Bias$ |
| 0.02 | 0.87 | 0.99 | -0.15 | 1.19 | 0.98 | 0.29 |
| 0.05 | 0.80 | 0.99 | 0.34 | 0.86 | 0.99 | 0.26 |
| 0.10 | 1.03 | 0.98 | 0.45 | 0.96 | 0.98 | 0.36 |
| 0.15 | 1.53 | 0.96 | 0.65 | 1.25 | 0.97 | 0.43 |
| 0.20 | 1.65 | 0.94 | 0.72 | 1.29 | 0.97 | 0.50 |
| 0.25 | 1.78 | 0.93 | 0.80 | 1.32 | 0.96 | 0.53 |
| 0.30 | 1.56 | 0.94 | 0.77 | 1.12 | 0.97 | 0.56 |
| 0.35 | 1.68 | 0.93 | 0.85 | 1.14 | 0.97 | 0.54 |
| 0.40 | 1.61 | 0.94 | 0.84 | 1.12 | 0.97 | 0.56 |
| 0.50 | 1.49 | 0.95 | 0.86 | 1.02 | 0.98 | 0.57 |
| 0.60 | 1.46 | 0.95 | 0.92 | 1.02 | 0.98 | 0.61 |
| 0.70 | 1.39 | 0.96 | 0.95 | 0.96 | 0.98 | 0.63 |
| 0.80 | 1.21 | 0.97 | 0.99 | 0.87 | 0.99 | 0.72 |
| 1.00 | 1.20 | 0.98 | 1.02 | 0.97 | 0.99 | 0.80 |
| 1.25 | 1.47 | 0.97 | 1.27 | 1.13 | 0.98 | 0.94 |
| 1.50 | 1.65 | 0.97 | 1.47 | 1.29 | 0.98 | 1.14 |

$RMSE$ = Root Mean Squared Error, $R^2$ = Coefficient of determination, $Bias$ = $negative$ bias indicates cold bias in the model while $positive$ indicates a warm bias

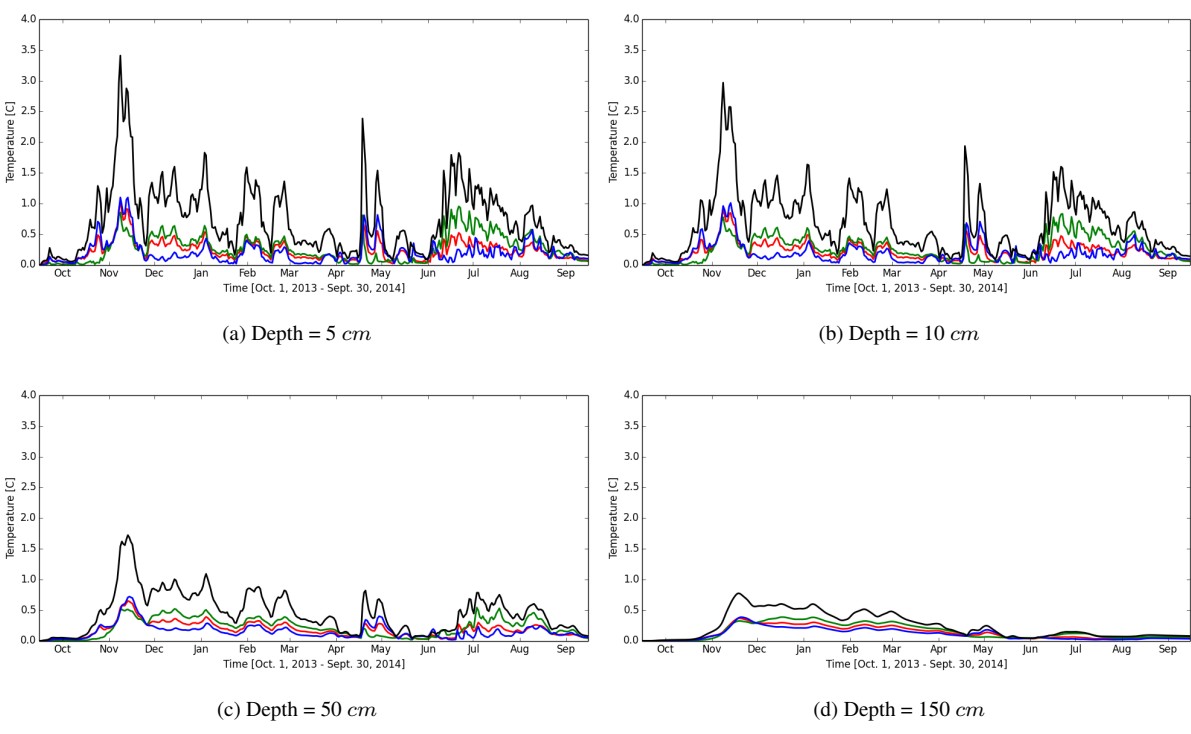

(a) Depth = 5 $cm$

(b) Depth = 10 $cm$

(c) Depth = 50 $cm$

(d) Depth = 150 $cm$

**Figure E.2.** Spatio temporal variability (standard deviation) in simulated soil temperature at Site C.

Red, green and blue lines represent center, rim and trough respectively, and black line represent the standard deviation across the Site C.

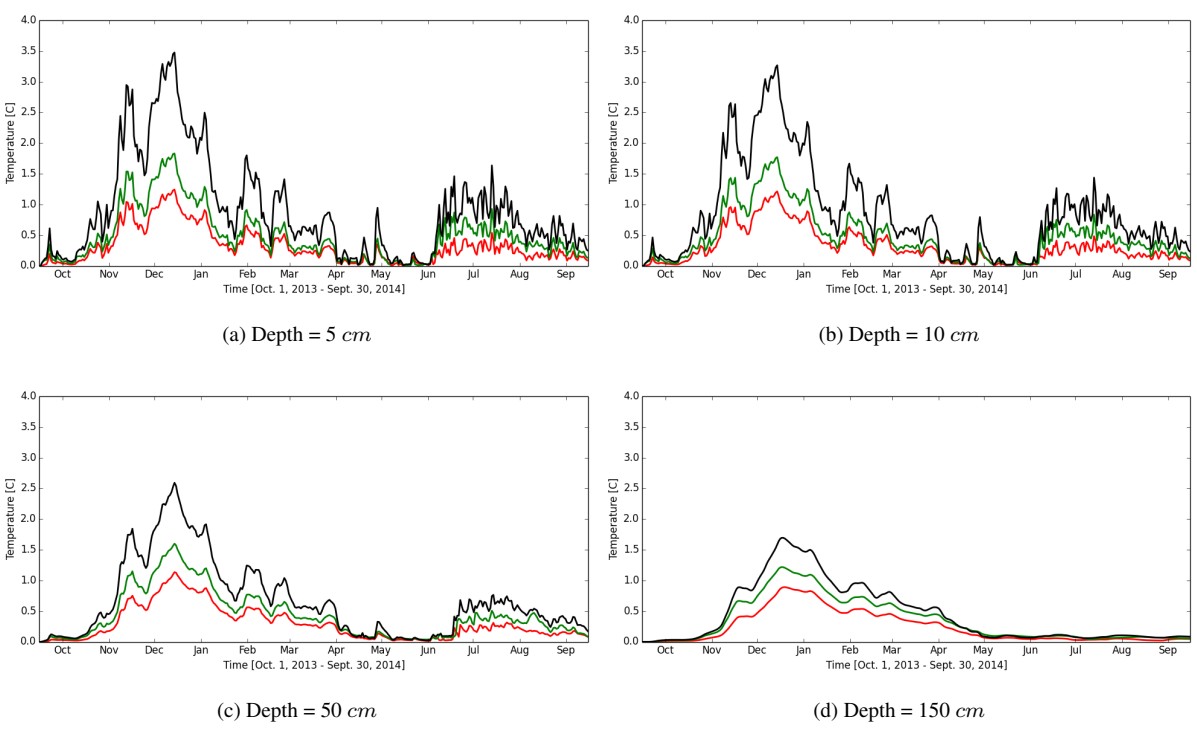

(a) Depth = 5 $cm$

(b) Depth = 10 $cm$

(c) Depth = 50 $cm$

(d) Depth = 150 $cm$

**Figure E.3.** Spatio temporal variability (standard deviation) in simulated soil temperature at Site D.

Red, green and blue lines represent center, rim and trough respectively, and black line represent the standard deviation across the Site D.

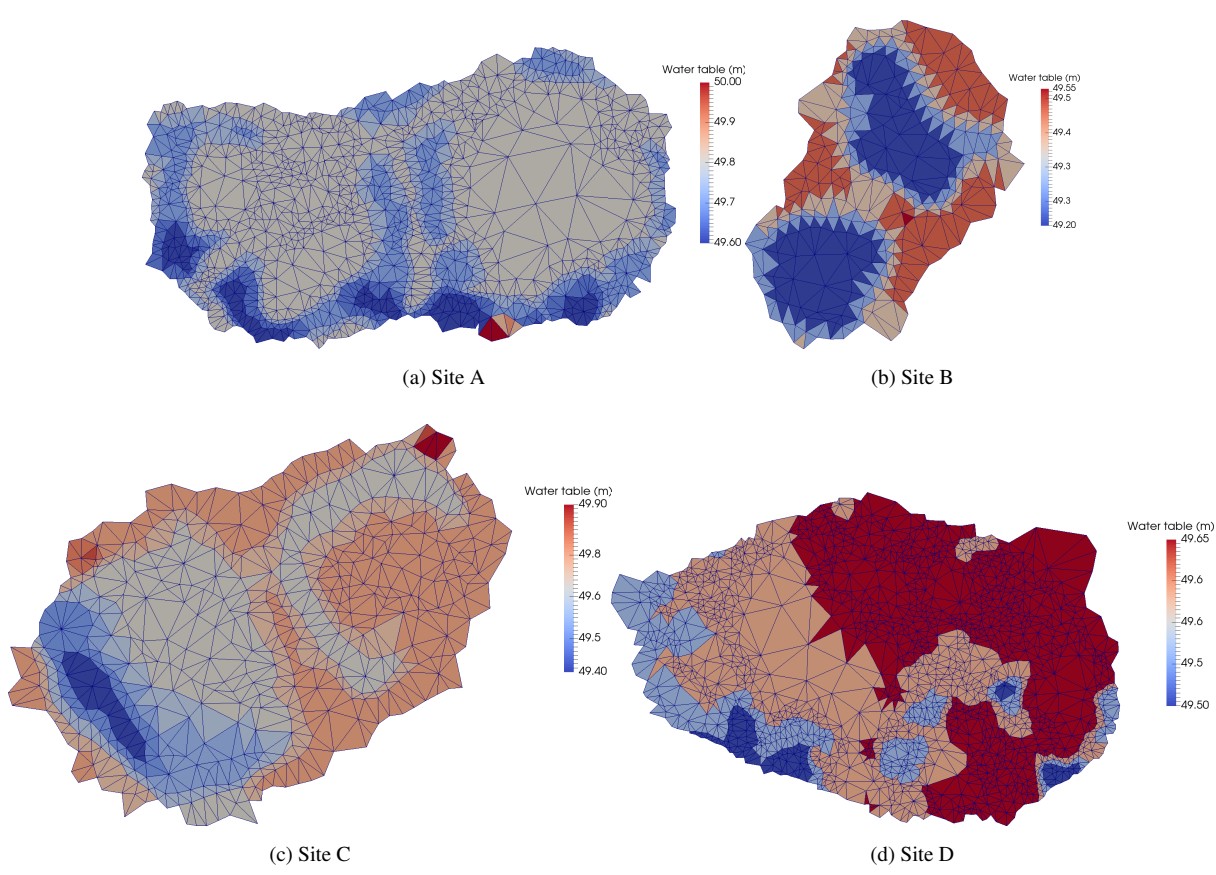

**Figure E.4.** Elevation of maximum water table across the study region. Ground surface is at $50\ m$ elevation. Bold line represents the mean thaw depth across the site, while shaded curve represents standard deviation

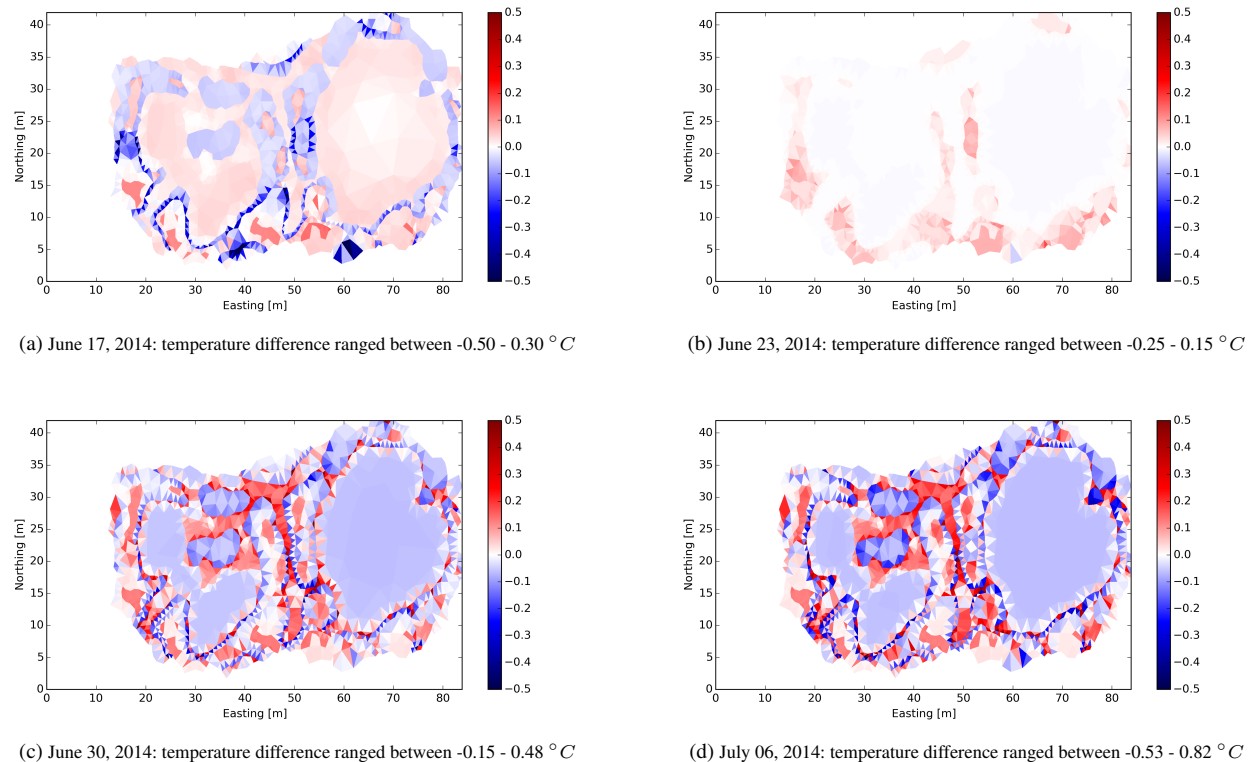

(a) June 17, 2014: temperature difference ranged between -0.50 - 0.30 $^{\circ}C$

(b) June 23, 2014: temperature difference ranged between -0.25 - 0.15 $^{\circ}C$

(c) June 30, 2014: temperature difference ranged between -0.15 - 0.48 $^{\circ}C$

(d) July 06, 2014: temperature difference ranged between -0.53 - 0.82 $^{\circ}C$

**Figure E.5.** Difference in mean ground surface temperature during early summer season in 1-D vs 3-D models. Compared to 3-D model (Figure 12), temperature differences observed in 1-D model are due to the lack of horizontal flows. Positive difference indicate warmer conditions in 1-D model compared to 3-D, while negative temperatures indicate cold bias.

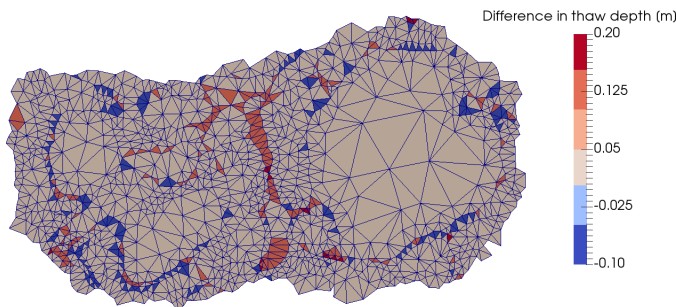

**Figure E.6.** Difference in maximum thaw depth between 1-D vs 3-D simulations. Positive bias indicate higher thaw depth in 1-D simulations compared to 3-D, while negative values indicate lower thaw depths.

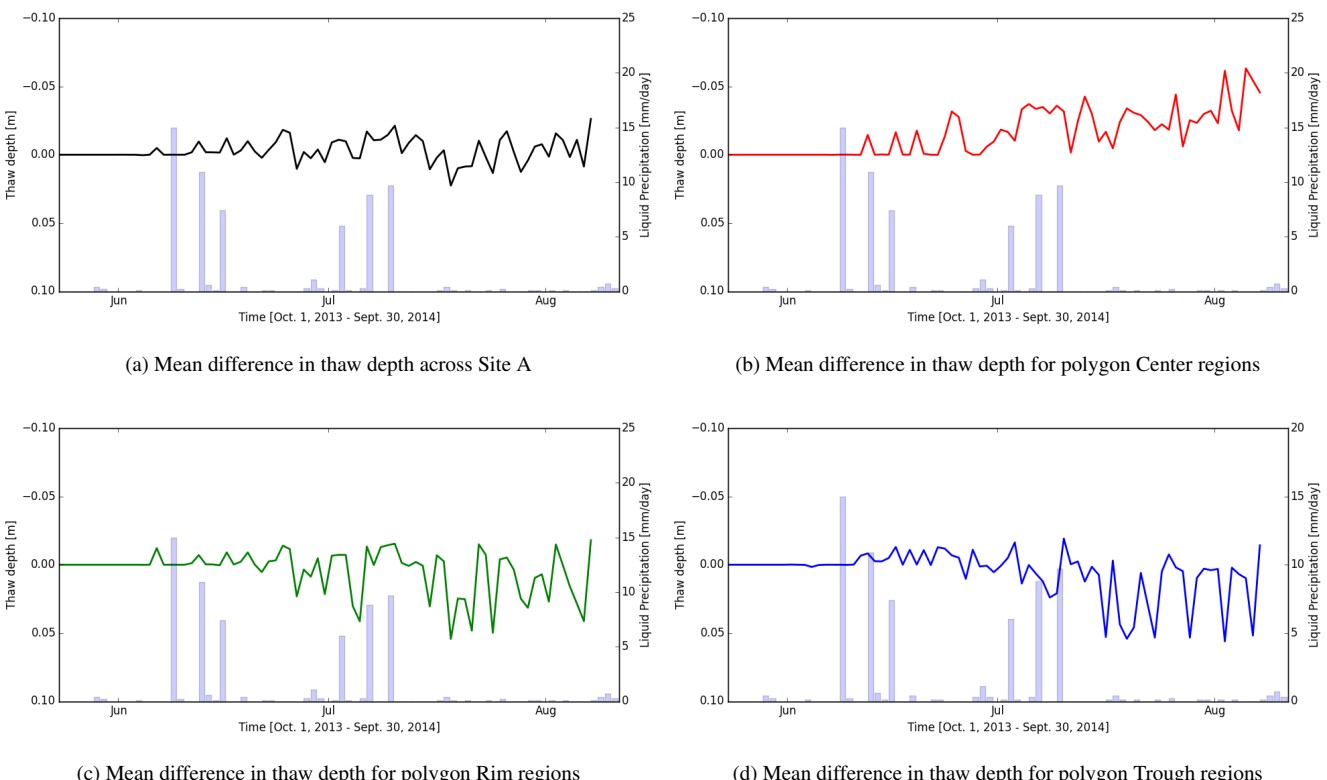

(a) Mean difference in thaw depth across Site A

(b) Mean difference in thaw depth for polygon Center regions

(c) Mean difference in thaw depth for polygon Rim regions

(d) Mean difference in thaw depth for polygon Trough regions

**Figure E.7.** Mean difference in thaw depth between 1-D model compared to 3-D. Positive bias indicate higher thaw depth in 1-D simulations compared to 3-D, while negative values indicate lower thaw depths. Daily precipitation patterns (mm per day) are shown by blue bars.

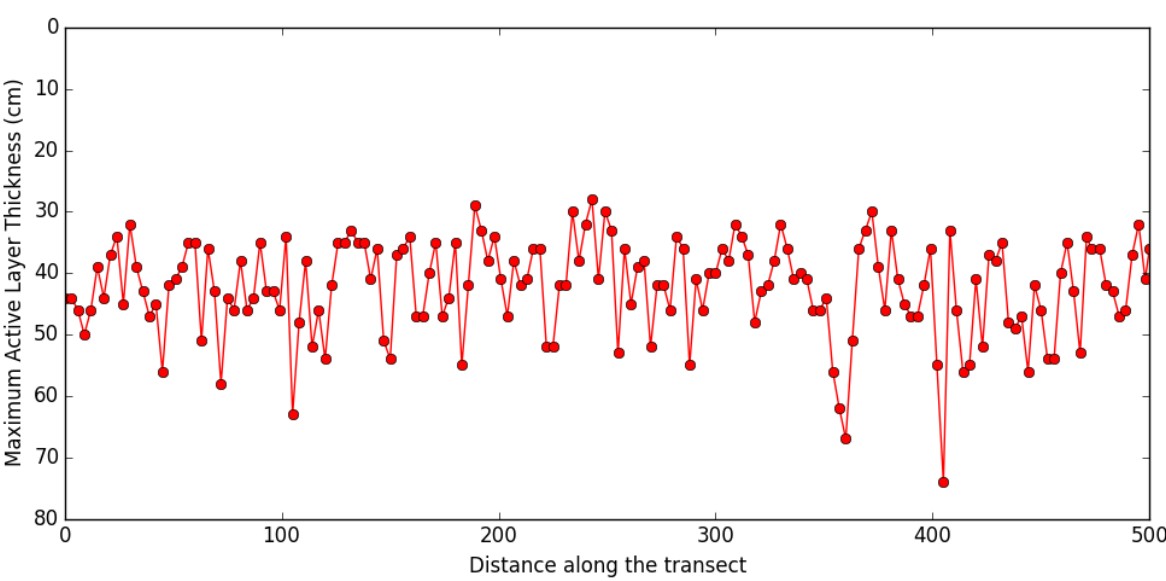

**Figure F.1.** Observed maximum active layer thickness at NGEE-Arctic Site 0 (data collected on July 02, 2014, August 16, 2014, September 23, 2014)

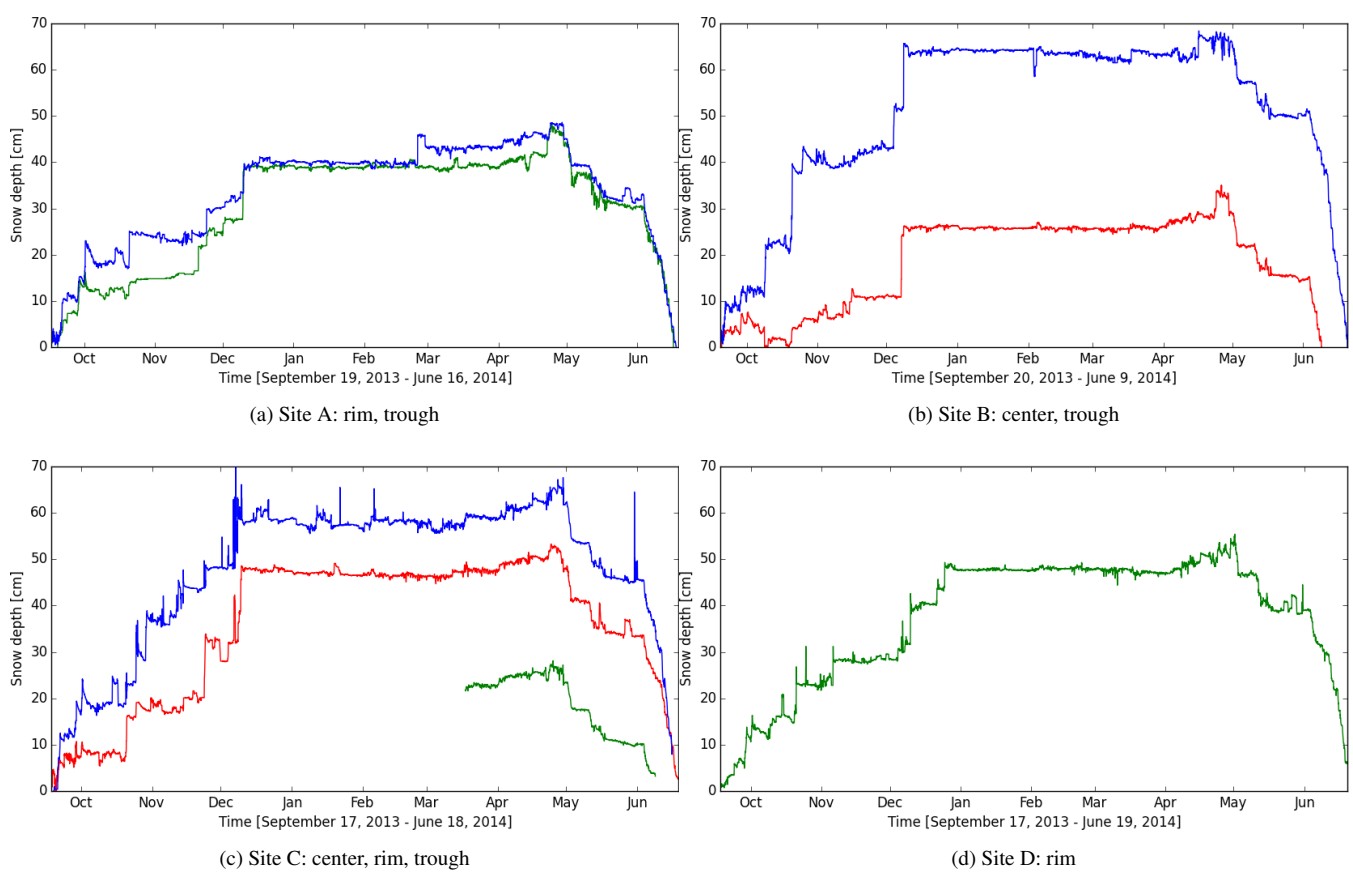

**Figure G.1.** Observed snow depths at different topographic positions center (in red), rim (in green), trough (in blue) at our study sites (Hinzman et al. (2014a).

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
