# Peer review of "Modeling the spatio-temporal variability in subsurface thermal regimes across a low-relief polygonal tundra landscape"

_The Cryosphere, 2016_

## Referee Comment (RC1) · Anonymous Referee #1 · 16 Mar 2016

The manuscript "Modeling the spatio-temporal variability in subsurface thermal regimes across a low-relief polygonal tundra landscape" by Kumar et al. presents high-resolution simulations of the ground thermal regime for tundra polygons in N Alaska. The manuscript seems publishable in TC after major revisions, although some crucial information on the model setup (mainly the exact choice of the lower boundary condition, see details under minor points) is missing, so that the soundness of the approach cannot be finally determined.

Major Points:

1. The simulations are driven by temperature measurements at the surface, as it was common in early 2000s-publications on 1D-heat conduction schemes. Such simple

approaches are generally no longer publishable in a journal like TC today. The authors add 3D-coupling of both heat and water transfer as a new state-of-the-art feature, which removes many of the limitations inherent in earlier approaches. However, the results are rather mediocre at best. While Figs. 8 ff seems to suggest a rather OK fit with measurements, it is actually not the case for active layer thickness (ALT). Although it is hard to see in the figures, the model predicts an ALT of ca. 100 cm at Site A, while it is in reality <50 cm. The same is true for Sites B, C, and D. This is much worse than many traditional 1D-schemes (which may or may not be tuned) and makes the model results virtually useless for further applications, where ALT is of interest. Even worse, the authors can present only a single year, so that it is impossible to determine whether the model can reproduce interannual variability of ALT and ground temperatures, or (e.g. decadal) trends in these variables, which is crucial for applications in the context of climate change. If the bad performance for ALT is indeed true, it must be clearly stated and the reasons investigated and discussed. The conclusion of the manuscript should then be that the model in the presented set-up is NOT suitable for studying the ground thermal regime of polygonal tundra near Barrow.

2. I doubt that the model scheme presented by the authors is scalable due to the computational requirements, so that it could be included in ESM frameworks or similar schemes. So what do we learn from the simulations of the four polygons then? I do not see that the study can provide any new insight in processes or process parameterizations. The main message seems to be that the authors managed to launch a model scheme of unprecedented complexity and computational requirements, but new insight in cryospheric processes and model parameterizations hereof are largely absent in my view. An example for a slightly similar study that does a significantly better job in this respect is Weismüller et al. (2011). They demonstrate a coupled 1D-scheme for heat conduction and water flow and use this model to show that a heat-conduction-only model scheme is more or less sufficient to reproduce the ground thermal regime at their study sites. So it would for instance be highly interesting, if a comparatively simple 1D-heat conduction model (e.g. GIPL2) with year-averaged

ground properties/water contents could yield a similar performance for the center/rim sites. The authors could investigate if 3D-coupling for both heat and water fluxes is really needed, or if 3D-coupling only for water fluxes is needed, or 3D-coupling for heat fluxes only. Such information would be crucial to help designing a robust scalable scheme for representation of polygonal tundra in ESMs.

Ref: Weismüller, J., Wollschläger, U., Boike, J., Pan, X., Yu, Q., and Roth, K.: Modeling the thermal dynamics of the active layer at two contrasting permafrost sites on Svalbard and on the Tibetan Plateau, The Cryosphere, 5, 741-757, doi:10.5194/tc-5-741-2011, 2011.

3. Excess ground ice is a main driver for the evolution of polygons and melting of excess ground ice will lead to changes of the microtopography, which in turn changes the hydrological regime. Are such processes represented in the model scheme? This should be explicitly stated and commented upon in the manuscript. If yes, is the surface stable during the 1-year test period? Are there sites where excess ground ice melt is observed and which could be used to test the model performance? If not, a key element determining the evolution of tundra polygons is missing, and it should be clearly stated that the scheme is not suitable for climate change studies in polygonal tundra.

4. The authors should state more clearly that the presented model is only a very first step towards a physical model of energy and water transfer within tundra polygons. Many of the key drivers of spatial variability in the system are implicitly prescribed by the forcing data (2cm temperature measurements) and not modeled. The authors present curves of snow depths at the various sites, but which factors lead to these differences? (How) could this be modeled? The same is true for vegetation, surface energy balance, evapotranspiration, etc. The authors state that coupling to CLM is planned, but many crucial processes (e.g. wind drift of snow) are not contained in CLM, since it is mainly designed for large-scale applications.

Minor Points:

Fig. 1: strange values given in the color bar, units ([m a.s.l.]?) should be provided.

Fig 2: zero-degree-line missing in b

P. 8, L25: This is a major design flaw of the study which questions the use of such a sophisticated model scheme. Are there plans to obtain such data sets in the future?

Sect. 3.1.3 How about the vertical discretization of the system?

Sect. 3.2: please provide a clear overview of the processes and parameterizations that are considered in the model (and the ones that are not), i.e. heat conduction, saturated flow, unsaturated flow, water vapor transport, how is the freeze curve determined, etc., etc.

P.14, l. 10: Why -1 degree, that appears to be much too warm??

P. 14, l. 10: At what depth is the deep bottom boundary? How does the choice of the lower boundary condition interfere with the selected spin-up-procedure? What are the resulting temperature gradients below the depth of zero annual amplitude? Why not use a heat flux as lower boundary condition, and perform a spin-up so that steady-state conditions are reached in the entire model domain?

P. 14, l. 11: West Dock is several 100 km E of Barrow – how realistic is this assumption and in how far would errors in this temperature compromise the results. Does this temperature roughly correspond to a steady-state condition given the applied surface forcing, or does it introduce a heat sink/source at the bottom? See also comment above.

P. 14, l. 23: What is "thermal hydrology", and why is soil moisture not discussed? This doesn't make sense to me since it is one of the assets of the new model, that the 3D-interplay between moisture and heat fluxes is explicitly considered. The authors should investigate their model results further to show how for instance water fluxes change the thermal properties of the system, which in turn affects heat conduction.

P. 14. L. 5: How is it determined that periodic steady-state conditions are reached?

P. 14, l.9ff: The authors should quantify the magnitude of the advective heat flow, and set them in relation to the conductive heat fluxes. Could a similar model accuracy (considering Figs. 8 ff) be achieved when such fluxes are neglected, as it is done in most model approaches? See major comments.

P. 14, l. 20: Any idea on the accuracy of the precipitation measurements? And why do the authors use daily precipitation, not better resolved in time? Are only daily values available?

Fig. 13: Why is the thermal conductivity for saturated soils higher with some liquid water compared to fully frozen conditions? Is that a real physical process, or an artifact of the employed parameterization? If it's the latter, what is the effect on the simulation results?

P. 20., l. 1ff: I very much agree with this statement! Therefore, many of the results could be strongly influenced by the particular parameterization of the thermal conductivity chosen by the authors. It is a standard parameterization used in many models, but it is not based on first principles and could thus be prone to biases at the particular study site. This is in particular crucial since the authors attempt to reproduce fine-scale ground temperatures, rather than provide a coarse assessment of the ground thermal regime with a simple thermal model.

P. 23, Sect. 5.1: I like that the authors present non-optimal fits between measurements and model, rather than tuning the model to fit the available ground data perfectly. In addition, the authors could/should present a sensitivity analysis at least for some of the crucial model parameters, to show which parameters need to be better estimated in order to improve results. But this is probably difficult due to the model complexity and computational requirements?

P. 24, l. 3: Not sure this is explainable by missing soil properties, etc. The bias is

systematic, and the authors should also investigate and comment on the short time period (1y) of their runs and the way they handle spin-up and the lower boundary condition (see comment above).

P. 26, l. 3: The authors do not provide any quantitative evidence that the hydrology is really reproduced. In a qualitative way, it is (high rims are dry, depressed centers wet, etc.), but quantitative validation information is not presented. Therefore, this statement should be formulated more carefully.

P. 26, l. 7: The authors must present evidence (e.g. sensitivity analysis) that the bias in temperatures is really explainable by deep soil properties (see comments above). If so, they should elaborate on which soil parameters have the largest influence on simulation results.

P. 26, l. 7: The statement on C fluxes is misplaced in this discussion.

---

## Referee Comment (RC2) · Anonymous Referee #2 · 16 Mar 2016

The authors present a new model (PFLOTRAN), in the framework of the New Generation Ecosystem Experiments initiative, which can assess the thermal hydrology for permafrost regions. The authors apply the model at four study sites in Barrow, where they compare the model performance with in situ collected data. The model presented is interesting and physically sound, and the the authors carefully run tests to evaluate model against measurements, On the other hand, some information on the model is missing, and the structure of the paper seems sometimes unclear to me. Therefore I recommend the manuscript for publication after major revisions.

Major points:

1. There is a problem in the data-model comparison that the authors fail to mention.

In Figures 8-11, the authors state that the model performs well against data except from the deeper soil layer. This is not actually true, since the model seems to largely overestimate the soil temperature in the summer months (May - October) also in the upper layers, as measurements and model only agree well at 5 cm depth. This seems to me to be a major issue, since this influences the estimation of key variables as the active layer depth. This problem, of course, would greatly limit the applicability of the model and its potential coupling to any biogeochemical model for estimation, e.g., of methane emissions. The authors should at least discuss this point in the paper. Does PFLOTRAN provide estimations of the active layer depth? If yes, how do they compare against measurements? What can be done to improve this key issue? And why does the model overestimate the temperature in the first place? It is clear that biogeochemistry is outside of the purposes of the paper, but nevertheless there seems to be a relevant issue, since this overestimation occurs in all four sites.

2. Another issue is the interannual variability. The authors state that 2013-2014 was the only year in which the needed information was available at the desired time step. It would be though helpful to show simulations also of other years, if possible, also against partially complete datasets. If only one year is shown, it is difficult to assess the validity of the model in dealing with potential differences in, for example, precipitation regimes. The authors state that the model is not calibrated for a specific year and a specific time, but nevertheless, this issue should be addressed.

3. On this note: how would such a topography-based model react to topography changes due to, say, ice-wedge degradation? This seems to be an important issue, as highlighted by Liljedahl et al., 2016. This is a big issue in the sense that if the model cannot cope to topography transformations, it would only work under present-day conditions, and would not be very useful for future simulations.

4. There is also a problem of scalability of the results. The model seems to be very complex and computationally demanding, since it is a 3D representation of thermal regimes working at a very small spacial scale. The authors mention a potential coupling

with CLM. How this coupling could work is not clear to me. Also, upscaling the detailed information of the 3D model at "just" the ecosystem-scale would be already a significant first step. How do the authors imagine such an upscaling? A stochastic approach would may be help, but how to link the very detailed and small-scale information needed to initialize the model to the larger scale ecosystem dynamics?

Minor points:

1. Page 1, Line 19: blank between words and hyphen.

2. Page 2, Line 20: the link does not help readability. I suggest to insert the link in the Appendix.

3. Page 2, Line 25: Please check the citation, it should be Cresto Aleina et al., 2013.

4. Figure 2: The Paper has a large amount of Figures, and I am not sure if all of them are needed. This Figure, for example, only shows data that are used in the model, but might as well be shown in the Appendix.

5. Page 7, Line 11: "the features": which features? Please, be specific.

6. Page 7, Line 23: the information about the contribution of Dr. Craig Tweedie can be inserted in the Acknowledgments.

7. Page 8, Line 13: You do not enforce any information on polygonal shape. But this information seems to me to be needed for simulation of other properties, such as, for example, water table dynamics. What do you mean then to scale the results to the whole region? Please elaborate this sentence, since it is not clear how this scaling would work.

8. Figure 3 and Figure 4 could be incorporated.

9. Page 11, Line 21 and further: please highlight in the text which ones are the prognostic variables you are evaluating in the equations and which ones the parameters. What is the model time-step? I might have missed it, but it should be clearly stated

here.

10. Page 14: The first paragraph of Chapter 4 is about Methodology (initial and boundary conditions) and should should be moved there.

11. Page 14, Line 17: How long is the spin up?

12. Figures 8 to 11: Please check the style of the caption. There isa a racket at the beginning of the sentence that does not make sense.

13. Figure 12: Why do you only show Site A? Is the behavior representative for the other Sites?

14. Figure 13: Is this figure really needed? In any case, if yes, it should go in the methodology. The color scale should also be changed, to improve readability for color-blind readers.

15. Tables 2 to 5. I do not think that all this information is needed. I suggest to give the information only at the depths showed in Figures 8 to 11. In this way, the authors could summarize the 4 tables in only one, improving readability.

16. Page 21, Line 6: Are such surface processes then implicitly included in the model? If yes how?

17. Page 25, Line 3: Which parameters could be tuned? It would be interesting to understand how tuning which parameter would improve model-data comparison.

18. Figure 15 does not seem to be needed in the text and be moved to the Appendix.

19. Page 27, Lines 4 and further: This discussion can be moved in the Conclusions, since it outlines ongoing and future work, which is not part of the paper.

20. Page 27, Line32: please change "models" in "model".

References:

Liljedahl et al., Pan-Arctic ice-wedge degradation in warming permafrost and its influence on tundra hydrology, Nature Geosciences, 2016. doi:10.1038/ngeo2674

---

## Referee Comment (RC3) · Anonymous Referee #3 · 17 Mar 2016

J. Kumar et al. presented a pilot study using PFLOTRAN model to investigate the role of micro-topography in soil thermal dynamics of different types of ice-wedged tundra, which is important for further studying the responses of large-amount of frozen soil carbon to warming. Field measurements were provided for parameterization and validation. Therefore, I think the topic is important and the method is appropriate . The 3-D modeling is computing-intensive, it is hard, if not impossible, to be coupled in large-scale climate or terrestrial ecosystem models to investigate the effects of fine scale heterogeneity. Meanwhile, it is well-known that micro-topography of ice-wedged tundra ecosystem plays an important role in redistribution of surface water and vegetation growth. Therefore, the manuscript should focus on quantitatively assessing the

role of micro-topography in soil thermal dynamics by comparing sensitivity tests with and without 3-D heat transfer. Unfortunately, the manuscript reached two main conclusions: 1) the 3-D modeling can properly simulate the soil thermal dynamics under the complex micro-topography, which is good; and 2) microtopography is important, which is already known without the 3-D modeling. I believe the authors can do better work with this 3-D model and provide readers more informative results than the current one. I do not recommend publication of the manuscript in the current form.

I would suggest the authors to split the manuscript into two since there were already too much content in the current manuscript. The first deals with model description, model validation and detailed sensitivity tests on only one of the four sites. The second manuscript then deals with differences among different types of ice-wedged tundra ecosystems and upscaling to larger regions.

More specifically, I would suggest to do the following model runs on one site in the first manuscript: 1) shut down the lateral heat flow in the 3-D model and compare the results with those using fully 3-D heat transfer. This work is to demonstrate the importance of lateral heat exchange; 2) in one simulation, use the same soil texture for all micro-topography positions, e.g. rim and center. Compare results with different soil textures; 3) prepare future climate data using GCM outputs under different scenarios. You might also need to convert atmospheric driving to near surface soil temperatures for different micro-topographic components. The long-term simulating might reveal some modeling issues, e.g. lateral boundary conditions; 4) implement different amount of excess ice in soil column to test whether excess ice causes disagreement between simulated or measured soil temperatures.

---

## Short Comment (SC1) · 12 Apr 2016

Comment by Adam L. Atchley, Dylan R. Harp, Ethan T. Coon, Cathy J. Wilson

The authors present an impressive modeling effort investigating 3D water and energy simulations of polygonal tundra. Their research is of interest to the community as it undoubtedly yields insight into the thermal hydrology of polygonal tundra. Furthermore, a modeling effort informed by extensive field measurements provides a unique opportunity to validate and properly shape the process rich models currently being developed for terrestrial Arctic applications. It is for this particular reason, that we are first interested in this manuscript and second concerned with the message in section 5.1. In particular, we refer to line 2-3 on page 25, "At our study sites, while calibration may

compensate for lack of data, it does not improve our understanding of the system."

In section 5.1 the authors provide reasons for not calibrating the model to the observed data available at the study site, specifically that process rich models have high degrees of freedom and therefore are plagued with non-uniqueness (equifinality). In other words, there are multiple combinations of parameters, or more generally model structures (Beven, 2006) that can produce optimized results which fit observed data equally well. While the authors do not quantitatively demonstrate the existence of equifinality here, non-unique parameter combinations certainty exist in this situation, as has been systematically identified for thermal hydrological models at the same site by Atchley et al. (2015) using multi-try calibration and rigorously quantified by Harp et al., (2016) using Null-Space Monte Carlo. However, it is our understanding that the literature addressing equifinalty does not argue for giving up calibration as a lost cause, but rather strongly suggests that additional efforts are required to account for a set or distribution of parameter combinations consistent with observations (e.g. Vrugt et al., 2009, Vrugt and Ter Braak, 2011; Bárdossy, 2007; Tonkin, 2009) and model structural error (Beven, 2005; Clark et al., 2008; Fenicia et al., 2011; Larson et al., 2014). The research behind calibration and model optimization has long since evolved from simple parameter fits to more strategic calibration methods (Hill, 1998). Therefore, we believe that equifinality does not provide a justification to avoid calibration, especially if the objective of the modeling exercise is to improve understanding of system and model behavior. On the contrary, it has been our experience that, while difficult, time-consuming, and computationally expensive, extensive, systematic multi-try calibration can yield important system understanding and identify model capabilities and limitations. The work presented in Atchley et al., (2015) at the same site at the Barrow Environmental Observatory, shows that systematic multi-try calibration can be used as a tool to reduce model structural error and achieve system understanding. For example, calibration efforts led to the recognition of the importance of the representation of snow distribution and depth hoar formation in our models. These insights are not simply better model parameters, but are physical representations of system components;
this effort led to better system understanding. Furthermore, quantifying the equifinality of the combined model and represented system then allows for quantification of model uncertainty, where for example the projected ALT uncertainty attributed to parameter uncertainty can be measured and compared to meteorological and/or climate model uncertainty (Harp et al., 2016). Moreover, the parameter sensitivity quantified by such exercises has, in our opinion, provided valuable information for reducing model uncertainty. Porosity and material thermal conductivity measurements are shown to have the greatest potential to reduce projected ALT uncertainty (Harp et al., 2016), thereby directing which additional field data and process understanding are necessary to reduce uncertainty.

In the context of the model presented in this manuscript, we realize that exhaustive model calibration may be computationally infeasible, and we also do not over look the valuable contribution presented here as the 3D representation of energy and water fluxes in freeze-thaw polygonal tundra indeed pushes the boundaries of process-rich mechanistic modeling. Therefore, it is not our wish to force model calibration and parameter sensitivity analysis on the current manuscript. However, we strongly encourage the authors to reconsider the stated view of model calibration and to discuss how calibration and parameter sensitivity may provide insight into model performance as well as system understanding in polygonal tundra.

References

Atchley, A., S. Painter, D. Harp, E. Coon, C. Wilson, A. Liljedahl, and V. Romanovsky (2015a), Using field observations to inform thermal hydrology models of permafrost dynamics with ATS (v0.83), Geoscientific Model Development, 8, 2701-2722. doi: 10.5194/gmd-8-2701-2015.

Bárdossy, A. "Calibration of hydrological model parameters for ungauged catchments." Hydrology and Earth System Sciences Discussions 11.2 (2007): 703-710.

Beven K. 2005. On the concept of model structural error. Water Science and Technology. 52(6): 167-175

Beven K. 2006. A manifesto for the equifinality thesis. Journal of Hydrology. 320: 18-36. doi:10.1016/j.jhydrol.2005.07.007.

Clark, M. P., Slater, A. G., Rupp, D. E., Woods, R. A., Vrugt, J. A., Gupta, H. V., ... & Hay, L. E. 2008. Framework for Understanding Structural Errors (FUSE): A modular framework to diagnose differences between hydrological models. Water Resources Research, 44(12) W00B02, doi:10.1029/2007WR006735,

Fenicia, F., Kavetski, D., & Savenije, H. H. 2011. Elements of a flexible approach for conceptual hydrological modeling: 1. Motivation and theoretical development. Water Resources Research, 47(11). W11510, doi:10.1029/2010WR010174

Harp, D., A. L. Atchley, S. L. Painter, E. Coon, C. Wilson, V. Romanovsky, and J. Rowland (2015), Effect of soil property uncertainties on permafrost thaw projections: a calibration-constrained analysis, The Cryosphere 10(3), 1-18. doi: 10.5194/tc-10-1-2016.

Hill, Mary C. Methods and guidelines for effective model calibration. Denver, CO, USA: US Geological Survey, 1998.

Kavetski, D., & Fenicia, F. 2011. Elements of a flexible approach for conceptual hydrological modeling: 2. Application and experimental insights. Water Resources Research, 47(11). W11511, doi:10.1029/2011WR010748,

Keating, Elizabeth H., et al. "Optimization and uncertainty assessment of strongly nonlinear groundwater models with high parameter dimensionality." Water Resources Research 46.10 (2010).

Larsen L., Thomas C., Eppinga M. 2014. Exploratory modeling: extracting causality from complexity. EOS, 95(12) 285-292.

Vrugt, Jasper A., Cajo JF Ter Braak, Hoshin V. Gupta, and Bruce A. Robinson. "Equifinality of formal (DREAM) and informal (GLUE) Bayesian approaches in hydrologic modeling?." Stochastic environmental research and risk assessment 23, no. 7 (2009): 1011-1026.

Tonkin, Matthew, and John Doherty. "Calibration‐constrained Monte Carlo analysis of highly parameterized models using subspace techniques." Water Resources Research 45, no. 12 (2009).

Vrugt, Jasper A., and C. J. Ter Braak. "DREAM (D): an adaptive Markov Chain Monte Carlo simulation algorithm to solve discrete, noncontinuous, and combinatorial posterior parameter estimation problems." Hydrology and Earth System Sciences 15, no. 12 (2011).

---

## Author Comment (AC1) · 13 May 2016

The manuscript "Modeling the spatio-temporal variability in subsurface thermal regimes across a low-relief polygonal tundra landscape" by Kumar et al. presents high-resolution simulations of the ground thermal regime for tundra polygons in N Alaska. The manuscript seems publishable in TC after major revisions, although some crucial information on the model setup (mainly the exact choice of the lower boundary condition, see details under minor points) is missing, so that the soundness of the approach cannot be finally determined.

We thank the reviewer for critical review and feedback on our manuscript. We have presented a methodology for modeling of thermal-hydrologic processes in polygonal tundra ecosystem dominated by micro-topographic relief. 3-D modeling approach presented is a first critical step toward study of fine scale processes in tundra ecosystem. We have also presented the application of the model at four sites representing different polygonal types present at our sites at Barrow, Alaska. Our tests and evaluations were designed to validate the model using observations and analyze the model agreements and disagreements to gain insight in the processes at the site, and identify model deficiencies and gaps in data availability which can guide model improvements and data collection to enable improved modeling of the study site.

**Major Points:**

1. The simulations are driven by temperature measurements at the surface, as it was common in early 2000s-publications on 1D-heat conduction schemes. Such simple approaches are generally no longer publishable in a journal like TC today. The authors add 3D-coupling of both heat and water transfer as a new state-of-the-art feature, which removes many of the limitations inherent in earlier approaches. However, the results are rather mediocre at best. While Figs. 8 ff seems to suggest a rather OK fit with measurements, it is actually not the case for active layer thickness (ALT). Although it is hard to see in the figures, the model predicts an ALT of ca. 100 cm at Site A, while it is in reality <50 cm. The same is true for Sites B, C, and D. This is much worse than many traditional 1D-schemes (which may or may not be tuned) and makes the model results virtually useless for further applications, where ALT is of interest. Even worse, the authors can present only a single year, so that it is impossible to determine whether the model can reproduce interannual variability of ALT and ground temperatures, or (e.g. decadal) trends in these variables, which is crucial for applications in the context of climate change. If the bad performance for ALT is indeed true, it must be clearly stated and the reasons investigated and discussed. The conclusion of the manuscript should then be that the model in the presented set-up is NOT suitable for studying the ground thermal regime of polygonal tundra near Barrow.

We have added simulated estimates of thaw depths at four sites (Figures 14, 15, 16, 17). The thaw depths show significant variability across the study region. Compared to an observed average active layer thickness of 50 cm (with thickness of up to 74 cm observed by Hubbard et al. 2013; Peterson 2016; Figure 22), the model is biased towards deeper thaw depths. We have discussed these biases and potential reasons on Page 18 Line 4-Page 21 Line 9.

We were limited to single year period in our study for which all forcing and validation data sets were available. Thus we were not able to investigate the model performance to reproduce interannual variability. We have identified this limitation on Page 27 Lines 1-2 and noted our plan to address this particular issue as more data become available.

While we agree that the presented case studies for our modeling approach show biases in comparison with the observations at the sites, one of our objectives was to identify model deficiencies and data gaps. Thus, one key conclusion from our study is the need for better characterization of site via co-located measurements Page 26 Lines 33-35. "While the models demonstrated the ability to simulate the soil temperature at shallow depths, the deviations from observations in deep soils highlights the need for better soil characterization using deep cores in these ecosystems. Our study also highlights the need for co-located observations for accurate modeling and understanding of the tundra landscape."

2. I doubt that the model scheme presented by the authors is scalable due to the computational requirements, so that it could be included in ESM frameworks or similar schemes. So what do we learn from the simulations of the four polygons then? I do not see that the study can provide any new insight in processes or process parameterizations. The main message seems to be that the authors managed to launch a model scheme of unprecedented complexity and computational requirements, but new insight in cryospheric processes and model parameterizations hereof are largely absent in my view. An example for a slightly similar study that does a significantly better job in this respect is Weismüller et al. (2011). They demonstrate a coupled 1D-scheme for heat conduction and water flow and use this model to show that a heat-conduction-only model scheme is more or less sufficient to reproduce the ground thermal regime at their study sites. So it would for instance be highly interesting, if a comparatively simple 1D-heat conduction model (e.g. GIPL2) with year-averaged ground properties/water contents could yield a similar performance for the center/rim sites. The authors could investigate if 3D-coupling for both heat and water fluxes is really needed, or if 3D-coupling only for water fluxes is needed, or 3D-coupling for heat fluxes only. Such information would be crucial to help designing a robust scalable scheme for representation of polygonal tundra in ESMs.

PFLOTRAN is a massively parallel software which has been extensively tested and optimized on a number of Department of Energy's Leadership Computing Facilities (Hammond et al. 2014, 2012, 2008; Mills et al. 2009). While computationally intensive it has capability to make efficient use of state-of-the-art supercomputing to address large computational problems.

While our interest is to use this process based complex model to inform ESM scale, we do not propose to do that at resolution and complexity detailed in this paper. Work presented here is a building block in the larger scaling philosophy under NGEE-Arctic project which we have discussed on Page 27 Line 10-20.

We disagree that the model doesn't provide any new insights in cryospheric processes. In contrast to traditional 1D heat conduction model, we have presented a 3-dimensional model that resolves the advective and diffusive flow of mass, advective and conductive heat flow in a tightly coupled fashion. The complexity of the model adds to the computational as well as data requirements but it also provides insights in processes which have been often ignored in traditional approaches. We have demonstrated the ability to represent the microtopography effect in a 3-D system. The spatial patterns of Centers, Rims, and Troughs emerge from the topography-aware model are not prescribed. While our case studies were limited to small set of polygons (to allow for comparison with the observations), our approach enables extension of the study region of regional scale.

PFLOTRAN solves a coupled system of PDEs for flow of mass and energy, thus 3-D coupling for heat flux only is not possible. However, the analysis of 3-D flow patterns of mass and energy will give us insights in dominance of lateral vs vertical flow patterns.

Glenn E. Hammond, Peter C. Lichtner, and Richard T. Mills. Evaluating the performance of parallel subsurface simulators: An illustrative example with pflotran. *WATER RESOURCES RESEARCH*, 50:208-228, JAN 2014 doi: 10.1002/2012WR013483

Peter C. Lichtner and Glenn E. Hammond. Using high performance computing to understand roles of labile and nonlabile uranium(vi) on hanford 300 area plume longevity. *VADOSE ZONE JOURNAL*, 11(2), MAY 2012 doi: 10.2136/vzj2011.0097

G. E. Hammond, P. C. Lichtner, C. Lu, and Mills R.T. Pflotran: Reactive flow and transport code for use on laptops to leadership-class supercomputers. In Fan Zhang, G.T. Yeh, and Jack C. Parker, editors, *Groundwater Reactive Transport Models*, pages 141-159. Bentham Science Publishers, Sharjah, UAE, 2012 doi: 10.2174/97816080530631120101

Richard Tran Mills, Glenn E. Hammond, Peter C. Lichtner, Vamsi Sripathi, G. (Kumar) Mahinthakumar, and Barry F. Smith. Modeling subsurface reactive flows using leadership-class computing. In H Simon, editor, *SCIDAC 2009: SCIENTIFIC DISCOVERY THROUGH ADVANCED COMPUTING*, volume 180 of *Journal of Physics Conference Series*, 2009. 5th Annual Conference of Scientific Discovery through Advanced

Computing (SciDAC 2009), San Diego, CA, JUN 14-18, 2009 doi: 10.1088/1742-6596/180/1/012062

Glenn E. Hammond, Peter C. Lichtner, Richard Tran Milis, and Chuan Lu. Toward petascale computing in geosciences: application to the hanford 300 area - art. no. 012051. In RL Stevens, editor, *SCIDAC 2008: SCIENTIFIC DISCOVERY THROUGH ADVANCED COMPUTING*, volume 125 of *JOURNAL OF PHYSICS CONFERENCE SERIES*, page 12051. US DOE Off Sci; Cray; IBM; Intel; HP; SiCortex, 2008. 4th Annual Scientific Discovery through Advanced Computing Conference (SciDAC 2008), Seattle, WA, JUL 13-17, 2008 doi: 10.1088/1742-6596/125/1/012051

Ref: Weismüller, J., Wollschläger, U., Boike, J., Pan, X., Yu, Q., and Roth, K.: Modeling the thermal dynamics of the active layer at two contrasting permafrost sites on Svalbard and on the Tibetan Plateau, The Cryosphere, 5, 741-757, doi:10.5194/tc-5-741-2011, 2011.

3. Excess ground ice is a main driver for the evolution of polygons and melting of excess ground ice will lead to changes of the microtopography, which in turn changes the hydrological regime. Are such processes represented in the model scheme? This should be explicitly stated and commented upon in the manuscript. If yes, is the surface stable during the 1-year test period? Are there sites where excess ground ice melt is observed and which could be used to test the model performance? If not, a key element determining the evolution of tundra polygons is missing, and it should be clearly stated that the scheme is not suitable for climate change studies in polygonal tundra.

Ground ice and other cryostructures are not represented in the model, which has important consequences for modeling of thermal regime. We have included a discussion on this issue on Page  14 Line 18 - Page 15 Line 26.

"For example, while we know that presence of ground ice (like ice wedges, segregated ice, ice lens etc.) is common in the subsurface of Arctic tundra, their representation in the model is completely missing. Lack of representation of these cryostructures are potentially one of the reasons for warmer soils in our simulations. While PFLOTRAN has the ability to capture and model such cryostructures (via heterogeneous subsurface structure and properties but not their formation and evolution), we lack any quantitative data to characterize them for representation in the model. Ongoing efforts under NGEE--Arctic project by Kneafsey and Ulrich 2016 and Dafflon 2016 using X-ray computed tomography (CT) scanner technology on ice cores from BEO can potentially provide detailed 3-dimensional soil structure and density information and help address this missing piece. "

**Evolution of tundra polygons through the process of ice wedge degradations is a complex process which has not been studied widely and approach for modeling them has been limited. Limitation of the presented model to represent dynamic geomorphological changes due to ice-wedge degradation has been noted on Page 26 Line 27-28.**

4. The authors should state more clearly that the presented model is only a very first step towards a physical model of energy and water transfer within tundra polygons. Many of the key drivers of spatial variability in the system are implicitly prescribed by the forcing data (2cm temperature measurements) and not modeled. The authors present curves of snow depths at the various sites, but which factors lead to these differences? (How) could this be modeled? The same is true for vegetation, surface energy balance, evapotranspiration, etc. The authors state that coupling to CLM is planned, but many crucial processes (e.g. wind drift of snow) are not contained in CLM, since it is mainly designed for large-scale applications.

**We agree with that the presented work is a first step towards a process based model for Arctic tundra ecosystem processes. We have added a statement on Page 27 Line 5-10 to highlight that.**
**"While we have not addressed all the deficiencies in model process representation and parameterization identified and reported here in this study, we believe we have developed and presented a process rich modeling framework as a first critical step that would enable such studies. The modeling approach developed in this study will allow accurate modeling of permafrost thermal hydrology and will help identify and guide the future observations required for improved modeling and understanding of the polygonal tundra ecosystem."**

**We have also added a description of NGEE-Arctic scaling philosophy (Page 27 Line 10-25) that the present study will serve as a critical building block for.**

**"While the knowledge gained by developing and evaluating fine-scale 3D simulations is valuable from the perspective of increased understanding of complex process interactions, the explicit long-term goal of the NGEE--Arctic project is to improve predictions of Arctic ecosystem processes at scales relevant to coupled climate and Earth system simulation. One element of our strategy to migrate knowledge across scales is to improve the grid and sub-grid representations in the land model component of our Earth system models to capture observed modes of variability in physical, biological, and biogeochemical processes. For example, our new top-level grid topology for global-scale land modeling follows watershed boundaries instead of the typical and arbitrary rectangular gridcell arrangement (Tesfa et al. 2014). Sub-grid schemes are being developed that represent topographic variation within basins, and our goal is to apply those methods in the micro-topographic setting of polygonal tundra to capture the variation in thermal, hydrologic, and biogeochemical regimes, and interactions with**

vegetation communities. The current study is one step toward identifying the relevant modes of variation among diverse landforms in the polygonal tundra region. Another element of our scaling strategy is to use, to the full extent possible, a common set of modeling tools to construct simulations at various spatial scales. Even though many processes that can be represented explicitly at the finest scales (such as lateral flows of energy and water) must be parameterized for efficiency in a larger-scale simulation, having a common underlying set of equations helps to reduce unintentional loss of information across scales due, for example, to aggregation and disaggregation operators."

**Minor Points:**
**Fig. 1: strange values given in the color bar, units ([m a.s.l.]?) should be provided.**
Yes, elevations are in meter above mean sea level. The caption has been updated to clarify that.

**Fig 2: zero-degree-line missing in b**
The figure has been updated to include the zero degree line

**P. 8, L25: This is a major design flaw of the study which questions the use of such a sophisticated model scheme. Are there plans to obtain such data sets in the future?**

Unfortunately it is a common practice that field observations are often designed and conducted by disciplinary teams based on their objectives. Under the NGEE-Arctic project we are making efforts to coordinate a large multi-disciplinary team of modelers and field scientists. While in this study we employed observations by various teams (and often from literature), many of which were not coordinated or co-located. Under NGEE-Arctic Phase II we are engaging closely with various observational teams for coordinated and co-located observations needed for modeling studies (beyond the one presented here). Thus, we expect to be able to better parameterize our models at our study sites.

**Sect. 3.1.3 How about the vertical discretization of the system?**
Variable resolution was used to discretize the the system vertically, fine near the surface to better resolve the active layer and coarser in the deep subsurface. Table 2 describing the scheme has been added to the text and referenced on Page 10 Line 10.

**Sect. 3.2: please provide a clear overview of the processes and parameterizations that are considered in the model (and the ones that are not), i.e. heat conduction, saturated flow, unsaturated flow, water vapor transport, how is the freeze curve determined, etc., etc.**

Section 3.2 describes the governing equations for flow of mass and energy in variably saturated porous media modeled by PFLOTRAN. Formulation for three phase (ice, water, vapor) flow in PFLOTRAN are based on works by Painter (2011), Painter and Karra (2014), and Karra et al. (2014).

P.14, l. 10: Why -1 degree, that appears to be much too warm??
-1C degree temperature was used as the initial condition for the spin up phase of the simulations and had no effect on the final periodic steady state conditions after spin up. A statement to clarify the choice has been added on Page 13 Line 15-20.
"3-D subsurface models for each of the four sites were initialized by freezing the entire modeling domain at a temperature of -1.0 degC. The models were spun up to a thermal periodic steady state using a time series of mean daily temperatures applied to the top of the domain (ground surface). Spin up simulations were conducted for a period of by cycling annual time series of forcing. Spin up simulations were continued until a periodic steady state was achieved (i.e. close to zero interannual variability in annual thermal regime). Spin up duration of 10 years was used at all the sites and was determined to be sufficient. We conducted a series of initialization simulations by varying initial temperatures at start of spin up and found them to not have any significant impact on the final periodic steady state, besides simulation period required to reach that steady state."

P. 14, l. 10: At what depth is the deep bottom boundary? How does the choice of the lower boundary condition interfere with the selected spin-up-procedure? What are the resulting temperature gradients below the depth of zero annual amplitude? Why not use a heat flux as lower boundary condition, and perform a spin-up so that steady-state conditions are reached in the entire model domain?

The bottom boundary was at the depth of 50 m with variable resolution scheme for vertical discretization (Table 2). The choice of the bottom boundary temperature and the initial temperature has direct impact on the spin up simulation required to achieve thermal steady state condition.
Heat flux measurements in deep permafrost soil were not available at our sites, thus a fixed temperature boundary was used. We selected our model bottom boundary to be at significant depth of 50 m to best avoid the boundary condition effect on the simulated thermal states. We have identified heat flux measurements from deep cores as one of the data set necessary for modeling thermal regimes at the site (Page 25 Line 31-33).

P. 14, l. 11: West Dock is several 100 km E of Barrow – how realistic is this assumption and in how far would errors in this temperature compromise the results. Does this temperature roughly correspond to a steady-state condition given the applied surface forcing, or does it introduce a heat sink/source at the bottom? See also comment Above.

Romanovsky et al. (2010) analyzed permafrost thermal state in the polar northern hemisphere and found the permafrost temperature of -8 to -10 C around 70N latitude. We used temperature from the West Dock which was closest site at comparable latitude studied by Romanovsky et al. 2010. In our simulation, we did not observed any source/sink effect introduced by the bottom boundary at deep 50m.

P. 14, l. 23: What is "thermal hydrology", and why is soil moisture not discussed? This doesn't make sense to me since it is one of the assets of the new model, that the 3D-interplay between moisture and heat fluxes is explicitly considered. The authors should investigate their model results further to show how for instance water fluxes change the thermal properties of the system, which in turn affects heat conduction.

Term "thermal hydrology" refers to coupled processes of water and energy flow. Focus of the presented work was primarily on thermal regimes (Page 14 Line 5-6). We agree that the interplay of soil moisture and heat fluxes is a key process represented in the model. However, on Page 13 Line 29-31 we have acknowledged the caveat of our choice of flow boundary condition due to lack of data on drainage patterns. In Figure 21 we have presented the results of simulated maximum water table across our site. While we have discussed the patterns of soil moisture qualitatively in Sec 4.2 Page 22-23, we have also highlighted the implications of the boundary condition on thermal regimes on Page 17 Line 3.

P. 14. L. 5: How is it determined that periodic steady-state conditions are reached?

The spin up simulations were continued until close to zero interannual variability in the thermal regime was achieved (i.e. annual pattern of soil temperature were same between one year to next). In our analysis, we found 10 years to be sufficient period to achieve that state across all sites.

P. 14, l.9ff: The authors should quantify the magnitude of the advective heat flow, and set them in relation to the conductive heat fluxes. Could a similar model accuracy (considering Figs. 8 ff) be achieved when such fluxes are neglected, as it is done in most model approaches? See major comments.

PFLOTRAN does not compute the advective/conductive heat fluxes separately. Thus we do not have a quantitative way to set and comment on the differences in magnitude. While we anticipate that conductive fluxes dominate the convective, our aim was to improve beyond existing simplified conductive only approaches and provide a process rich model. While simple models can be parameterized and tuned for accuracy against observations, our approach, while complex, would enable understanding of fine scale processes.

**P. 14, l. 20: Any idea on the accuracy of the precipitation measurements? And why do the authors use daily precipitation, not better resolved in time? Are only daily values available?**

**Sensors at the site were checked for accuracy every two weeks to a month interval and precipitation measurements compared with the NOAA CRN facility. However, quantitative accuracy assessments for the precipitation measurements were not available to us. The observations were available at hourly interval, however we aggregated them to daily for input to the models. While model time steps are of seconds to 30 minutes size, daily time series was used to smooth the forcings to the model for faster numerical convergence. Use of finer in time time series for the model is however possible.**

**Fig. 13: Why is the thermal conductivity for saturated soils higher with some liquid water compared to fully frozen conditions? Is that a real physical process, or an artifact of the employed parameterization? If it's the latter, what is the effect on the simulation results?**

**Behavior of thermal conductivity noted by reviewer is an effect of parameterization. The issue is that we have estimates of conductivity in the pure states, that is, pure ice or pure liquid water. However we have soils at a wide range of intermediate states. We have used a standard method for blending the pure state information from published literature by Painter et al. 2012, Karra et al. 2014, Painter and Karra et. al. 2014.**

**P. 20., l. 1ff: I very much agree with this statement! Therefore, many of the results could be strongly influenced by the particular parameterization of the thermal conductivity chosen by the authors. It is a standard parameterization used in many models, but it is not based on first principles and could thus be prone to biases at the particular study site. This is in particular crucial since the authors attempt to reproduce fine-scale ground temperatures, rather than provide a coarse assessment of the ground thermal regime with a simple thermal model.**

**We agree the parameterization of thermal conductivity can be prone to bias. Formulation implemented in our model is based on the Painter 2011.**

**P. 23, Sect. 5.1: I like that the authors present non-optimal fits between measurements and model, rather than tuning the model to fit the available ground data perfectly. In addition, the authors could/should present a sensitivity analysis at least for some of the crucial model parameters, to show which parameters need to be better estimated in order to improve results. But this is probably difficult due to the model complexity and computational requirements?**

**We thank the reviewer for appreciating the value of non-optimal results. We believe that understanding the reasons (missing processes or data) behind the mismatch and**

addressing them should come first before we calibrate the model, which is essential for understanding the processes at the site.
Focus of our presented work was to develop a modeling framework and our case studies without any calibration essentially demonstrates the developed capability using the best off the shelf data sets available to us. Mechanistic representation of processes in our approach does add complexity, increases data requirement for parameterization and computational requirements. However, PFLOTRAN is highly scalable in high performance computing environment and well placed to address such problem by through efficient use of increasingly available computational resources.

While we have not presented a systematic model sensitivity and calibration study, such efforts has been undertaken and recently published by our colleagues Atchley et al, 2015, Harp et. al. 2016 and are complementary to our work. We have added references to their relevant work on Page 25 Line 15-16.

**P. 24, l. 3: Not sure this is explainable by missing soil properties, etc. The bias is systematic, and the authors should also investigate and comment on the short time period (1y) of their runs and the way they handle spin-up and the lower boundary condition (see comment above).**

Page 26 Line 9-11 we have acknowledged these limitations and commented on the need for heat flux observations to better handle the lower boundary conditions.

**P. 26, l. 3: The authors do not provide any quantitative evidence that the hydrology is really reproduced. In a qualitative way, it is (high rims are dry, depressed centers wet, etc.), but quantitative validation information is not presented. Therefore, this statement should be formulated more carefully.**

We agree that we have only presented qualitative and not quantitative evidence for reproducing hydrology. One of our goals in the study was to identify the gaps in observations and motivate future data collections. In absence of data to correctly inform our flow boundary condition, instead of making assumptions to match the observations we have chosen to use a simple no flow boundary condition and analyze the implications of such choice, and thus motivating the need for necessary observations.

**P. 26, l. 7: The authors must present evidence (e.g. sensitivity analysis) that the bias in temperatures is really explainable by deep soil properties (see comments above). If so, they should elaborate on which soil parameters have the largest influence on simulation results.**
Recent complementary work and papers by our NGEE-Arctic colleagues have investigated this issues in detail. Atchley et al. 2015 conducted a comprehensive 1-D model based calibration study at Site C and identified limitations due to lack of lateral flows that can be addressed by a 3-D model. Harp et. al 2016 extended the work of

Atchley et al. 2015 to conduct a Null-Space Monte Carlo method based systematic uncertainty analysis to quantify the effect of soil property uncertainties on permafrost thaw under CESM projected RCP 8.5 scenario from year 2006 to 2100.

**P. 26, l. 7: The statement on C fluxes is misplaced in this discussion.**
Zona et al. 2016 studied cold season emissions in Arctic tundra, highlighting the need for not just understanding active layer thickness, but soil temperature even during winter months. We have revised the text to be more clear (Page 25 Line 25-30).
"Modeling soil temperatures, beyond the high level estimation of thaw depth (or active layer thickeness) is important to understand the thermal regime of permafrost soil and its behavior under warming conditions. For example, during winter seasons even when the soils are completely frozen, variability in in soil temperatures (Figure7, 8, 9, 10) exist and may impact carbon fluxes from the system even during the winter season when soils are frozen (Zona et al. 2016)."

---

## Author Comment (AC2) · 13 May 2016

The authors present a new model (PFLOTRAN), in the framework of the New Generation Ecosystem Experiments initiative, which can assess the thermal hydrology for permafrost regions. The authors apply the model at four study sites in Barrow, where they compare the model performance with in situ collected data. The model presented is interesting and physically sound, and the the authors carefully run tests to evaluate model against measurements, On the other hand, some information on the model is missing, and the structure of the paper seems sometimes unclear to me. Therefore I recommend the manuscript for publication after major revisions.

Major points:

1. There is a problem in the data-model comparison that the authors fail to mention In Figures 8-11, the authors state that the model performs well against data except from the deeper soil layer. This is not actually true, since the model seems to largely overestimate the soil temperature in the summer months (May - October) also in the upper layers, as measurements and model only agree well at 5 cm depth. This seems to me to be a major issue, since this influences the estimation of key variables as the active layer depth. This problem, of course, would greatly limit the applicability of the model and its potential coupling to any biogeochemical model for estimation, e.g., of methane emissions. The authors should at least discuss this point in the paper. Does PFLOTRAN provide estimations of the active layer depth? If yes, how do they
compare against measurements? What can be done to improve this key issue? And why does the model overestimate the temperature in the first place? It is clear that biogeochemistry is outside of the purposes of the paper, but nevertheless there seems to be a relevant issue, since this overestimation occurs in all four sites.

PFLOTRAN solves for soil temperature as primary variables and provides fraction of water in ice, water and vapor states in every computational cell. It does not directly provide the estimates of active layer depth. However, the depth of thawed layer (instead of active layer since we have not addressed BGC) can be estimated using soil temperatures and/or fraction of frozen vs unfrozen water content. We used $0 \degree$C soil temperature to define the threshold temperature for thaw and calculated the thaw depth time series at each site. Figure 14, 15, 16 and 17 show the temporal pattern of thaw depths during the simulation period and the spatial distribution of maximum thaw depth. Spatial distribution of the maximum thaw depths are four sites show strong correlation with the micro-topography. Model does show a warm bias in soil temperatures which translates to a bias towards deeper than observed thaw depths at all the study sites. Tables 3-6, 8-11 presents the validation statistics and bias when compared against observations. We have discussed the potential reasons for these bias in Section 4.1. We believe the primary reason for this bias is the parameterization of soil thermal and hydraulic properties, which in absence of data from our sites, were derived from Hinzman

**et al. 1998 collected at a different site. With the difference in model representation of subsurface structure from the real world, the periodic steady state achieved at the end of model spin up phase had a warm bias. Poorly constrained flow boundary conditions may also be contributing towards this warm bias. Based on our analysis we were able to identify a deficiency in the model (for example missing representation of ground ice), data to characterize the subsurface and to constrain the boundary condition. We plan to use the modeling results to guide the collection of additional data at our sites and improve the simulated estimates of thermal and hydrologic states at our sites.**

**Warm bias in thermal regime would certainly lead to errors in biogeochemical processes, however, we believe these biases can addressed by our planned additional data collection and through model calibration process as demonstrated by Atchley et al. 2015 (Page ).**

2. Another issue is the interannual variability. The authors state that 2013-2014 was the only year in which the needed information was available at the desired time step. It would be though helpful to show simulations also of other years, if possible, also against partially complete datasets. If only one year is shown, it is difficult to assess the validity of the model in dealing with potential differences in, for example, precipitation regimes. The authors state that the model is not calibrated for a specific year and a specific time, but nevertheless, this issue should be addressed.

**We were limited to a single year for this study due the availability of data sets. While we plan to investigate the important issue of interannual variability as data becomes available, we have added a discussion at Page 27 Line 4.**

**"Present study was limited to single year when we had all the necessary data for model forcings and validation available, thus was not able to investigate and address the role of interannual variability. We plan to address this important problem as more data from our sites become available."**

3. On this note: how would such a topography-based model react to topography changes due to, say, ice-wedge degradation? This seems to be an important issue, as highlighted by Liljedahl et al., 2016. This is a big issue in the sense that if the model cannot cope to topography transformations, it would only work under present-day conditions, and would not be very useful for future simulations.

**Ice wedge degradation is an important process in polygonal tundra, however, while presented model can model the effect of topography it currently does not have the capability to model dynamically evolving topography. We have added statement at Page 26 Lines 28-29 to discuss this limitation.**

**"Model developed here does not have the ability to simulate the dynamic changes in microtopography expected due to ice-wedge degradation (Liljedahl et al. 2016})."**

**4. There is also a problem of scalability of the results. The model seems to be very complex and computationally demanding, since it is a 3D representation of thermal regimes working at a very small spacial scale. The authors mention a potential coupling with CLM. How this coupling could work is not clear to me. Also, upscaling the detailed information of the 3D model at "just" the ecosystem-scale would be already a significant first step. How do the authors imagine such an upscaling? A stochastic approach would may be help, but how to link the very detailed and small-scale information needed to initialize the model to the larger scale ecosystem dynamics?**

We agree that upscale transfer of knowledge gained through fine-scale simulations to inform and improve simulations at larger spatial scales is a critical issue. While we can not comprehensively address that research problem in the current study, we have added text to indicate how the fine-scale simulation capability explored in this study contributes to a broader scaling strategy for the NGEE Arctic project. The following text has been added at Page 27 Lines 11-25:

"While the knowledge gained by developing and evaluating fine-scale 3D simulations is valuable from the perspective of increased understanding of complex process interactions, the explicit long-term goal of the NGEE Arctic project is to improve predictions of Arctic ecosystem processes at scales relevant to coupled climate and Earth system simulation. One element of our strategy to migrate knowledge across scales is to improve the grid and sub-grid representations in the land model component of our Earth system models to capture observed modes of variability in physical, biological, and biogeochemical processes. For example, our new top-level grid topology for global-scale land modeling follows watershed boundaries instead of the typical and arbitrary rectangular gridcell arrangement (Tesfa et al. 2015). Sub-grid schemes are being developed that represent topographic variation within basins, and our goal is to apply those methods in the micro-topographic setting of polygonal tundra to capture the variation in thermal, hydrologic, and biogeochemical regimes, and interactions with vegetation communities. The current study is one step toward identifying the relevant modes of variation among diverse landforms in the polygonal tundra region. Another element of our scaling strategy is to use, to the full extent possible, a common set of modeling tools to construct simulations at various spatial scales. Even though many processes that can be represented explicitly at the finest scales (such as lateral flows of energy and water) must be parameterized for efficiency in a larger-scale simulation, having a common underlying set of equations helps to reduce unintentional loss of information across scales due, for example, to aggregation and disaggregation operators."

- Tesfa TK, H Li, LYR Leung, M Huang, Y Ke, Y Sun, and Y Liu. 2014. "A Subbasin-based framework to represent land surface processes in an Earth System Model." Geoscientific Model Development 7(3):947-963. doi:10.5194/gmd-7-947-2014

**Minor points:**

1. Page 1, Line 19: blank between words and hyphen.

Blank space added between words and hyphen at page 1 Line 19. Rest of the document updated for similar consistency as well.

2. Page 2, Line 20: the link does not help readability. I suggest to insert the link in the Appendix.
We have moved the URL to be part of the citation in the bibliography.

3. Page 2, Line 25: Please check the citation, it should be Cresto Aleina et al., 2013.
We have update the citation correctly.

4. Figure 2: The Paper has a large amount of Figures, and I am not sure if all of them are needed. This Figure, for example, only shows data that are used in the model, but might as well be shown in the Appendix.

We have moved several figures to appendix and referenced them in the text as needed. We have also reduced the number of subfigures in Figures 8-11 to show plots at 5cm ,10cm, 50cm, and 150 cm depths only. Tables showing statistics at all 16 sensor depths has been included in the appendix (Tables 8-11 ).

5. Page 7, Line 11: "the features": which features? Please, be specific.
Updated to "polygonal features".

6. Page 7, Line 23: the information about the contribution of Dr. Craig Tweedie can be inserted in the Acknowledgments.
Citation to the public archive of the LiDAR data has been included in the text
Page 5 Line 27.
"High-resolution LiDAR data (25 cm resolution) were collected on October 4, 2005 by Tweedie (2010)."

7. Page 8, Line 13: You do not enforce any information on polygonal shape. But this information seems to me to be needed for simulation of other properties, such as, for example, water table dynamics. What do you mean then to scale the results to the whole region? Please elaborate this sentence, since it is not clear how this scaling would work.

In our approach characterization of microtopographic features (Center, Ridge, Trough) is only used to parameterize the soil properties and determine the surface boundary conditions to applied in the 3-D PFLOTRAN model. Polygonal shape information is not used by PFLOTRAN, rather it simulates the water and energy dynamics from first principles in 3-D where topography drives the simulated patterns. While our test cases were focused on only two polygons to allow for fair comparison with observed data, our approach would allow application of the model (thus scaling up of the problem) to larger regions of interest where high resolution elevation data is available.

8. Figure 3 and Figure 4 could be incorporated.
We have consolidated Figures 3 and 4 in one.

9. Page 11, Line 21 and further: please highlight in the text which ones are the prognostic variables you are evaluating in the equations and which ones the parameters. What is the model time-step? I might have missed it, but it should be clearly stated here.

Liquid pressure and bulk temperature are the prognostic variables in the model.
Page 12 Line 8 "the liquid pressure P and the bulk temperature T are the unknown variables" has been changed to "the liquid pressure P and the bulk temperature T are the primary unknown variables."

The time stepping scheme is Backward-Euler and the time step size is dynamically varied to balance error and the solvability of the nonlinear system. In practice this amounts to a time step size on the order of seconds when a phase transition occurs, and 30 minutes otherwise.
Page 10 Line 22-23 "The PDEs are spatially discretized using a finite volume technique, and backward Euler scheme is used for implicit time discretization."

10. Page 14: The first paragraph of Chapter 4 is about Methodology (initial and boundary conditions) and should should be moved there.

Subsection: Initial and boundary conditions has been moved to Section: Methodology

11. Page 14, Line 17: How long is the spin up?
Spin up simulations were conducted for a period of 10 years. A statement clarifying that has been added to Page 13, Line 17-21.
"Spin up simulations were conducted for a period of by cycling annual time series of forcing. Spin up simulations were continued until a periodic steady state was achieved (i.e. close to zero interannual variability in annual thermal regime). Spin up duration of 10 years was used at all the sites and was determined to be sufficient."

**12. Figures 8 to 11: Please check the style of the caption. There isa a racket at the beginning of the sentence that does not make sense.**
**Caption has been updated.**

**13. Figure 12: Why do you only show Site A? Is the behavior representative for the other Sites?**
**Similar spatio-temporal variability was observed in simulations at the sites. Results for Site A has been included in the text for illustrations, however, we have added the results for other sites too in the Appendix D1 Figure 18.**

**14. Figure 13: Is this figure really needed? In any case, if yes, it should go in the methodology. The color scale should also be changed, to improve readability for colorblind readers.**
**Figure 13 illustrates the approach for calculation of effective thermal conductivity in the model. It has been moved to methodology as Figure 7.**

**15. Tables 2 to 5. I do not think that all this information is needed. I suggest to give the information only at the depths showed in Figures 8 to 11. In this way, the authors could summarize the 4 tables in only one, improving readability.**

**We have reduced the detail to show results only at 5, 10, 50, 150 cm depths. Detailed tables has been added to the Appendix D1.**

**16. Page 21, Line 6: Are such surface processes then implicitly included in the model? If yes how?**
**Surface processes are not represented in the model. We have updated the statement on Page 21 Line 6 to "not modeled in our study" to clarify that.**

**17. Page 25, Line 3: Which parameters could be tuned? It would be interesting to understand how tuning which parameter would improve model-data comparison.**

**A statement pointing to the parameter table have been added in model description. Page 14 Line 4 "Key parameters for the model relevant for current study are described in Table 7." We have also added a reference to relevant parameter sensitivity and calibration analysis study by Atchley et al. 2015.**

**18. Figure 15 does not seem to be needed in the text and be moved to the Appendix.**
**Figure has been moved to the Appendix.**

**19. Page 27, Lines 4 and further: This discussion can be moved in the Conclusions, since it outlines ongoing and future work, which is not part of the paper.**

We have moved the discussion to the conclusion section.

20. Page 27, Line32: please change "models" in "model".
"Models" has been changed to "model".

References:
Liljedahl et al., Pan-Arctic ice-wedge degradation in warming permafrost and its influence on tundra hydrology, Nature Geosciences, 2016. doi:10.1038/ngeo2674

---

## Author Comment (AC3) · 13 May 2016

J. Kumar et al. presented a pilot study using PFLOTRAN model to investigate the role of micro-topography in soil thermal dynamics of different types of ice-wedged tundra, which is important for further studying the responses of large-amount of frozen soil carbon to warming. Field measurements were provided for parameterization and validation. Therefore, I think the topic is important and the method is appropriate .

The 3-D modeling is computing-intensive, it is hard, if not impossible, to be coupled in large-scale climate or terrestrial ecosystem models to investigate the effects of fine scale heterogeneity. Meanwhile, it is well-known that micro-topography of ice-wedged tundra ecosystem plays an important role in redistribution of surface water and vegetation growth. Therefore, the manuscript should focus on quantitatively assessing the role of micro-topography in soil thermal dynamics by comparing sensitivity tests with and without 3-D heat transfer.

We agree with reviewer comments. While the role of micro-topography in tundra ecosystem, effort to develop high resolution process based models for these processes have been limited in the literature. In the presented study we have developed a framework for numerical rigorous high resolution modeling of hydrologic processes. We have demonstrated the ability of the models to capture the effect of micro-topography. We have presented qualitative and quantitative results for a set of simple case studies. Our study provides the base framework to make possible the further detailed analysis suggested by the reviewer. We are actively working on these issues and will report the results in future publications. And we also have made our best effort to archive and publicly release all source codes, workflows, and input/output data sets from the study to allow others interested in conducting such studies.

Unfortunately, the manuscript reached two main conclusions: 1) the 3-D modeling can properly simulate the soil thermal dynamics under the complex micro-topography, which is good; and 2) microtopography is important, which is already known without the 3-D modeling.

The role of microtopography in tundra ecosystems has been studied primarily through field-based investigation. Modeling studies using traditional 1D approaches have been focused around particular sites and thus may or may not be extensible and/or applicable to new or larger regions of interest. The 3-D modeling approach developed in our study enables the modeling of these processes from first principles using a microtopography-resolving model. Patterns due to polygonal microtopography emerge in the model organically, without explicit spatio-temporal flow patterns defined in the model. This enables such studies in any region of interest where high resolution DEM are available to capture the topography in the model. We believe this is key and a new contribution of the presented work beyond published literature.

I believe the authors can do better work with this 3-D model and provide readers more informative results than the current one. I do not recommend publication of the manuscript in the current form.

I would suggest the authors to split the manuscript into two since there were already too much content in the current manuscript. The first deals with model description, model validation and detailed sensitivity tests on only one of the four sites. The second manuscript then deals with differences among different types of ice-wedged tundra ecosystems and upscaling to larger regions.

We thank the reviewer for suggesting a set of analysis where the presented 3-D modeling framework can applied to gain important insights. Under the NGEE-Arctic project we have been working on addressing these questions. However, it's a multi-faceted and complex problem that requires a series of designed steps.
The presented study is the first key step towards that goal and is focused on developing an end-to-end modeling framework to simulate the thermal flow processes in this micro-topographic environments. We applied the methodology at four different study sites of interest and developed methods to synthesize and use the best available data for a field scale study. We have identified a number of key processes and analysis that are needed but were not included but are subject of ongoing work and will be reported in the future.

In close coordination with observation team we currently are working on addressing the data gaps identified in our study which would allow us to conduct a watershed to regional scale application of our framework at Barrow, Alaska.

While we agree with and are working on majority of analysis suggested by the reviewer, we believe we have designed our steps slightly different. The presented approach is our first step that provides us with the numerical framework to conduct the other investigations.

More specifically, I would suggest to do the following model runs on one site in the first manuscript: 1) shut down the lateral heat flow in the 3-D model and compare the results with those using fully 3-D heat transfer. This work is to demonstrate the importance of lateral heat exchange; 2) in one simulation, use the same soil texture for all microtopography positions, e.g. rim and center. Compare results with different soil textures; 3) prepare future climate data using GCM outputs under different scenarios. You might also need to convert atmospheric driving to near surface soil temperatures for different micro-topographic components. The long-term simulating might reveal some modeling issues, e.g. lateral boundary conditions; 4) implement different amount of

excess ice in soil column to test whether excess ice causes disagreement between simulated or measured soil temperatures.

We would also like to highlight a number of recent studies conducted by our NGEE-Arctic colleagues that complement ours and address some of the suggestions identified by the reviewer. Atchley et al. 2015 conducted a comprehensive 1-D model based calibration study at Site C and identified limitations due to lack of lateral flows that can be addressed by a 3-D model. Harp et. al 2016 extended the work of Atchley et al. 2015 to conduct a Null-Space Monte Carlo method based systematic uncertainty analysis to quantify the effect of soil property uncertainties on permafrost thaw under CESM projected RCP 8.5 scenario from year 2006 to 2100. Findings from their study would help guide the parameterization in our model.

---

## Author Comment (AC4) · 13 May 2016

**Comment by Adam L. Atchley, Dylan R. Harp, Ethan T. Coon, Cathy J. Wilson**

The authors present an impressive modeling effort investigating 3D water and energy simulations of polygonal tundra. Their research is of interest to the community as it undoubtedly yields insight into the thermal hydrology of polygonal tundra. Furthermore, a modeling effort informed by extensive field measurements provides a unique opportunity to validate and properly shape the process rich models currently being developed for terrestrial Arctic applications. It is for this particular reason, that we are first interested in this manuscript and second concerned with the message in section 5.1. In particular, we refer to line 2-3 on page 25, "At our study sites, while calibration may compensate for lack of data, it does not improve our understanding of the system." In section 5.1 the authors provide reasons for not calibrating the model to the observed data available at the study site, specifically that process rich models have high degrees of freedom and therefore are plagued with non-uniqueness (equifinality). In other words, there are multiple combinations of parameters, or more generally model structures (Beven, 2006) that can produce optimized results which fit observed data equally well. While the authors do not quantitatively demonstrate the existence of equifinality here, non-unique parameter combinations certainty exist in this situation, as has been systematically identified for thermal hydrological models at the same site by Atchley et al. (2015) using multi-try calibration and rigorously quantified by Harp et al., (2016) using Null-Space Monte Carlo. However, it is our understanding that the literature addressing equifinalty does not argue for giving up calibration as a lost cause, but rather strongly suggests that additional efforts are required to account for a set or distribution of parameter combinations consistent with observations (e.g. Vrugt et al., 2009, Vrugt and Ter Braak, 2011; Bárdossy, 2007; Tonkin, 2009) and model structural error (Beven, 2005; Clark et al., 2008; Fenicia et al., 2011; Larson et al., 2014).

The research behind calibration and model optimization has long since evolved from simple parameter fits to more strategic calibration methods (Hill, 1998). Therefore, we believe that equifinality does not provide a justification to avoid calibration, especially if the objective of the modeling exercise is to improve understanding of system and model behavior. On the contrary, it has been our experience that, while difficult, time-consuming, and computationally expensive, extensive, systematic multi-try calibration can yield important system understanding and identify model capabilities and limitations. The work presented in Atchley et al., (2015) at the same site at the Barrow Environmental Observatory, shows that systematic multi-try calibration can be used as a tool to reduce model structural error and achieve system understanding. For example, calibration efforts led to the recognition of the importance of the representation of snow distribution and depth hoar formation in our models. These insights are not simply better

model parameters, but are physical representations of system components; this effort led to better system understanding. Furthermore, quantifying the equifinality of the combined model and represented system then allows for quantification of model uncertainty, where for example the projected ALT uncertainty attributed to parameter uncertainty can be measured and compared to meteorological and/or climate model uncertainty (Harp et al., 2016). Moreover, the parameter sensitivity quantified by such exercises has, in our opinion, provided valuable information for reducing model uncertainty. Porosity and material thermal conductivity measurements are shown to have the greatest potential to reduce projected ALT uncertainty (Harp et al., 2016), thereby directing which additional field data and process understanding are necessary to reduce uncertainty.

In the context of the model presented in this manuscript, we realize that exhaustive model calibration may be computationally infeasible, and we also do not over look the valuable contribution presented here as the 3D representation of energy and water fluxes in freeze-thaw polygonal tundra indeed pushes the boundaries of process-rich mechanistic modeling. Therefore, it is not our wish to force model calibration and parameter sensitivity analysis on the current manuscript. However, we strongly encourage the authors to reconsider the stated view of model calibration and to discuss how calibration and parameter sensitivity may provide insight into model performance as well as system understanding in polygonal tundra.

We thank Atchley et al. for their interest in our work and insightful comments. We completely agree with the comments above. Model calibration has since long been used a powerful and insightful tool in hydrology and it was certainly not our intention to overlook its power, usefulness and validity. Objective of our presented study was to apply the modeling framework to the study site and synthesize and use available observational data sets for the presented case studies. Work by Atchley et al. 2015 and Harp et al. 2016 provides methodology for calibration, insights in parameter sensitivity and calibrated parameters for three phase thermal hydrology model similar to ours and would certainly complement and guide our future modeling studies. We have updated our manuscript to add a reference to Atchley et al. 2015, Harp et al. 2016 and relevance of their work for our study (Page 24 Lines 16-18).

References
Atchley, A., S. Painter, D. Harp, E. Coon, C. Wilson, A. Liljedahl, and V. Romanovsky (2015a), Using field observations to inform thermal hydrology models of permafrost dynamics with ATS (v0.83), Geoscientific Model Development, 8, 2701-2722. Doi: 10.5194/gmd-8-2701-2015.
Bárdossy, A. "Calibration of hydrological model parameters for ungauged catchments." Hydrology and Earth System Sciences Discussions 11.2 (2007): 703-710.
Beven K. 2005. On the concept of model structural error. Water Science and Technology. 52(6): 167-175

Beven K. 2006. A manifesto for the equifinality thesis. Journal of Hydrology. 320: 18-36. doi:10.1016/j.jhydrol.2005.07.007.

Clark, M. P., Slater, A. G., Rupp, D. E., Woods, R. A., Vrugt, J. A., Gupta, H. V., ... & Hay, L. E. 2008. Framework for Understanding Structural Errors (FUSE): A modular framework to diagnose differences between hydrological models. Water Resources Research, 44(12) W00B02, doi:10.1029/2007WR006735,

Fenicia, F., Kavetski, D., & Savenije, H. H. 2011. Elements of a flexible approach for conceptual hydrological modeling: 1. Motivation and theoretical development. Water Resources Research, 47(11). W11510, doi:10.1029/2010WR010174

Harp, D., A. L. Atchley, S. L. Painter, E. Coon, C. Wilson, V. Romanovsky, and J. Rowland (2015), Effect of soil property uncertainties on permafrost thaw projections: a calibration-constrained analysis, The Cryosphere 10(3), 1-18. doi: 10.5194/tc-10-1-2016.

Hill, Mary C. Methods and guidelines for effective model calibration. Denver, CO, USA: US Geological Survey, 1998.

Kavetski, D., & Fenicia, F. 2011. Elements of a flexible approach for conceptual hydrological modeling: 2. Application and experimental insights. Water Resources Research, 47(11). W11511, doi:10.1029/2011WR010748,

Keating, Elizabeth H., et al. "Optimization and uncertainty assessment of strongly nonlinear groundwater models with high parameter dimensionality." Water Resources Research 46.10 (2010).

Larsen L., Thomas C., Eppinga M. 2014. Exploratory modeling: extracting causality from complexity. EOS, 95(12) 285-292.

Vrugt, Jasper A., Cajo JF Ter Braak, Hoshin V. Gupta, and Bruce A. Robinson. "Equi-finality of formal (DREAM) and informal (GLUE) Bayesian approaches in hydrologic modeling?." Stochastic environmental research and risk assessment 23, no. 7 (2009): 1011-1026.

TCD

Tonkin, Matthew, and John Doherty. "Calibration‐constrained Monte Carlo analysis of highly parameterized models using subspace techniques." Water Resources Research 45, no. 12 (2009).

Interactive
comment

Vrugt, Jasper A., and C. J. Ter Braak. "DREAM (D): an adaptive Markov Chain Monte Carlo simulation algorithm to solve discrete, noncontinuous, and combinatorial posterior parameter estimation problems." Hydrology and Earth System Sciences 15, no. 12 (2011).

---

## Author Response (AR3)

Dear authors,

I am very pleased with the revised manuscript. In addition to the final comments of one reviewer, please prepare the the format of your paper following the guidelines: http://www.the-cryosphere.net/for\_authors/manuscript\_preparation.html (for example, correct labeling of the figures in text and appendix). Furthermore, the new added text needs English language correction. Sincerely, Julia

We have corrected the numbering for the figures and tables in the appendix as per the manuscript preparation guidelines. We have also checked and corrected for spelling and grammatical errors throughout the manuscript.

**Reviewer 1**

Thanks for the replies to my revisions - in general, I am happy with the revised version of the manuscript. Please consider the two additional points below.

**We thank the reviewer for the comments which help improve the quality of the paper.**

p. 19. l. 4. Please replace "which will have important implications" by "which may have implications" - it is not at all clear if the differences in temperature and active layer thickness correspond to reality, so it is not appropriate to state at this stage that they will have implication for biogeochemistry. "Might"/"may" is OK.

We have updated the text in the manuscript as suggested on page 19 line 1.

p. 20, l. 9 ff: This is only true for models that have a coupled energy and water cycle. For this reason, many permafrost models (e.g. GIPL2, CryoGrid2) prescribe a fixed ground profile including soil moisture. This "effective" soil moisture profile (which is partly a result of the lateral water fluxes) might at least capture some of the effects mentioned by the authors, and in practice can give quite robust results (soil moisture profile measurements are often more reliable/available than precipitation+hydraulic conductivity measurements in the Arctic). I agree that a fully coupled model, as presented by the authors, is desirable from a conceptual point of view, though.

We agree with the comment and as noted by the reviewer focus of present work was to develop coupled model for water and energy cycle that allow improved mechanistic understanding of the system.

**Modeling the spatio-temporal variability in subsurface thermal regimes across a low-relief polygonal tundra landscape**

Jitendra Kumar1, Nathan Collier2, Gautam Bisht3, Richard T. Mills4, Peter E. Thornton1, Colleen M. Iversen1, and Vladimir Romanovsky5

[revised manuscript text omitted]